# Achieving Logarithmic Regret in KL-Regularized Zero-Sum Markov Games

**Anupam Nayak** [1]  **Tong Yang** [1]  **Osman Yağan** [1]  **Gauri Joshi** [1]  **Yuejie Chi** [2]

## Abstract

Reverse Kullback–Leibler (KL) divergence-based regularization with respect to a fixed reference policy is widely used in modern reinforcement learning to preserve the desired traits of the reference policy and sometimes to promote exploration (using uniform reference policy, known as entropy regularization). Beyond serving as a mere anchor, the reference policy can also be interpreted as encoding prior knowledge about good actions in the environment. In the context of alignment, recent game-theoretic approaches have leveraged KL regularization with pretrained language models as reference policies, achieving notable empirical success in self-play methods. Despite these advances, the theoretical benefits of KL regularization in game-theoretic settings remain poorly understood. In this work, we develop and analyze algorithms that provably achieve improved sample efficiency under KL regularization. We study both two-player zero-sum Matrix games and Markov games: for Matrix games, we propose OMG, an algorithm based on best response sampling with optimistic bonuses, and extend this idea to Markov games through the algorithm SOMG, which also uses best response sampling and a novel concept of superoptimistic bonuses. Both algorithms achieve a logarithmic regret in $T$ that scales inversely with the KL regularization strength $\beta$ in addition to the traditional $\widetilde{\mathcal{O}}(\sqrt{T})$ regret without the $\beta^{-1}$ dependence.

## 1. Introduction

Multi-agent reinforcement learning (MARL) has emerged as a key framework for modeling strategic interactions among multiple decision makers, providing a powerful tool for analyzing both cooperative and competitive dynamics in domains such as robotics, game playing, and intelligent systems (Busoniu et al., 2008). A fundamental and well-studied case of competitive interactions is the finite-horizon two-player zero-sum Markov game (Shapley, 1953), where agents share a common state, the transition dynamics depend on both agents' actions, and the stagewise rewards sum to zero. The matrix game is a further special case corresponding to the one-step setting (horizon $H = 1$) with no state transitions. Considerable progress has been made in designing sample-efficient online learning algorithms for both zero-sum matrix games (O'Donoghue et al., 2021; Yang et al., 2025a) and Markov games (Bai et al., 2020; Bai & Jin, 2020; Jin et al., 2022; Liu et al., 2021; Xie et al., 2023; Chen et al., 2022; Huang et al., 2022; Cai et al., 2023), leading to nearly optimal rates and a deeper understanding of the computational and statistical challenges inherent in multi-agent systems. Most existing works assume agents learn from scratch, starting with random policies and no knowledge of the environment. This neglects practical settings where prior demonstrations, expert policies, or structural knowledge could accelerate learning and improve performance.

Modern deep reinforcement learning algorithms often use some form of KL or entropy regularization to encourage exploration or to incorporate prior knowledge from a reference policy (Schulman et al., 2015; Haarnoja et al., 2018; Mnih et al., 2016), often initialized via imitation learning from expert demonstrations. These techniques have recently gained substantial attention due to their success in post-training large language models (LLMs) with RL, using either preference feedback (Ouyang et al., 2022) or a learned verifier/reward model (Guo et al., 2025). In this setting, the pretrained LLM serves as the reference policy. Game-theoretic alignment methods and self-play relying on KL regularization (Calandriello et al., 2024; Ye et al., 2024; Munos et al., 2024; Tiapkin et al., 2025; Zhang et al., 2025c; Chen et al., 2024; Wang et al., 2025; Shani et al., 2024; Yang et al., 2025b; Park et al., 2025a) have demonstrated superior empirical performance in reducing over-optimization and improving sample efficiency (Zhang et al., 2025b; Son et al., 2024). Within this paradigm, self-play optimization is framed as a two-player game, where models iteratively improve using their own responses by solving for the Nash Equilibrium (NE) (Nash Jr, 1950) of the regularized game, also known as the Quantal Response Equilibrium (QRE)

---

[1]Carnegie Mellon University. [2]Department of Statistics and Data Science, Yale University. Correspondence to: Anupam Nayak <anupamn@andrew.cmu.edu>.

*Proceedings of the 43rd International Conference on Machine Learning*, Seoul, South Korea. PMLR 306, 2026. Copyright 2026 by the author(s).

(McKelvey & Palfrey, 1995). Under the full information setting, the computational benefits of KL regularization are well understood in terms of fas ter convergence to the NE of the regularized game (Cen et al., 2023; 2024; Zeng et al., 2022).

However, their sample efficiency gains over unregularized methods remains poorly understood since these analyses that demonstrate superior performance under KL regularization assume access to the ground-truth payoff function/oracle. None address the practical setting where the reward function/transition model is unknown and must be learned online simultaneously via exploration using adaptive queries in a *sample-efficient* manner (known as online learning under bandit feedback). Recent work has established logarithmic regret for single-agent settings under KL regularization in the bandit feedback regime (Tiapkin et al., 2024; Zhao et al., 2025b; Foster et al., 2025). In contrast, no such results exist for game-theoretic settings, where current analyses under KL regularization (Ye et al., 2024; Yang et al., 2025a) still maintain $\mathcal{O}(\sqrt{T})$ regret, matching the unregularized case. In this paper, we develop algorithms to close this gap and answer the following question:

*Can we design learning algorithms that, when equipped with KL regularization, achieve provably superior sample efficiency in game-theoretic settings?*

**Our Contributions:** In this work, we develop provably efficient algorithms for competitive games that achieve logarithmic regret in the number of episodes $T$ under KL-regularized settings, in contrast to the standard $\mathcal{O}(\sqrt{T})$ regret typically obtained in unregularized settings. Under KL regularization, the best response of a player to a fixed opponent strategy admits a Gibbs distribution with closed-form expression that depends on the environment parameters to be estimated and the opponent's fixed strategy, both in matrix and Markov games. Our algorithms systematically leverage this property by collecting best-response pairs and exploiting the resulting structure. For matrix games, we design algorithms based on *optimistic* bonuses, while for Markov games, we introduce an algorithm based on a novel *super-optimistic* bonus to achieve logarithmic regret dependent on the regularization strength ($\beta > 0$). Given $\delta \in (0, 1)$,

• For two-player zero-sum matrix games, in Section 2, we propose OMG (Algorithm 1) based on *optimistic bonuses* and *best response sampling*, which achieves with probability at least $1 - \delta$, a regularization-dependent regret of $\mathcal{O}(\beta^{-1} d^2 \log^2(T/\delta))$ and a regularization-independent regret of $\mathcal{O}(d\sqrt{T}\log(T/\delta))$, where $d$ is the feature dimension and $T$ is the number of iterations.

• For two-player zero-sum Markov games, in Section 3, we propose SOMG (Algorithm 2), which learns the NE via solving stage-wise zero-sum matrix games using *best-*

*response sampling* and a novel concept of *super-optimistic bonuses*. These bonuses are chosen such that the superoptimistic $Q$-function exceeds its standard optimistic estimate. With probability at least $1 - \delta$, SOMG achieves a linear in $\beta^{-1}$ logarithmic regret of $\tilde{\mathcal{O}}(\beta^{-1} d^3 H^7 \log^2(dT/\delta))$ alongside a standard $\tilde{\mathcal{O}}(d^{3/2}H^3\sqrt{T})$, where $d$ is the feature dimension, $H$ is the horizon length, and $T$ is the number of episodes.

To the best of our knowledge, this is the first work to establish logarithmic regret guarantees and sample complexities for learning an $\varepsilon$-NE that only scale linearly in $1/\varepsilon$ in any KL regularized game-theoretic setting.[1] Table 1 summarizes our results against prior work. Discussion of related works and full proofs are deferred to the appendix.

**Notation:** For $n \in \mathbb{N}^+$, we use $[n]$ to denote the index set $\{1, \cdots, n\}$. We use $\Delta^n$ to denote the $n$-dimensional simplex, i.e., $\Delta^n := \{x \in \mathbb{R}^n : x \geq 0, \sum_{i=1}^n x_i = 1\}$. The Kullback-Leibler (KL) divergence between two distributions $P$ and $Q$ is denoted by $\mathrm{KL}(P \| Q) := \sum_x P(x) \log \frac{P(x)}{Q(x)}$. For a matrix $M \in \mathbb{R}^{m \times n}$, we denote by $M(i, :)$ its $i$-th row and by $M(:, j)$ its $j$-th column. We use $\mathcal{O}(\cdot)$ to denote the standard order-wise notation and $\tilde{\mathcal{O}}(\cdot)$ is used to denote order-wise notation which suppresses any logarithmic dependencies.

## 2. Two-Player Zero-Sum Matrix Games

### 2.1. Problem Setup

We first consider two-player zero-sum matrix games as the foundation of our algorithmic framework. The KL-regularized payoff function is given as

$$f^{\mu,\nu}(A) = \mu^\top A \nu - \beta\mathrm{KL}(\mu\|\mu_{\mathrm{ref}}) + \beta\mathrm{KL}(\nu\|\nu_{\mathrm{ref}}), \quad (1)$$

where $\mu \in \Delta^m$ (resp. $\nu \in \Delta^n$) denotes the policy of the max (resp. min) player. The reference policy $\mu_{\mathrm{ref}} \in \Delta^m$ (resp. $\nu_{\mathrm{ref}} \in \Delta^n$) encodes prior strategies for the max (resp. min) player and is used to incorporate prior knowledge about the game (e.g., pretrained policies). Here, $A \in \mathbb{R}^{m \times n}$ is the true (unknown) payoff matrix and $\beta \geq 0$ is the regularization parameter. The Nash Equilibrium (NE) $(\mu^\star, \nu^\star)$ is defined as the solution of the following saddle-point problem.

$$\mu^\star = \arg\max_{\mu \in \Delta^m} \min_{\nu \in \Delta^n} f^{\mu,\nu}(A) \qquad (2a)$$

$$\nu^\star = \arg\min_{\nu \in \Delta^n} \max_{\mu \in \Delta^m} f^{\mu,\nu}(A) \qquad (2b)$$

For the NE policies $(\mu^\star, \nu^\star)$ and all $\mu \in \Delta^m, \nu \in \Delta^n$ we have

$$f^{\mu,\nu^\star}(A) \leq f^{\mu^\star,\nu^\star}(A) \leq f^{\mu^\star,\nu}(A). \qquad (3)$$

---

[1]The sample complexities follow using standard regret-to-batch conversion for the time-averaged policy.

| Problem | Algorithm | Setting | Regret | Sample Comp. |
|---|---|---|---|---|
| Matrix Games | Matrix-UCB(O'Donoghue et al., 2021) | Unreg. | $\widetilde{\mathcal{O}}(d\sqrt{T})$ | $\widetilde{\mathcal{O}}(d^2/\varepsilon^2)$ |
| | VMG (Yang et al., 2025a) | Unreg. | $\widetilde{\mathcal{O}}(d\sqrt{T})$ | $\widetilde{\mathcal{O}}(d^2/\varepsilon^2)$ |
| | | Reg. | $\widetilde{\mathcal{O}}(d\sqrt{T})$ | $\widetilde{\mathcal{O}}(d^2/\varepsilon^2)$ |
| | OMG (Algorithm 1) | Unreg. | $\widetilde{\mathcal{O}}(d\sqrt{T})$ | $\widetilde{\mathcal{O}}(d^2/\varepsilon^2)$ |
| | | Reg. | $\min\left\{\widetilde{\mathcal{O}}(d\sqrt{T}),\ \mathcal{O}(\beta^{-1}d^2\log^2(T))\right\}$ | $\min\left\{\widetilde{\mathcal{O}}(d^2/\varepsilon^2),\ \widetilde{\mathcal{O}}(\beta^{-1}d^2/\varepsilon)\right\}$ |
| Markov Games | VMG (Yang et al., 2025a) | Reg. | $\widetilde{\mathcal{O}}(dH^{3/2}\sqrt{T})$ | $\widetilde{\mathcal{O}}(d^2H^3/\varepsilon^2)$ |
| | SOMG (Algorithm 2) | Reg. | $\min\left\{\widetilde{\mathcal{O}}\left(d^{3/2}H^3\sqrt{T}\right),\ \widetilde{\mathcal{O}}(\beta^{-1}d^3H^7\log^2(T))\right\}$ | $\min\left\{\widetilde{\mathcal{O}}\left(d^3H^6/\varepsilon^2\right),\ \widetilde{\mathcal{O}}(\beta^{-1}d^3H^7/\varepsilon)\right\}$ |

*Table 1.* Summary of results: For uniformity, we report all sample complexities (number of samples needed to learn $\varepsilon$-NE) in terms of the number of episodes $T$, results from (O'Donoghue et al., 2021) are translated from tabular to linear function approximation. "Reg." refers to the case with the regularization parameter $\beta$ and bounds for learning the regularized NE, while "Unreg." denotes the standard unregularized setting with $\beta = 0$. $\widetilde{\mathcal{O}}(\cdot)$ hides the logarithmic terms. We only report the dominant $\mathcal{O}(\sqrt{T})$ terms for prior works; the omitted lower-order terms typically exhibit worse dependence on $H$ and $d$.

**Noisy Bandit Feedback:** The matrix $A$ is unknown and can be accessed through noisy oracle bandit queries. For any $i \in [m]$ and $j \in [n]$, we can query the oracle and receive feedback $\widehat{A}(i,j)$ where

$$\widehat{A}(i,j) = A(i,j) + \xi.$$

Here, $\xi$ is i.i.d zero mean subgaussian random variable with parameter $\sigma > 0$. We are interested in learning the NE of the matrix game (1) in a sample-efficient manner using as few queries as possible.

**Goal: Regret minimization.** We define the dual-gap corresponding to the policy pair $(\mu, \nu)$ as

$$\begin{aligned}
\mathsf{DualGap}(\mu,\nu) &:= f^{\star,\nu}(A) - f^{\mu,\star}(A) \\
&= \underbrace{f^{\star,\nu}(A) - f^{\mu,\nu}(A)}_{\text{min player exploitability}(\nu)} + \underbrace{f^{\mu,\nu}(A) - f^{\mu,\star}(A)}_{\text{max player exploitability}(\mu)},
\end{aligned}$$

where

$$f^{\star,\nu}(A) := \max_{\mu \in \Delta^m} f^{\mu,\nu}(A), \quad f^{\mu,\star}(A) := \min_{\nu \in \Delta^n} f^{\mu,\nu}(A). \tag{4}$$

The dual gap can be viewed as the total *exploitability* (Davis et al., 2014) of the policy pair $(\mu, \nu)$ by the respective opponent. The dual gap of the NE policy pair $(\mu^\star, \nu^\star)$ is zero (see (3)). In order to capture the cumulative regret of both the players over T rounds, for a sequence of policy pairs $\{(\mu_t, \nu_t)\}_{t=1}^T$, the cumulative regret over $T$ rounds is given by the sum of dual gaps $\sum_{t=1}^T \mathsf{DualGap}(\mu_t, \nu_t) =$

$$\text{Regret}(T) = \sum_{t=1}^T \left(f^{\star,\nu_t}(A) - f^{\mu_t,\star}(A)\right).$$

## 2.2. Algorithm Development

We propose a model-based algorithm (Algorithm 1) called Optimistic Matrix Game (OMG) based on UCB-style bonuses (Auer et al., 2002). To enable function approximation, we parameterize the payoff matrix by $A_\omega$ with $\omega \in \mathbb{R}^d$ as the parameter vector. At each step $t \in [T]$, OMG estimates the payoff matrix based on collected samples and collects bandit feedback using the optimistic best response policy pairs. To elaborate further,

• *Payoff matrix update:* Given the set $\mathcal{D}_{t-1}$, the matrix $\overline{A}_t$ is computed as the model that minimizes the regularized least-squares loss between the model and the collected feedback (6). The policy pair $(\mu_t, \nu_t)$ is computed as the KL-regularized NE policies under the payoff matrix $\overline{A}_t$.

• *Data collection using optimistic best response pairs:* The optimistic model $A_t^+$ (resp. $A_t^-$) for the max (resp. min) players is computed by adding (resp. subtracting) the bonus matrix $b_t$ to the MSE matrix $\overline{A}_t$ (7). Each player's best response under its respective optimistic model is obtained by fixing the other's strategy (8), yielding policy pairs $(\tilde{\mu}_t, \nu_t)$ and $(\mu_t, \tilde{\nu}_t)$. We sample $(i_t^+, j_t^+) \sim (\tilde{\mu}_t, \nu_t)$, $(i_t^-, j_t^-) \sim (\mu_t, \tilde{\nu}_t)$ and collect noisy feedback $\widehat{A}(i_t^+, j_t^+)$ and $\widehat{A}(i_t^-, j_t^-)$. Update $\mathcal{D}_t = \mathcal{D}_t^+ \cup \mathcal{D}_t^-$ where $\mathcal{D}_t^+ = \mathcal{D}_{t-1}^+ \cup \left\{(i_t^+, j_t^+, \widehat{A}(i_t^+, j_t^+))\right\}$ and $\mathcal{D}_t^- = \mathcal{D}_{t-1}^- \cup \left\{(i_t^-, j_t^-, \widehat{A}(i_t^-, j_t^-))\right\}$.

## 2.3. Theoretical Guarantees

**Assumption 2.1** (Linear function approximation (Yang et al., 2025a))**.** The true payoff matrix belongs to the function class

$$A_\omega(i,j) := \langle \omega, \phi(i,j) \rangle, \quad \forall i \in [m], j \in [n],$$

where $\omega \in \mathbb{R}^d$ is the parameter vector, and $\phi(i,j) \in \mathbb{R}^d$ is the feature vector associated with the $(i,j)^{\text{th}}$ entry. The feature vectors are known and fixed, satisfying $\|\phi(i,j)\|_2 \leq 1 \ \forall \ i \in [m], j \in [n]$.

**Assumption 2.2** (Realizability)**.** There exists $\omega^\star \in \mathbb{R}^d$ such that $A = A_{\omega^\star}$ and $\|\omega^\star\|_2 \leq \sqrt{d}$.

**Bonus Function:** Under Assumption 2.1, given $\delta \in (0,1)$, the bonus matrix $b_t$ at time $t$ is defined as

$$b_t(i,j) = \eta_T \|\phi(i,j)\|_{\Sigma_t^{-1}}, \tag{5}$$

wherein $\Sigma_t = \lambda \mathbf{I} + \sum_{(i,j) \in \mathcal{D}_{t-1}} \phi(i,j)\phi(i,j)^\top$ and $\eta_T = \sigma\sqrt{d\log\left(\frac{3(1+2T/\lambda)}{\delta}\right)} + \sqrt{\lambda d}$.

**Regret Guarantees.** We now present the main results for the OMG algorithm. Full proofs are deferred to Appendix E.

**Theorem 2.3.** *Under Assumptions 2.1 and 2.2, for any fixed $\delta \in (0,1)$ and reference policies $(\mu_{ref}, \nu_{ref})$, choosing $\lambda = 1$ and $b_t(i,j)$ per eq. (5) in Algorithm 1, we have the following guarantees hold simultaneously w.p. $1 - \delta$*

• *Regularization-dependent guarantee: For any $\beta > 0$, we have $\forall\, T \in \mathbb{N}^+$*

$$\text{Regret}(T) \leq \mathcal{O}\left(\beta^{-1}d^2 \max\{1, \sigma^2\} \log^2\left(\frac{T}{\delta}\right)\right).$$

• *Regularization-independent guarantee: For any $\beta \geq 0$, we have $\forall\, T \in \mathbb{N}^+$*

$$\text{Regret}(T) \leq \mathcal{O}\left((1+\sigma)d\sqrt{T}\log\left(\frac{T}{\delta}\right)\right).$$

Under bounded noise $\sigma$, OMG achieves a regret bound of $\min\{\widetilde{\mathcal{O}}(d\sqrt{T}), \mathcal{O}(\beta^{-1}d^2 \log^2(T/\delta))\}$, which grows only logarithmically with $T$. This significantly improves upon the prior rate $\widetilde{\mathcal{O}}(d\sqrt{T})$ in Yang et al. (2025a) under KL-regularization. For smaller values of $T$ or the regularization parameter $\beta$ (even $\beta = 0$), OMG recovers the $\widetilde{\mathcal{O}}(d\sqrt{T})$ regret guarantee of the standard algorithms designed for the unregularized setting through the regularization-independent bound. Consequently, OMG can learn an $\varepsilon$-NE using $\min\{\widetilde{\mathcal{O}}(d^2/\varepsilon^2), \widetilde{\mathcal{O}}(\beta^{-1}d^2/\varepsilon)\}$ samples.

## 3. Two-Player Zero-Sum Markov Games

### 3.1. Problem Setup

We consider a two-player zero-sum KL-regularized Markov game with a finite horizon represented as $\mathcal{M} := \{\mathcal{S}, \mathcal{U}, \mathcal{V}, P, r, H\}$ where $\mathcal{S}$ is a possibly infinite state space, $\mathcal{U}, \mathcal{V}$ are the finite action spaces of the max and min players

where $P : \mathcal{S} \times \mathcal{U} \times \mathcal{V} \to \Delta(\mathcal{S})$ is the set of inhomogeneous transition kernels and $r = \{r_h\}_{h=1}^H$ with $r_h : \mathcal{S} \times \mathcal{U} \times \mathcal{V} \to [0,1]$ the reward function. Here, we will focus on the class of Markovian policies $\mu := \{\mu_h\}_{h=1}^H$ (resp. $\nu := \{\nu_h\}_{h=1}^H$) for the max (resp. min) player, where the action of each player at any step $h$ only depends on the current state ($\mu_h : \mathcal{S} \times [H] \to \Delta(\mathcal{U})$ and $\nu_h : \mathcal{S} \times [H] \to \Delta(\mathcal{V})$) with no dependence on the history. For reference policies $\mu_{\text{ref}} : \mathcal{S} \times [H] \to \Delta(\mathcal{U})$, $\nu_{\text{ref}} : \mathcal{S} \times [H] \to \Delta(\mathcal{V})$ $\forall (s,i,j) \in \mathcal{S} \times \mathcal{U} \times \mathcal{V}, h \in [H]$ the KL-regularized value and Q-function under this setup is given as (Cen et al., 2024)

$$V_h^{\mu,\nu}(s) := \mathbb{E}\Bigg[ \sum_{k=h}^H r_k(s_k, i, j) - \beta \log \frac{\mu_k(i|s_k)}{\mu_{\text{ref},k}(i|s_k)}$$
$$+ \beta \log \frac{\nu_k(j|s_k)}{\nu_{\text{ref},k}(j|s_k)} \Bigg| s_h = s\Bigg], \quad (9)$$
$$Q_h^{\mu,\nu}(s,i,j) := r_h(s,i,j) + \mathop{\mathbb{E}}_{s' \sim P_h(\cdot|s_h,i,j)}\left[V_{h+1}^{\mu,\nu}(s')\right]. \tag{10}$$

The value function can be expressed in terms of the Q-function as follows

$$V_h^{\mu,\nu}(s) = \mathop{\mathbb{E}}_{\substack{i \sim \mu_h(\cdot|s) \\ j \sim \nu_h(\cdot|s)}}\Bigg[ Q_h^{\mu,\nu}(s,i,j) - \beta \log \frac{\mu_h(i|s)}{\mu_{\text{ref},h}(i|s)}$$
$$+ \beta \log \frac{\nu_h(j|s)}{\nu_{\text{ref},h}(j|s)} \Bigg] \tag{11}$$
$$= \mathop{\mathbb{E}}_{\substack{i \sim \mu_h(\cdot|s) \\ j \sim \nu_h(\cdot|s)}}[Q_h^{\mu,\nu}(s,i,j)] - \beta \text{KL}\left(\mu_h(\cdot|s)\|\mu_{\text{ref},h}(\cdot|s)\right)$$
$$+ \beta \text{KL}\left(\nu_h(\cdot|s)\|\nu_{\text{ref},h}(\cdot|s)\right).$$

For fixed policy $\nu$ of the min player, the best response value function of the max player is defined as

$$\forall s \in \mathcal{S}, h \in [H]: \quad V_h^{\star,\nu}(s) = \max_\mu V_h^{\mu,\nu}(s). \tag{12}$$

The associated best response policy, denoted $\mu^\dagger(\nu)$, follows from solving (12), admits a closed-form expression given by $\forall i \in \mathcal{U}, s \in \mathcal{S}, h \in [H]$

$$\mu_h^\dagger(i|s) := \frac{\mu_{\text{ref},h}(i|s) \exp\left(\mathop{\mathbb{E}}_{j \sim \nu_h(\cdot|s)}[Q^\dagger(s,i,j)/\beta]\right)}{\sum_{k \in \mathcal{U}} \mu_{\text{ref},h}(k|s) \exp\left(\mathop{\mathbb{E}}_{j \sim \nu_h(\cdot|s)}[Q^\dagger(s,k,j)/\beta]\right)}, \tag{13}$$

where $Q^\dagger = Q^{\mu^\dagger,\nu}$. Similarly we define $\nu^\dagger(\mu)$, the best response of the min player to a fixed strategy $\mu$ of the max

---

**Algorithm 1** Optimistic Matrix Game (OMG)

---

1: **Input:** Reg. parameter $\beta$, regularization, no. iterations $T$, ref. policies $(\mu_{\text{ref}}, \nu_{\text{ref}})$.
2: **Initialization:** Dataset $\mathcal{D}_0 := \emptyset$, $\lambda > 0$, initial parameter $\omega_0$
3: **for** $t = 1, \cdots, T$ **do**
4:     Compute the LMSE matrix $\overline{A}_t := A_{\overline{\omega}_t}$ where

$$\overline{\omega}_t = \arg\min_{\omega \in \mathbb{R}^d} \sum_{(i,j,\widehat{A}(i,j)) \in \mathcal{D}_{t-1}} \left( A_\omega(i,j) - \widehat{A}(i,j) \right)^2 + \lambda\|\omega\|_2^2. \tag{6}$$

5:     Compute optimistic matrix games for both players using $b_t$ in (5):

$$A_t^+ := \overline{A}_t + b_t \quad A_t^- := \overline{A}_t - b_t. \tag{7}$$

6:     Compute the NE $(\mu_t, \nu_t)$ of the matrix game $\overline{A}_t$, and the best response pairs under optimism

$$\tilde{\mu}_t = \arg\max_{\mu \in \Delta^m} f^{\mu,\nu_t}(A_t^+), \quad \tilde{\nu}_t = \arg\min_{\nu \in \Delta^n} f^{\mu_t,\nu}(A_t^-). \tag{8}$$

7:     Sample $(i_t^+, j_t^+) \sim (\tilde{\mu}_t, \nu_t)$, $(i_t^-, j_t^-) \sim (\mu_t, \tilde{\nu}_t)$, collect feedback, and update $\mathcal{D}_t$.
8: **end for**

---

player $\forall j \in \mathcal{V}, s \in \mathcal{S}, h \in [H]$

$$\nu_h^\dagger(j|s) = \frac{\nu_{\text{ref},h}(j|s) \exp\left( -\mathbb{E}_{i \sim \mu_h(\cdot|s)}[Q^\dagger(s,i,j)/\beta] \right)}{\sum_{k \in \mathcal{V}} \nu_{\text{ref},h}(k|s) \exp\left( -\mathbb{E}_{i \sim \mu_h(\cdot|s)}[Q^\dagger(s,i,k)/\beta] \right)}, \tag{14}$$

where $Q^\dagger = Q^{\mu,\nu^\dagger}$. A policy pair $(\mu^\star, \nu^\star)$ is called the Nash equilibrium of the Markov game if both the policies $\mu^\star$ and $\nu^\star$ are best responses to each other. The dual gap associated with a policy pair $(\mu, \nu)$ is given by

$$\mathsf{DualGap}(\mu, \nu) := V_1^{\star,\nu}(\rho) - V_1^{\mu,\star}(\rho).$$

Here $V_1^{\mu,\nu}(\rho) = \mathbb{E}_{s_1 \sim \rho}[V_1^{\mu,\nu}(s_1)]$ where $\rho$ is the initial state distribution. The cumulative regret associated with sequence of policies $\{(\mu_t, \nu_t)\}_{t=1}^T$ is given by the sum of dual gaps $\sum_{t=1}^T \mathsf{DualGap}(\mu_t, \nu_t)$

$$\text{Regret}(T) = \sum_{t=1}^T V_1^{\star,\nu_t}(\rho) - V_1^{\mu_t,\star}(\rho).$$

### 3.2. Algorithm Development

We propose a model-free algorithm (Algorithm 2) called SOMG which uses bonuses based on superoptimistic confidence intervals, larger than the ones used in standard UCB style analysis (Auer et al., 2002) to ensure efficient exploration-exploitation tradeoff and achieve logarithmic regret. To enable function approximation, we use the function class $f_h^\theta : \mathcal{S} \times \mathcal{U} \times \mathcal{V} \to \mathbb{R}$ parameterized by $\theta \in \Theta$ for the

regression step (15). The $Q$ functions are obtained subsequently using a projection operation (16). The algorithm, on a high level maintains three $Q$ and $V$ functions, estimates superoptimistic best response for each player by solving stagewise matrix games and performs data collection using the best response policy pairs. Here we further elaborate the algorithm:

• *Q function updates:* SOMG maintains three value $(\overline{V}_h, V_h^+ \text{ and } V_h^-)$ and $Q$ functions $(\overline{Q}_h, Q_h^+ \text{ and } Q_h^-)$. The Q functions are updated in two steps. 1) Solving the regularized least mean squared error with respective bellman targets $(r_h + V_{h+1})$ using data collected until $t-1$ $(\mathcal{D}_{t-1})$. (15) followed by a 2) projection step (16) wherein the $Q$ functions are projected onto respective feasible regions. The projection operator is defined as follows

$$\Pi_h(x) = \max\{0, \min\{x, H-h+1\}\} \tag{20a}$$

$$\Pi_h^+(x) = \max\left\{0, \min\{x, 3(H-h+1)^2\}\right\} \tag{20b}$$

$$\Pi_h^-(x) = \min\left\{ -3(H-h+1)^2, \right.$$
$$\left. \max\{x, H-h+1\} \right\}. \tag{20c}$$

The projection operator is designed to enable superoptimism by choosing a ceiling higher than the maximum attainable value. Standard optimistic algorithms use the same projection operator for the optimistic estimates of both the players $\Pi_h^{\text{opt}}(x) = \max\{0, \min\{x, (H-h+1)\}\}$.

• *Superoptimism:*[2] To calculate the superoptimistic $Q$ function for the max (resp. min) player we add (resp. subtract)

---

[2]A similar concept called *over-optimism* where extra padding is added to the bonus was used in single-agent RL (Agarwal et al.,

---

**Algorithm 2** Super-Optimistic Markov Game (SOMG)

---

1: **Input:** Reg. parameter $\beta$ no. iteration $T$, ref. policies $(\mu_{\text{ref}}, \nu_{\text{ref}})$.
2: **Initialization:** Dataset $\mathcal{D}_0 := \emptyset$, $\lambda \geq 0$, initial parameters $\{\overline{\theta}_{h,0}, \theta^+_{h,0}, \theta^-_{h,0}\}^H_{h=1}$.
3: **for** $t = 1, \cdots, T$ **do**
4:      **for** $h = H, H-1 \cdots, 1$ **do**
5:          Regress onto MSE Bellman target, optimistic Bellman targets for each player

$$\overline{\theta}_{h,t} \leftarrow \arg\min_{\theta \in \Theta} \sum_{k=1}^{|\mathcal{D}_{t-1}|} \left( f_h^\theta(s_{h,k}, i_{h,k}, j_{h,k}) - r_{h,k} - \overline{V}_{h+1,t}(s_{h+1,k}) \right)^2 + \lambda \|\theta\|_2^2, \tag{15a}$$

$$\theta^+_{h,t} \leftarrow \arg\min_{\theta \in \Theta} \sum_{k=1}^{|\mathcal{D}_{t-1}|} \left( f_h^\theta(s_{h,k}, i_{h,k}, j_{h,k}) - r_{h,k} - V^+_{h+1,t}(s_{h+1,k}) \right)^2 + \lambda \|\theta\|_2^2, \tag{15b}$$

$$\theta^-_{h,t} \leftarrow \arg\min_{\theta \in \Theta} \sum_{k=1}^{|\mathcal{D}_{t-1}|} \left( f_h^\theta(s_{h,k}, i_{h,k}, j_{h,k}) - r_{h,k} - V^-_{h+1,t}(s_{h+1,k}) \right)^2 + \lambda \|\theta\|_2^2. \tag{15c}$$

6:          Compute MSE, superoptimistic Q functions for both players

$$\overline{Q}_{h,t}(s, i, j) := \Pi_h \left\{ f_h^{\overline{\theta}_{h,t}}(s, i, j) \right\}, \tag{16a}$$

$$Q^+_{h,t}(s, i, j) := \Pi_h^+ \left\{ f_h^{\theta^+_{h,t}}(s, i, j) + b^{\sup}_{h,t}(s, i, j) \right\}, \tag{16b}$$

$$Q^-_{h,t}(s, i, j) := \Pi_h^- \left\{ f_h^{\theta^-_{h,t}}(s, i, j) - b^{\sup}_{h,t}(s, i, j) \right\}. \tag{16c}$$

7:          Compute Nash equilibrium w.r.t. LMSE game, and the

$$(\mu_{h,t}(\cdot|s), \nu_{h,t}(\cdot|s)) \leftarrow \text{Nash Zero-sum}_\beta((\overline{Q}_{h,t})(s, \cdot, \cdot)). \tag{17}$$

8:          Compute Optimistic Best Responses for both players

$$\tilde{\mu}_{h,t}(\cdot|s) \leftarrow \text{Best Response}_\beta(Q^+_{h,t}(s, \cdot, \cdot), \nu_{h,t}(\cdot|s)), \tag{18a}$$

$$\tilde{\nu}_{h,t}(\cdot|s) \leftarrow \text{Best Response}_\beta(Q^-_{h,t}(s, \cdot, \cdot), \mu_{h,t}(\cdot|s)). \tag{18b}$$

9:          Compute the value functions

$$\overline{V}_{h,t}(s) \leftarrow \mathbb{E}_{\substack{i \sim \mu_{h,t}(\cdot|s) \\ j \sim \nu_{h,t}(\cdot|s)}} \left[ \overline{Q}_{h,t}(s, i, j) \right] - \beta \text{KL}(\mu_{h,t} \| \mu_{\text{ref},h})(s) + \beta \text{KL}(\nu_{h,t} \| \nu_{\text{ref},h})(s) \tag{19a}$$

$$V^+_{h,t}(s) \leftarrow \mathbb{E}_{\substack{i \sim \tilde{\mu}_{h,t}(\cdot|s) \\ j \sim \nu_{h,t}(\cdot|s)}} \left[ Q^+_{h,t}(s, i, j) \right] - \beta \text{KL}(\tilde{\mu}_{h,t} \| \mu_{\text{ref},h})(s) + \beta \text{KL}(\nu_{h,t} \| \nu_{\text{ref},h})(s) \tag{19b}$$

$$V^-_{h,t}(s) \leftarrow \mathbb{E}_{\substack{i \sim \mu_{h,t}(\cdot|s) \\ j \sim \tilde{\nu}_{h,t}(\cdot|s)}} \left[ Q^-_{h,t}(s, i, j) \right] - \beta \text{KL}(\mu_{h,t} \| \mu_{\text{ref},h})(s) + \beta \text{KL}(\tilde{\nu}_{h,t} \| \nu_{\text{ref},h})(s) \tag{19c}$$

10:      **end for**
11:      Receive $s_{1,t} \sim \rho$, sample $\tau^+_t \sim (\tilde{\mu}_t, \nu_t)$ and $\tau^-_t \sim (\mu_t, \tilde{\nu}_t)$, and update $\mathcal{D}_t$.
12: **end for**

---

the super optimistic bonus ($b^{\sup}_{h,t}$). Standard optimism only

adds an *optimistic* bonus $b_{h,t}$ (21) which is a high probability upper bound on the Bellman error of the superoptimistic

2023) for a different purpose of maintaining monotonicity of variance estimates.

Q function $\left| f_h^{\theta^+_{h,t}}(s, i, j) - r_h(s, i, j) + PV^+_{h+1}(s, i, j) \right| \leq$

$b_{h,t}(s, i, j)$ (called optimistic $Q$ function under vanilla optimism) and computes the Q function as:

$$Q_{h,t}^+(s, i, j) = \Pi \left( f_h^{\theta_{h,t}^+}(s, i, j) + b_{h,t}(s, i, j) \right) \quad (21)$$

However SOMG uses a superoptimistic bonus defined as:

$$b_{h,t}^{\sup}(s, i, j) = b_{h,t}(s, i, j) + 2b_{h,t}^{\mathrm{mse}}(s, i, j), \quad (22)$$

where the additional bonus $b_{h,t}^{\mathrm{mse}}(s, i, j)$ is a high probability upper bound on the Bellman error in the MSE $Q$ function.

$$\left| \overline{Q}_h(s, i, j) - r_h(s, i, j) + P\overline{V}_{h+1}(s, i, j) \right| \leq b_{h,t}^{\mathrm{mse}}(s, i, j),$$

which results in the super optimistic $Q$ function being strictly greater than the high confidence upper bound (21) one obtains from optimism.

• *Best response computation:* The stagewise Nash Equilibrium policy pair $(\mu_{h,t}(\cdot|s), \nu_{h,t}(\cdot|s))$ is computed by solving the KL regularized zero-sum matrix (2) game with the payoff matrix being $A = \overline{Q}_{h,t}(s, \cdot, \cdot)$ and reference policies $\mu_{\mathrm{ref},h}(\cdot|s)$ and $\nu_{\mathrm{ref},h}(\cdot|s)$ (17). The policies $\tilde{\mu}_{h,t}(\cdot|s)$ and $\tilde{\nu}_{h,t}(\cdot|s)$ are computed as the best responses to policies $\nu_{h,t}(\cdot|s)$ and $\mu_{h,t}(\cdot|s)$ under matrix games with payoff matrices $Q_{h,t}^+(s, i, j)$ and $Q_{h,t}^-(s, i, j)$ respectively.

• *Value function update and Data collection:* The value functions $\overline{V}_{h,t}(s)$, $V_{h,t}^+(s)$ and $V_{h,t}^-(s)$ are updated via the Bellman equation (11) using policy pairs $(\mu_{h,t}, \nu_{h,t})$, $(\tilde{\mu}_{h,t}, \nu_{h,t})$, and $(\mu_{h,t}, \tilde{\nu}_{h,t})$, respectively (19). We use $\mathrm{KL}(a|b)(s)$ as shorthand for $\mathrm{KL}(a(\cdot|s)|b(\cdot|s))$. Two new trajectories

$$\tau_t^+ = \left\{ (s_{h,t}^+, i_{h,t}^+, j_{h,t}^+, r_{h,t}^+, s_{h+1,t}^+) \right\}_{h=1}^H$$

$$\tau_t^- = \left\{ (s_{h,t}^-, i_{h,t}^-, j_{h,t}^-, r_{h,t}^-, s_{h+1,t}^-) \right\}_{h=1}^H$$

are collected by following policies $(\tilde{\mu}_t, \nu_t) = \{(\tilde{\mu}_{h,t}, \nu_{h,t})\}_{h=1}^H$ and $(\mu_t, \tilde{\nu}_t) = \{(\mu_{h,t}, \tilde{\nu}_{h,t})\}_{h=1}^H$ respectively. Update the dataset $\mathcal{D}_t^+ = \mathcal{D}_{t-1}^+ \cup \{\tau_t^+\}$ and $\mathcal{D}_t^- = \mathcal{D}_{t-1}^- \cup \{\tau_t^-\}$, $\mathcal{D}_t = \mathcal{D}_t^+ \cup \mathcal{D}_t^-$.

**Computational benefit of Regularization:** The Nash equilibrium computation steps in line 6 of Algorithm 1, as well as equations (17) of Algorithm 2, require solving for the NE of a KL-regularized zero-sum matrix game. This can be accomplished using policy extragradient/Mirror descent based methods (Cen et al., 2023; 2024; Sokota et al., 2023), which guarantee last-iterate linear convergence. In contrast, solving the corresponding problem in the unregularized setting only yields an $\mathcal{O}(1/T)$ convergence rate.

### 3.3. Theoretical Guarantees

**Assumption 3.1** (Linear MDP (Jin et al., 2020; Xie et al., 2023)). The MDP $\mathcal{M} := \{\mathcal{S}, \mathcal{U}, \mathcal{V}, r, P, H\}$ is a linear MDP with features $\phi : \mathcal{S} \times \mathcal{U} \times \mathcal{V} \to \mathbb{R}^d$ and for every $h \in [H]$ there exists an unknown signed measure $\psi_h(\cdot) \in \mathbb{R}^d$ over $\mathcal{S}$ and an unknown fixed vector $\omega_h \in \mathbb{R}^d$ such that

$$P_h(\cdot \mid s, i, j) = \langle \phi(s, i, j), \psi_h(\cdot) \rangle,$$
$$r_h(s, i, j) = \langle \phi(s, i, j), \omega_h \rangle.$$

Without loss of generality, we assume $\|\phi(s, i, j)\| \leq 1$ for all $(s, i, j) \in \mathcal{S} \times \mathcal{U} \times \mathcal{V}$, and $\max\{\|\psi_h(\mathcal{S})\|, \|\omega_h\|\} \leq \sqrt{d}$ for all $h \in [H]$.

We use linear function approximation with $f_h^\theta(s, i, j) := \langle \theta, \phi(s, i, j) \rangle$ and $\Theta = \mathbb{R}^d$.

**Proposition 3.2.** *Under Assumption 3.1, for the Nash equilibrium policy $(\mu^\star, \nu^\star) = (\mu_h^\star, \nu_h^\star)_{h=1}^H$ there exist weights $\{\theta_h^{\mu^\star, \nu^\star}\}_{h=1}^H$ such that $\forall (s, i, j) \in \mathcal{S} \times \mathcal{U} \times \mathcal{V}, h \in [H]$, we have*

$$Q_h^{\mu^\star, \nu^\star}(s, i, j) = \left\langle \phi(s, i, j), \theta_h^{\mu^\star, \nu^\star} \right\rangle.$$
$$\left| Q_h^{\mu^\star, \nu^\star}(s, i, j) \right| \in [0, H - h + 1]$$

Note that $\mathcal{D}_{t-1}$ contains $2(t-1)$ trajectories; for convenience we index them by $\tau$, with each trajectory of the form $\left\{ (s_h^\tau, i_h^\tau, j_h^\tau, r_h^\tau, s_{h+1}^\tau) \right\}_{h=1}^H$. We define $\Sigma_{h,t}$ as follows:

$$\Sigma_{h,t} := \lambda \mathbf{I} + \sum_{\tau \in \mathcal{D}_{t-1}} \phi(s_h^\tau, i_h^\tau, j_h^\tau) \phi(s_h^\tau, i_h^\tau, j_h^\tau)^\top.$$

The expressions for $\overline{\theta}_{h,t}, \theta_{h,t}^+$ and $\theta_{h,t}^-$ are given by

$$\overline{\theta}_{h,t} = \Sigma_{h,t}^{-1} \sum_{\tau \in \mathcal{D}_{t-1}} \phi_{h,\tau} \left[ r_{h,\tau} + \overline{V}_{h+1,t}(s_{h+1}^\tau) \right],$$

$$\theta_{h,t}^+ = \Sigma_{h,t}^{-1} \sum_{\tau \in \mathcal{D}_{t-1}} \phi_{h,\tau} \left[ r_{h,\tau} + V_{h+1,t}^+(s_{h+1}^\tau) \right],$$

$$\theta_{h,t}^- = \Sigma_{h,t}^{-1} \sum_{\tau \in \mathcal{D}_{t-1}} \phi_{h,\tau} \left[ r_{h,\tau} + V_{h+1,t}^-(s_{h+1}^\tau) \right].$$

where $\phi_{h,\tau}$ is the feature map corresponding to the state $s_h^\tau$.

**Bonus function:** Under Assumption 3.1, the superoptimistic bonus function $b_{h,t}^{\sup}$ is defined as in eq. (22) with

$$b_{h,t}^{\mathrm{mse}}(s, i, j) = \eta_1 \|\phi(s, i, j)\|_{\Sigma_{h,t}^{-1}} \quad (23a)$$

$$b_{h,t}(s, i, j) = \eta_2 \|\phi(s, i, j)\|_{\Sigma_{h,t}^{-1}}, \quad (23b)$$

where $\eta_1 = c_1 \sqrt{d} H \sqrt{\log\left(\frac{16T}{\delta}\right)}$ and $\eta_2 = c_2 d H^2 \sqrt{\log\left(\frac{16dTH}{\delta \min\{1, \beta\}}\right)}$ for some determinable universal constants $c_1, c_2 > 0$.

**Regret Guarantees:** We now present the main results for the SOMG algorithm. The complete results, presented as two separate theorems, are deferred to Appendix F.

**Theorem 3.3.** *Under Assumption 3.1, for any reference policies* $(\mu_{ref}, \nu_{ref}) = (\{\mu_{ref,h}(\cdot|\cdot)\}_{h=1}^{H}, \{\nu_{ref,h}(\cdot|\cdot)\}_{h=1}^{H})$, *any fixed* $\delta \in [0, 1]$, *choosing* $\lambda = 1$ *and* $b_{h,t}^{sup}(s, i, j)$ *as per eq.* (23) *in algorithm 2, we have the following guarantees hold simultaneously w.p.* $(1 - \delta)$

- *Linear in* $\beta^{-1}$ *logarithmic guarantee: For any* $\beta > 0$, $\forall\, T \in \mathbb{N}^{+}$, *we have*

$$\text{Regret}(T) \leq \mathcal{O}\left(\beta^{-1} d^3 H^7 \log^2\left(\frac{dTH}{\delta \min\{1, \beta\}}\right)\right).$$

- *Traditional* $\sqrt{T}$ *guarantee: For any* $\beta > 0$, $\forall\, T \in \mathbb{N}^{+}$ *we have*

$$\text{Regret}(T) \leq \mathcal{O}\left(d^{3/2} H^3 \sqrt{T} \log\left(\frac{dTH}{\delta \min\{1, \beta\}}\right)\right).$$

As demonstrated in Theorem 3.3, SOMG, achieves a regret bound of $\min\{\widetilde{\mathcal{O}}(d^{3/2}H^3\sqrt{T}), \widetilde{\mathcal{O}}(\beta^{-1}d^3H^7\log^2(T/\delta))\}$, [3] which grows only logarithmically with $T$. Consequently, SOMG needs only $\min\{\widetilde{\mathcal{O}}(d^3H^6/\varepsilon^2), \widetilde{\mathcal{O}}(\beta^{-1}d^3H^7/\varepsilon)\}$ samples to learn an $\varepsilon$-NE.

**Reduction to the single agent case:** Both OMG and SOMG naturally reduce to multi-armed Bandit and single-agent RL respectively when the min-player's action space is a singleton. As elaborated in Appendix, for single agent setting SOMG can additionally obtain improved regret guarantees of $\widetilde{\mathcal{O}}\left(\beta^{-1}d^3H^5\log^2\left(\frac{dT}{\delta}\right)\right)$ and $\widetilde{\mathcal{O}}\left(d^{3/2}H^2\sqrt{T}\log\left(\frac{dT}{\delta}\right)\right)$ respectively.

**Technical Challenges.**

**Absence of closed form expressions for NE policies:** In single-agent settings (bandits and RL), analyses of algorithms achieving logarithmic regret rely on the fact that the optimal policy for a given transition–reward model pair directly admits a Gibbs-style closed-form solution (Zhao et al., 2025b;a; Tiapkin et al., 2024). In contrast, in game-theoretic settings, no such direct closed-form expression exists for Nash equilibrium policies. The same absence of closed form expressions also arises in Coarse Correlated Equilibrium (CCE)–based approaches, which are commonly employed to achieve $\mathcal{O}(\sqrt{T})$ regret when learning Nash equilibrium for zero-sum games (Xie et al., 2023; Jin et al., 2022; Chen et al., 2022; Liu et al., 2021). We address this challenge by leveraging best response sampling, where the best response to a fixed opponent policy does admit a closed-form expression.

**Unbounded Value functions:** Moreover in the single-agent RL setting with KL regularization, the value function does

---

[3] By employing Bernstein-based (Xie et al., 2021) bonuses in SOMG, one could potentially shave off an additional $Hd^{1/2}$ factor in the $\sqrt{T}$ bound and an $H^2 d$ factor in the logarithmic regret bound.

not include any positive KL regularization terms. Thus, both the value and $Q$-functions are upper bounded by $H$. As a consequence, the optimistic $Q$-function is bounded within $[0, H]$. This boundedness enables the direct construction of confidence intervals for the optimistic $Q$-function using standard concentration results, which in turn allows algorithms from the unregularized setting to be carried over to the regularized setting with minimal modifications. However, in the KL-regularized game (9)(10), the value functions contain positive KL terms, which can cause them to take arbitrarily large values exceeding $H$. This makes it challenging to construct confidence intervals for the optimistic (superoptimistic in our case) $Q$-functions directly. We solve this problem using best response sampling and superoptimism. (More details in appendix section B.2)

**Covering number bounds:** Lastly bounding the covering number of the induced superoptimistic value function class is non-trivial due to its complicated structure involving the best response to the NE policy of a different game. We exploit the properties of smoothness in variation of the NE policies in KL regularized setting to bound the covering number which results in the $\log(1/\min\{1, \beta\})$ term in our regret bounds which might be of independent interest. Without KL regularization, one will have to resort to discretizing the $Q$ function class (Xie et al., 2023) to control the difference between KL terms.

## 4. Conclusion

In this work, we develop algorithms that achieve provably superior sample efficiency in competitive games under KL regularization. For matrix games, we introduced OMG, based on optimistic best-response sampling, and for Markov games, we developed SOMG, which relies on super-optimistic best-response sampling. Both methods attain regret that scales only logarithmically with the number of episodes $T$. Our analysis leverages the fact that in two-player zero-sum games, best responses to fixed opponent strategies admit closed-form solutions. To our knowledge, this is the first work to characterize the statistical efficiency gains under KL regularization in game-theoretic settings.

Several avenues for future work remain open, including deriving instance/gap-dependent regret guarantees under KL regularization that also capture the dependence on reference policies and developing offline counterparts of optimistic best-response sampling that achieve superior sample efficiency with KL regularization under reasonable coverage assumptions. Extending our methods to general multi-agent settings, where the objective is to compute CCE and best responses or optimal policies do not admit a closed-form expression is another promising direction.

## Impact Statement

This paper presents algorithms and theoretical guarantees aimed at improving the understanding of KL regularized competitive games. Given the theoretical nature of the work the potential societal and ethical consequences of our work are minimal, none which we feel must be specifically highlighted here.

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

# Appendix

## Contents

## LLM usage

We used LLMs minimally, focusing on making sentences more concise to fit the page limit.

## A. Related works

In this section we will discuss theoretical works that are related to ours

**Two Player Matrix Games:** Two-player zero-sum matrix games have been studied extensively, from the foundational work of (Shapley, 1953) to more recent analyses of convergence in the unregularized setting (Mertikopoulos et al., 2018; Daskalakis & Panageas, 2018; Wei et al., 2021). In settings with KL regularization, faster last-iterate linear convergence guarantees have also been established (Cen et al., 2023; 2024). However, these works focus on the tabular full-information setting. Closer to our setting are (O'Donoghue et al., 2021; Yang et al., 2025a), where the payoff matrix is unknown and must be estimated through noisy oracle queries. (O'Donoghue et al., 2021) introduced UCB/optimism (Lai, 1987) and K-Learning (similar to Thompson sampling (Russo et al., 2018)) based approaches in the tabular unregularized setting, while (Yang et al., 2025a) proposed a value-incentivization based approach (Liu et al., 2023) and established regret guarantees in the regularized setting with function approximation. Learning from preference feedback has also been studied in (Ye et al., 2024). However, none of these approaches exploit the structure of the KL-regularized problem to achieve logarithmic regret; instead, they maintain $\mathcal{O}(\sqrt{T})$ regret.

**Two Player Markov Games:** Two-player zero-sum Markov games (Littman, 1994) generalize single-agent MDPs to competitive two-player settings. The problem has widely studied in the finite horizon tabular setting (Bai & Jin, 2020; Bai et al., 2020; Liu et al., 2021), under linear function approximation (Xie et al., 2023; Chen et al., 2022), in the context of general function approximation (Jin et al., 2022; Huang et al., 2022) and under the infinite horizon setting (Sidford et al., 2020; Sayin et al., 2021). Many of these algorithms use optimism-based methods, using upper and lower bounds on the value functions to define a general-sum game. They sidestep the need to solve for a Nash equilibrium in general-sum games by employing CCE-based sampling, exploiting the fact that in two-player settings the dual gap of a joint policy over the joint action space matches that of the corresponding marginal independent policies. In addition there have also been works solving the problem under full information setting with exact/first order oracle access (Zeng et al., 2022; Cen et al., 2023; 2024; Yang & Ma, 2023) and offline setting (Cui & Du, 2022; Zhong et al., 2022; Yan et al., 2024). All prior works consider

the unregularized setting, except (Zeng et al., 2022; Cen et al., 2024), which achieves linear convergence under entropy regularization, compared to the $\mathcal{O}(T^{-1})$ rate in the unregularized case.

**Entropy/KL Regularization in Decision Making:** Entropy regularization methods are widely used as a mechanism for encouraging exploration (Neu et al., 2017; Geist et al., 2019). These methods have been studied from a policy optimization perspective with some form of gradient oracle/first-order oracle access in single agent RL (Cen et al., 2022b; Lan, 2023), zero-sum matrix and markov games (Cen et al., 2023; 2024), zero-sum polymatrix games (Leonardos et al., 2021) and potential games (Cen et al., 2022a). Under bandit/preference feedback, value-biased bandit-based methods have been proposed that, like DPO (Rafailov et al., 2023), exploit the closed-form optimal policy to bypass the two-step RLHF procedure, for both offline (Cen et al., 2025) and online settings (Cen et al., 2025; Xie et al., 2025; Zhang et al., 2025a). These results were further extended to game-theoretic settings (Wang et al., 2023; Ye et al., 2024). (Yang et al., 2025a) develop value-biased algorithms for learning Nash Equilibrium in zero-sum matrix games and Coarse Correlated Equilibrium (CCE) in general-sum Markov games. However, none of these approaches leverage the structure of KL regularization and maintain a $\mathcal{O}(\sqrt{T})$ regret. More recently (Zhao et al., 2025a) achieved $\mathcal{O}(1/\varepsilon)$ sample complexity in the KL-regularized contextual bandits setting with a strong coverage assumption on the reference policy. Subsequently, (Zhao et al., 2025b; Tiapkin et al., 2024) proposed optimistic bonus–based algorithms for KL-regularized bandits and RL that achieve logarithmic regret ($\mathcal{O}(\beta^{-1}d^2\log^2(T))$ in bandits and $\mathcal{O}(\beta^{-1}H^5d^3\log^2(T))$ in RL)[4] without coverage assumptions, leveraging the closed-form *optimal policy* in their analysis. However, their results are limited to the single-player setting, where the *optimal policy* admits a closed-form expression in terms of the reward model. Similar faster convergence guarantees were also achieved for the RL setting by (Foster et al., 2025) and for offline contextual bandits with $f$-divergences (Zhao et al., 2025c).

**Game Theoretic Methods in LLM Alignment:** Fine-tuning large language models with reinforcement learning is a core part of modern post-training pipelines, enhancing reasoning and problem-solving (Guo et al., 2025). Game-theoretic and self-play methods extend reinforcement learning to multi-agent settings, with applications in alignment (Calandriello et al., 2024; Rosset et al., 2024; Munos et al., 2024; Zhang et al., 2025c) and reasoning (Cheng et al., 2024; Liu et al., 2025). Within this paradigm, self-play optimization is framed as an online two player matrix/markov game, where models iteratively improve using their own responses by solving for the Nash Equilibrium (Wu et al., 2025a; Chen et al., 2024; Swamy et al., 2024; Tang et al., 2025; Wang et al., 2025). More broadly, game theory has been applied to modeling non-transitive preferences (Swamy et al., 2024; Ye et al., 2024; Tiapkin et al., 2025), enabling collaborative post-training and decision-making (Park et al., 2025a;b), accelerating Best-of-N distillation (Yang et al., 2025b), and for multi-turn alignment/RLHF (Wu et al., 2025b; Shani et al., 2024) among other LLM applications.

## B. Proof Overview and Mechanisms

### B.1. Matrix Games

The cumulative regret can be decomposed as the cumulative sum of *exploitability* of the min and the max player

$$\text{Regret(T)} = \sum_{t=1}^{T}\left(f^{\star,\nu_t}(A) - f^{\mu_t,\star}(A)\right)$$

$$= \underbrace{\sum_{t=1}^{T}(f^{\star,\nu_t}(A) - f^{\mu_t,\nu_t}(A))}_{\text{Exploitability of the max player}} + \underbrace{\sum_{t=1}^{T}(f^{\mu_t,\nu_t}(A) - f^{\mu_t,\star}(A))}_{\text{Exploitability of the min player}}. \tag{24}$$

We bound the first term (exploitability of the max player) and the bounding of the second term follows analogous arguments. Now we have the following concentration inequality for Matrix games The first term in eq. (24) can be further decomposed as

$$\underbrace{\sum_{t=1}^{T}(f^{\star,\nu_t}(A) - f^{\mu_t,\nu_t}(A))}_{\text{Exploitability of the max player}} = \underbrace{\sum_{t=1}^{T}(f^{\star,\nu_t}(A) - f^{\tilde{\mu}_t,\nu_t}(A))}_{T_1} + \underbrace{\sum_{t=1}^{T}(f^{\tilde{\mu}_t,\nu_t}(A) - f^{\mu_t,\nu_t}(A))}_{T_2}$$

---

[4]For uniformity, we report the sample complexities under linear function approximation/linear MDP and per-step rewards $r_h \in [0,1]$ and trajectory reward $\sum_{h=1}^{H} r_h \in [0, H]$.

We will now analyze these terms individually.

**Bandits view for bounding $T_1$:** By construction of the algorithm, the strategies $\mu_t$, $\tilde{\mu}_t$, and $\dot{\mu}_t$ are best responses to the common fixed strategy $\nu_t$ of the min-player under the payoff matrices $\overline{A}_t$, $A_t^+$, and $A$ respectively. This property not only provides closed-form representations but also facilitates cancellation of the KL terms corresponding to $\nu_t$ in $T_1$ and $T_2$. As a result of fixed $\nu_t$, one can view the min-player strategy $\nu_t$ as part of the environment and bound $T_1$ the same way as done in bandits with the max player as the decision making entity.

REGULARIZATION-DEPENDENT BOUND

Traditional regret analysis in matrix games ignores the regularization terms and bounds the regret using the sum of bonuses $c \sum_{t=1}^T \mathbb{E}[b_t(i,j)]$ which is further bounded as $\sqrt{T} \log(T)$ using Jensen's inequality and the elliptical potential lemma/eluder dimension (Lemma D.6). However in the presence of regularization the originally payoff landscape, linear in $\mu$ and $\nu$ (1) becomes $\beta$ strongly convex in the policy $\nu$ and $\beta$ strongly concave in $\mu$. Under the full information setting it is well known that this facilitates design of algorithms that achieve faster convergence to the equilibrium (Cen et al., 2023; 2024). This intuitively suggests one can also design algorithms which achieve sharper regret guarantees in the regularized setting under bandit feedback. Specifically we show that we can bound the regret by the sum of squared bonuses $c\beta^{-1} \sum_{t=1}^T \mathbb{E}[b_t(i,j)^2]$ which enables using to circumvent the need for Jensen's inequality which contributes the $\sqrt{T}$ term and directly bound the terms using the elliptical potential lemma (Lemma D.6) to obtain a $\mathcal{O}(\beta^{-1} \log^2(T))$ regret. We detail the analysis as follows

Leveraging the bandits view, one can bound the term $T_1$ adapting the arguments from (Zhao et al., 2025b) (Theorem 4.1) as detailed in section E.1 to obtain $T_1 \le c\beta^{-1} \mathbb{E}_{i \sim \tilde{\mu}_t} \left[ (\mathbb{E}_{j \sim \nu_t}[(b_t(i,j))])^2 \right]$. In order to bound the term $T_2$ we use a mean value theorem based argument (detailed in section E.1 Step 2) and the property

$$2(|A_t^+(i,:) - \overline{A}_t(i,:)|\nu_t) \ge (|A_t^+(i,:) - A(i,:)|\nu_t), \tag{25}$$

to show that $T_2 \le c'\beta^{-1} \mathbb{E}_{i \sim \tilde{\mu}_t} \left[ (\mathbb{E}_{j \sim \nu_t}[(b_t(i,j))])^2 \right]$. The property in eq. (25) is a direct consequence of optimistic bonus function used in algorithm 1, however, we will need a superoptimistic bonus to obtain a similar property in Markov Games. Thus we have

$$T_1 + T_2 \le c''\beta^{-1} \sum_{t=1}^T \mathbb{E}_{i \sim \tilde{\mu}_t} \left[ \left( \mathbb{E}_{j \sim \nu_t} [(b_t(i,j))] \right)^2 \right] \le c''\beta^{-1} \sum_{t=1}^T \mathbb{E}_{\substack{i \sim \tilde{\mu}_t \\ j \sim \nu_t}} \left[ (b_t(i,j))^2 \right].$$

The final bound is obtained by substituting the expression for the bonus terms and using Lemmas D.2 and D.6 and using analogous arguments to bound the second term in eq. (24) resulting in

$$\text{Regret}(T) \le \mathcal{O} \left( \beta^{-1} d^2 \left( 1 + \sigma^2 \log \left( \frac{T}{\delta} \right) \right) \log \left( \frac{T}{d} \right) \right).$$

REGULARIZATION-INDEPENDENT BOUND

Using the bandits view, the term $T_1$ can be bounded by $\mathcal{O} \left( (1+\sigma)d\sqrt{T} \log \left( \frac{T}{\delta} \right) \right)$ using the similar arguments to ones used in standard UCB bounds as done in section E.2 step 1. We bound $T_2$ by $\mathcal{O} \left( (1+\sigma)d\sqrt{T} \log \left( \frac{T}{\delta} \right) \right)$ as detailed in section E.2 step 2. Similarly bounding the second term in eq. (24) we have

$$\text{Regret}(T) \le \mathcal{O} \left( (1+\sigma)d\sqrt{T} \log \left( \frac{T}{\delta} \right) \right).$$

## B.2. Markov Games

In this section we extend the arguments from the matrix games section to design and analyse the SOMG Algorithm 2 for achieving logarithmic regret in Markov games. We begin by elaborating some algorithmic choices before proceeding with the proof outline. The value function in eq. (9) which can be rewritten as

$$V_h^{\mu_t, \nu_t}(s) :=$$

$$\mathbb{E}^{\mu_t, \nu_t} \left[ \sum_{k=h}^{H} r_k(s_k, i, j) - \beta \mathrm{KL} \left( \mu_k(\cdot|s_k) \| \mu_{\mathrm{ref},k}(\cdot|s_k) \right) + \beta \mathrm{KL} \left( \nu_k(\cdot|s_k) \| \nu_{\mathrm{ref},k}(\cdot|s_k) \right) \middle| s_h = s \right].$$

This can be unbounded from both above and below depending on $\mu_t$ and $\nu_t$ due to the unbounded nature of the KL regularization terms. For instance, if $\nu_t$ deviates substantially from the reference policy $\nu_{\mathrm{ref}}$ in certain states, the max-player can exploit this by selecting policies that steer the MDP toward those states, thereby attaining a higher overall return in regions where the KL divergence between $\nu_t$ and $\nu_{\mathrm{ref}}$ is large. This unbounded nature of the value function is problematic when designing confidence intervals for bellman errors. We address this problem by choosing the policy pair $(\mu_{h,t}, \nu_{h,t})$ to the Nash equilibrium policies under the matrix game $\overline{Q}_{h,t}$ in eq. (17). As a consequence of this choice we have for any $\beta > 0$ (full details in Lemma F.10)

$$\beta \mathrm{KL} \left( \mu_{h,t}(.|s_h) \| \mu_{\mathrm{ref},h}(.|s_h) \right) \in [0, H - h + 1], \tag{26}$$
$$\beta \mathrm{KL} \left( \nu_{h,t}(.|s_h) \| \nu_{\mathrm{ref},h}(.|s_h) \right) \in [0, H - h + 1]. \tag{27}$$

From eq. 27 one can show for the policies $(\mu_t, \nu_t)$ Algorithm 2 chooses, we have $V_h^{\mu_t, \nu_t}(s) \in [-c_1(H - h + 1)^2, c_2(H - h + 1)^2]$. (Lemma F.11) and one can proceed to bound Bellman errors for the resulting policies. This is also the reason our projection operator (20) has the ceiling of the order $(H - h + 1)^2$ as opposed to standard $(H - h + 1)$ as done in most unregularized works (Xie et al., 2023). The constant 3 comes from superoptimism (lemma F.8).

We also use properties of optimism and superoptimistic gap in our proofs. For notational simplicity, while stating the these properties we will omit the superscript $\nu_t$ and also the dependence on $t$. The properties hold for all $t \in [T]$. Consequently, the symbol $\mu$ here should be interpreted as the time-indexed policy $\mu_t$, rather than an arbitrary policy.

**Optimism:** For the setting in algorithm 2 and any policy $\mu'$, we have

$$Q_h^+(s_h, i_h, j_h) \geq \overline{Q}_h(s_h, i_h, j_h) \quad \text{and} \quad Q_h^+(s_h, i_h, j_h) \geq Q_h^{\mu'}(s_h, i_h, j_h). \tag{28}$$

**Superoptimistic gap:** For the setting in algorithm 2, we have

$$2 \left| \left( Q_h^+(s_h, i_h, j_h) - \overline{Q}_h(s_h, i_h, j_h) \right) \right| \geq \left| Q_h^+(s_h, i_h, j_h) - Q_h^\mu(s_h, i_h, j_h) \right|. \tag{29}$$

Standard analysis that achieves $\tilde{\mathcal{O}}(\sqrt{T})$ regret uses just optimism meaning they just need $Q_h^+(s_h, i_h, j_h) \geq Q_h^\dagger(s_h, i_h, j_h)$ and thus they only add the bonus term $b_h(s_h, i_h, j_h)$ to account for the bellman error incurred while regression used to compute $Q_h^+(s_h, i_h, j_h)$ (since the bellman error of the term $Q_h^\dagger(s_h, i_h, j_h)$ is 0). However for our proof technique we additionally require the property in eq. (29) to hold. Under optimism property in eq. (28) the eq. (29) is equivalent to

$$\left( Q_h^+(s_h, i_h, j_h) - \overline{Q}_h(s_h, i_h, j_h) \right) \geq \overline{Q}_h(s_h, i_h, j_h) - Q_h^\mu(s_h, i_h, j_h). \tag{30}$$

This property follows as a consequence of the design of the superoptimistic bonus (23) and projection operator (20). As detailed in Lemma F.8, we enable this by the addition of the bonus $b_h^{\mathrm{sup}}(s_h, i_h, j_h) = b_h(s_h, i_h, j_h) + 2b_h^{\mathrm{mse}}(s_h, i_h, j_h)$ where $b_h^{\mathrm{sup}}(s_h, i_h, j_h)$ adjusts for the Bellman error in the term $Q_h^+(s_h, i_h, j_h)$ while $2b_h^{\mathrm{mse}}(s_h, i_h, j_h)$ adjusts for the bellman errors in the the two $\overline{Q}_h(s_h, i_h, j_h)$ terms while the Bellman error of the term $Q_h^\mu(s_h, i_h, j_h)$ is 0 in (30). The property holds with just plain optimism when $H = 1$ for matrix games.

Lastly note that the bonus is superoptimistic in the sense that we add the term $b_h^{\mathrm{sup}}(s_h, i_h, j_h)$ while constructing $Q_h^+(s_h, i_h, j_h)$ in eq. (16b) although we have with high probability the highest value (optimistic value) of $Q_h^+(s_h, i_h, j_h)$ can be upperbounded just by adding $b_h(s_h, i_h, j_h)$ - the standard *optimistic bonus* yet we add $b_h^{\mathrm{sup}}(s_h, i_h, j_h) = b_h(s_h, i_h, j_h) + 2b_h^{\mathrm{mse}}(s_h, i_h, j_h)$ where $b_h^{\mathrm{mse}}(s_h, i_h, j_h)$ is the bonus used in addition to optimism.

**Design of the Superoptimistic projection operator:** Recall that the projection operator in eq. (20b) is given by

$$\Pi_h^+(x) = \max \left\{ 0, \min\{x, 3(H - h + 1)^2\} \right\}.$$

We can show (Lemma F.11) that the maximum value that can be attained by any policy's $(\mu')$ value function

$$Q_h^{\mu', \nu_t}(s, i, j) \leq (H - h + 1)^2.$$

However, during the projection operation we set the projection ceiling to $3(H - h + 1)^2$. This is again done to facilitate the superoptimistic gap in eq. (29) when the $Q_h^+(s, i, j)$ attains its ceiling value.

The dual gap at time $t$ can be decomposed as follows

$$\text{DualGap}(\mu_t, \nu_t) = V_1^{\star, \nu_t}(s_1) - V_1^{\mu_t, \star}(s_1) = \underbrace{V_1^{\star, \nu_t}(s_1) - V_1^{\mu_t, \nu_t}(s_1)}_{\text{Exploitability of the max player}} + \underbrace{V_1^{\mu_t, \nu_t}(s_1) - V_1^{\mu_t, \star}(s_1)}_{\text{Explotaibility of the min player}}. \tag{31}$$

We elaborate the bounding of the first term (exploitability of the max player) and the bounding of the second term follows analogous arguments. One can further decompose the first term in eq. (31) as

$$V_1^{\star, \nu}(s_1) - V_1^{\mu, \nu}(s_1) = \underbrace{V_1^{\star, \nu}(s_1) - V_1^{\tilde{\mu}, \nu}(s_1)}_{T_5} + \underbrace{V_1^{\tilde{\mu}, \nu}(s_1) - V_1^{\mu, \nu}(s_1)}_{T_6}. \tag{32}$$

**RL view for bounding $T_5$:** As a result of fixed $\nu_t$, one can view the min-player strategy $\nu_t$ as part of the environment and bound $T_5$ the same way as done in RL with the max player as the decision making entity. Here $\mu_h^\dagger$ and $\tilde{\mu}_h$ are stagewise best responses to the fixed strategy $\nu_h$ under matrix games with parameters $Q_h^{\mu^\dagger, \nu}$ and $Q_h^+$ respectively

**Regret Guarantees:** Leveraging the RL view one can bound the term $T_5$ adapting the arguments from (Zhao et al., 2025b) (Theorem 5.1) and accounting for changing $\nu_t$ as detailed in section F.2 step 1 for the logarithmic regret bound and standard single agent RL analysis as detailed in F.3.1 step 1 for the traditional $\sqrt{T}$ bound. This does not require anything beyond the standard optimism property (28). The bounding of $T_6$ is elaborated in section F.2 step 2 for the logarithmic regret bound and section F.3.1 step 2 for the $\sqrt{T}$ and requires both optimism (28) and superoptimistic gap (29) properties.

**Bounding the Covering Number of the Superoptimistic Value Function Class:** The superoptimistic value function class is induced by the $Q$ functions in Algorithm 2 equation (19). The covering number of MSE value functions $\overline{V}$ is easier to bound (lemma F.14) and relies on the smoothness of the value of the game under payoff matrix perturbations arguments. Meanwhile, the superoptimistic value functions $V^+$ ($V^-$ can be bounded using symmetric arguments) is harder to bound (lemma F.15) as it is the value function corresponding to the best response under $Q^+$ to the min player Nash equilibrium policy under $\overline{Q}$. We exploit the smoothness property of the KL regularized setting to obtain this covering number and subsequently use an appropriate cover on the $Q^+$ and $\overline{Q}$ function classes to get the covering number of the superoptimistic value function class (lemma F.15). Moreover, since the $Q$ functions use different projection operators for different $h \in [H]$ we have to split the $\delta$ budget across $h \in [H]$. A similar argument can be extended to the unregularized setting also getting rid of the $\log(1/\min\{1, \beta\})$ term in the bounds using discretization of the $Q$ function class (Xie et al., 2023).

## C. Numerical Experiments

To evaluate whether SOMG (Algorithm 2) stabilizes learning, we conduct experiments on randomly generated linear MDPs, as shown in Figure 1. We randomly generate two MDP environments with the parameter settings indicated in the figure and track the dual gap (log scale) as a function of the number of collected trajectories. Note that in each iteration of Step 11 in SOMG, two trajectories are sampled. The reference policies for both the players $(\mu_{\text{ref}}, \nu_{\text{ref}})$ for all states is set to uniform of actions (Entropy regularization).

For each MDP we compute the stagewise Nash equilibrium which essentially involves solving a zero sum KL regularized matrix game (SOMG step 7 equation (17)) with the estimated MSE $Q$ function $\overline{Q}_{h,t}(s, \cdot, \cdot)$ as the payoff matrix for step $h$ at time $t$. The estimated game is then solved using policy extragradient methods. More specifically we use the Predictive Update (PU) method from Algorithm 1 in (Cen et al., 2024) which given a payoff matrix can find the $\varepsilon_{\text{comp}}$-NE in $\log(1/\varepsilon_{\text{comp}})$ steps. The plots for both the settings are shown for 5 different values of the regularization strength $\beta = [0.01, 0.05, 0.1, 0.2, 0.5]$ with higher $\beta$ demonstrating faster convergence validating our theoretical results from section 3.

## D. Useful lemmas

**Lemma D.1** (Covering number of the $\ell_2$ ball, Lemma D.5 in (Jin et al., 2020)). *For any $\epsilon > 0$ and $d \in \mathbb{N}^+$, the $\epsilon$-covering number of the $\ell_2$ ball of radius $R$ in $\mathbb{R}^d$ is at most $\left(1 + \frac{2R}{\epsilon}\right)^d$.*

**Lemma D.2** (Martingale Concentration, Lemma B.2 in (Foster et al., 2021)). *Let $(X_t)_{t \leq T}$ be a sequence of real-valued random variables adapted to a filtration $\mathcal{F}_t$ and $\mathbb{E}_t[\cdot] := \mathbb{E}[\cdot | \mathcal{F}_t]$ denote the conditional expectation. Suppose that $|X_t| \leq R$*

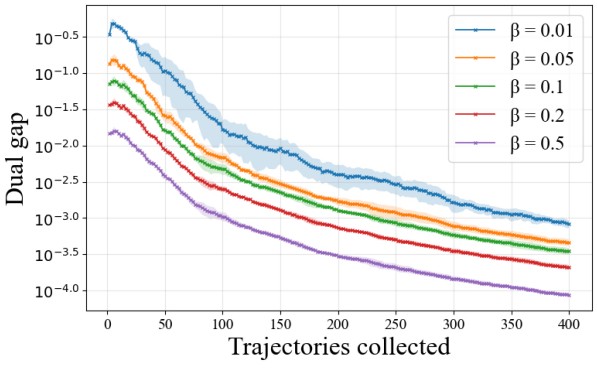 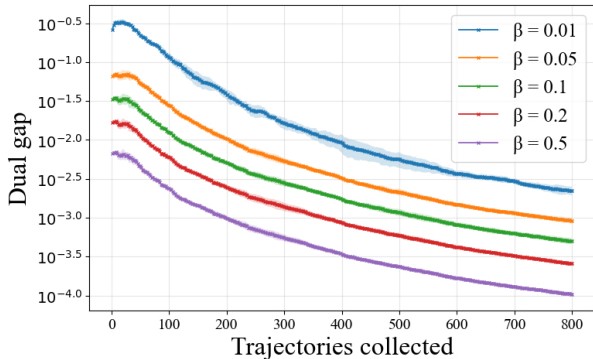

*(a) $H = 4, |S| = 7, A_1 = A_2 = 7, d = 5$*  *(b) $H = 7, |S| = 20, A_1 = A_2 = 11, d = 10$*

*Figure 1.* Dual gap (log scale) vs trajectories collected for KL regularized Markov Games, $H$ denotes the horizon length, $|S|$ denotes the number of states, $A_i$ denotes the number of actions of player $i$ and $d$ denotes the feature dimension. The spread shows standard deviation averaged over 3 runs

*almost surely for all $t$. Then, with probability at least $1 - \delta$, the following inequalities hold:*

$$\sum_{t=1}^{T} X_t \leq \frac{3}{2}\sum_{t=1}^{T}\mathbb{E}_{t-1}[X_t] + 4R\log(2\delta^{-1}), \quad and \quad \sum_{t=1}^{T}\mathbb{E}_{t-1}[X_t] \leq 2\sum_{t=1}^{T} X_t + 8R\log(2\delta^{-1}).$$

**Lemma D.3** (Confidence Ellipsoid: Theorem 2 (Abbasi-Yadkori et al., 2011)). *Let $\xi_t$ be a conditionally $R$ sub-gaussian random variable adapted to the filtration $\mathcal{F}_t$ and $\{X_t\}_{t=1}^{\infty}$, $\|X_t\| \leq L$ be a $\mathcal{F}_{t-1}$ measurable stochastic process in $\mathbb{R}^d$. Define $Y_t = \langle X_t, \theta_\star\rangle + \xi_t$ where $\|\theta_\star\|_2 \leq \sqrt{S}$. Let $\bar{\theta}_t$ be the solution to the regularized least squares problem given by*

$$\bar{\theta}_t = \arg\min_{\theta\in\mathbb{R}}\sum_{i=1}^{t-1}\left(\langle X_t, \theta\rangle - Y_t\right)^2 + \lambda\|\theta\|_2^2,$$

*then for any $\delta \in [0, 1]$, for all $t \geq 0$, with probability atleast $1 - \delta$ we have*

$$\left\|\bar{\theta}_t - \theta_\star\right\|_{V_t} \leq R\sqrt{d\log\left(\frac{1 + tL^2/\lambda}{\delta}\right)} + \sqrt{\lambda S}.$$

**Lemma D.4** (Lemma 11 in (Abbasi-Yadkori et al., 2011)). *Let $\{\phi_s\}_{s\in[T]}$ be a sequence of vectors with $\phi_s \in \mathbb{R}^d$ and $\|\phi_s\| \leq L$. Suppose $\Lambda_0$ is a positive definite matrix and define $\Lambda_t = \Lambda_0 + \sum_{s=1}^{t}\phi_s\phi_s^\top$. Then if $\lambda_{\min}(\Lambda_0) > \max\{1, L^2\}$, the following inequality holds:*

$$\sum_{s=1}^{T}\min\left\{1, \|\phi_s\|_{\Lambda_{s-1}^{-1}}^2\right\} \leq 2\log\left(\frac{\det(\Lambda_T)}{\det(\Lambda_0)}\right).$$

**Lemma D.5** (Lemma F.3 in (Du et al., 2021)). *Let $\mathcal{X} \subset \mathbb{R}^d$ and suppose $\sup_{x\in\mathcal{X}}\|x\|_2 \leq B_\mathcal{X}$. Then for any $n \in \mathbb{N}$, we have*

$$\forall \lambda > 0 : \quad \max_{x_1,\ldots,x_n\in\mathcal{X}}\log\det\left(I_d + \frac{1}{\lambda}\sum_{i=1}^{n} x_i x_i^\top\right) \leq d\log\left(1 + \frac{nB_\mathcal{X}^2}{d\lambda}\right).$$

As a direct consequence of lemmas D.4 and D.5 we have

**Lemma D.6** (Elliptical Potential Lemma). *Let $\mathbf{x}_1, \ldots, \mathbf{x}_T \in \mathbb{R}^d$ satisfy $\|\mathbf{x}_t\|_2 \leq 1$ for all $t \in [T]$. Fix $\lambda > 0$, and let $V_t = \lambda\mathbf{I} + \sum_{i=1}^{t-1}\mathbf{x}_i\mathbf{x}_i^\top$. Then*

$$\sum_{t=1}^{T}\min\left\{1, \|\mathbf{x}_t\|_{V_t^{-1}}^2\right\} \leq 2d\log\left(1 + \lambda^{-1}T/d\right).$$

*Specifically for $\lambda = 1$ we have*

$$\sum_{t=1}^{T} \min\left\{1, \|\mathbf{x}_t\|_{V_t^{-1}}^2\right\} = \sum_{t=1}^{T} \|\mathbf{x}_t\|_{V_t^{-1}}^2 \leq 2d \log\left(1 + T/d\right).$$

**Lemma D.7** (Lemma D.1 in (Jin et al., 2020)). *Consider the matrix $\Sigma_t = \lambda \mathbf{I} + \sum_{i=1}^{t-1} \phi_i \phi_i^\top$, where $\phi_i \in \mathbb{R}^d$ and $\lambda > 0$. Then the following inequality holds $\forall\, t$:*

$$\sum_{i=1}^{t-1} \phi_i^\top \Sigma_t^{-1} \phi_i \leq d.$$

**Lemma D.8** (Lemma D.4 in (Jin et al., 2020)). *Consider a stochastic process $\{s_\tau\}_{\tau=1}^{\infty}$ on a state space $\mathcal{S}$ with associated filtration $\{\mathcal{F}_\tau\}_{\tau=0}^{\infty}$, and an $\mathbb{R}^d$-valued process $\{\phi_\tau\}_{\tau=0}^{\infty}$ such that $\phi_\tau \in \mathcal{F}_{\tau-1}$ and $\|\phi_\tau\| \leq 1$. Define $\Lambda_k = \lambda \mathbf{I} + \sum_{\tau=1}^{k} \phi_\tau \phi_\tau^\top$. Let $\mathcal{V}$ be a function class such that $\sup_x |V(x)| \leq B_1$ for some constant $B_1 > 0$, and let $\mathcal{N}_\epsilon$ be its $\epsilon$-covering number under the distance $\mathrm{dist}(V_1, V_2) = \sup_s |V_1(s) - V_2(s)|$. Then, for any $\delta \in (0,1)$, with probability at least $1 - \delta$, for all $k \geq 0$ and any $V \in \mathcal{V}$, we have:*

$$\left\| \sum_{\tau=1}^{k} \phi_\tau \left\{ V(s_\tau) - \mathbb{E}[V(s_\tau)|\mathcal{F}_{\tau-1}] \right\} \right\|_{\Lambda_k^{-1}}^2 \leq 4B_1^2 \left[ \frac{d}{2} \log\left(\frac{k+\lambda}{\lambda}\right) + \log\left(\frac{\mathcal{N}_\epsilon}{\delta}\right) \right] + \frac{8k^2 \epsilon^2}{\lambda}.$$

# E. Matrix Game Proofs

**Proposition E.1** (Optimism/Concentration). *Let $\mathcal{E}_1$ be the event $\|\overline{\omega}_t - \omega^\star\|_{\Sigma_t} \leq \eta_T$, then we have $\mathbb{P}(\mathcal{E}_1) \geq (1 - \delta/3)$, under the event $\mathcal{E}_1$ we have*

$$|(\overline{A}_t(i,j) - A(i,j))| \leq b_t(i,j) \quad \forall(i,j), \tag{33a}$$

$$A_t^+(i,j) - A(i,j) \leq 2b_t(i,j) \quad \text{and} \quad A_t^+(i,j) \geq A(i,j) \quad \forall(i,j), \tag{33b}$$

$$A(i,j) - A_t^-(i,j) \leq 2b_t(i,j) \quad \text{and} \quad A(i,j) \geq A_t^-(i,j) \quad \forall(i,j), \tag{33c}$$

*where $b_t(i,j) = \eta_T \|\phi(i,j)\|_{\Sigma_t^{-1}}$ and $\eta_T = \sigma\sqrt{d \log\left(\frac{3(1+2T/\lambda)}{\delta}\right)} + \sqrt{\lambda d}$.*

*Proof.* Recall that $\overline{\omega}_t$ is computed in algorithm 1 as

$$\overline{\omega}_t = \arg\min_{\omega \in \mathbb{R}^d} \sum_{(i,j,\widehat{A}(i,j)) \in \mathcal{D}_{t-1}} \left( A_\omega(i,j) - \widehat{A}(i,j) \right)^2 + \lambda \|\omega\|_2^2.$$

Now using Lemma D.3 with $S = d$, $L = 1$ (assumption 2.1) and accounting for the $2(t-1)$ points collected until $t$, we have $\forall\, t \geq 0$

$$\|\overline{\omega}_t - \omega^\star\|_{\Sigma_t} \leq \sigma\sqrt{d \log\left(\frac{3(1+2t/\lambda)}{\delta}\right)} + \sqrt{\lambda d} \quad \text{w.p. } 1 - \delta/3. \tag{34}$$

Since $\eta_T = \sigma\sqrt{d \log\left(\frac{3(1+2T/\lambda)}{\delta}\right)} + \sqrt{\lambda d}$ we have $\mathbb{P}(\mathcal{E}_1) = 1 - \delta/3$. Using eq. (34) we have

$$|(\overline{A}_t(i,j) - A(i,j)| = |\langle \overline{\omega}_t - \omega^\star, \phi(i,j)\rangle| \leq \|\overline{\omega}_t - \omega^\star\|_{\Sigma_t} \|\phi(i,j)\|_{\Sigma_t^{-1}}$$

$$\leq \left( \sigma\sqrt{d \log\left(\frac{3(1+T/\lambda)}{\delta}\right)} + \sqrt{\lambda d} \right) \|\phi(i,j)\|_{\Sigma_t^{-1}}$$

$$= \eta_T \|\phi(i,j)\|_{\Sigma_t^{-1}} = b_t(i,j) \tag{35}$$

Here eq. (35) follows from the result in eq. (34) under the event $\mathcal{E}_1$. Lastly $A_t^+(i,j) = \overline{A}_t(i,j) + b_t(i,j)$ implies $0 \leq A_t^+(i,j) - A(i,j) \leq 2b_t(i,j)$. Similar arguments can be used to prove eq. (33c). $\qquad\square$

Now Theorem 2.3 holds as long as for any fixed $\delta \in [0, 1]$, for some events $\mathcal{E}_{\text{dep}}^{\text{matrix}}$, $\mathcal{E}_{\text{ind}}^{\text{matrix}}$ and $\mathcal{E}^{\text{matrix}} := \mathcal{E}_{\text{dep}}^{\text{matrix}} \cap \mathcal{E}_{\text{ind}}^{\text{matrix}}$ with $\mathbb{P}(\mathcal{E}^{\text{matrix}}) \geq 1 - \delta$ the following theorems can be established.

**Theorem E.2** (Regularization-dependent guarantee). *Under assumptions 2.1 and 2.2, for any $\beta > 0$, reference policies $(\mu_{ref}, \nu_{ref})$, choosing $\lambda = 1$ and $b_t(i, j)$ as per eq. (5) in Algorithm 1, under the event $\mathcal{E}_{dep}^{matrix}$ we have*

$$\forall \, T \in \mathbb{N}^+ : \qquad \text{Regret}(T) \leq \mathcal{O}\left( \beta^{-1} d^2 \left( 1 + \sigma^2 \log\left( \frac{T}{\delta} \right) \right) \log\left( \frac{T}{d} \right) \right).$$

**Theorem E.3** (Regularization-independent guarantee). *Under assumptions 2.1 and 2.2, $\beta \geq 0$, reference policies $(\mu_{ref}, \nu_{ref})$, choosing $\lambda = 1$ and $b_t(i, j)$ as per eq. (5) in Algorithm 1, under the event $\mathcal{E}_{ind}^{matrix}$ we have*

$$\forall \, T \in \mathbb{N}^+ : \qquad \text{Regret}(T) \leq \mathcal{O}\left( (1 + \sigma) d \sqrt{T} \log\left( \frac{T}{\delta} \right) \right).$$

### E.1. Proof of Theorem E.2: Regularization-Dependent Bound

The regret can be upper bounded as follows

$$
\begin{aligned}
\text{Regret(T)} &= \sum_{t=1}^{T} \left( f^{\star, \nu_t}(A) - f^{\mu_t, \star}(A) \right) \\
&= \underbrace{\sum_{t=1}^{T} \left( f^{\star, \nu_t}(A) - f^{\tilde{\mu}_t, \nu_t}(A) \right)}_{T_1} + \underbrace{\sum_{t=1}^{T} \left( f^{\tilde{\mu}_t, \nu_t}(A) - f^{\mu_t, \nu_t}(A) \right)}_{T_2} \\
&\quad + \underbrace{\sum_{t=1}^{T} \left( f^{\mu_t, \nu_t}(A) - f^{\mu_t, \tilde{\nu}_t}(A) \right)}_{T_3} + \underbrace{\sum_{t=1}^{T} \left( f^{\mu_t, \tilde{\nu}_t}(A) - f^{\mu_t, \star}(A) \right)}_{T_4}.
\end{aligned}
\tag{36}
$$

Here we will bound the terms $T_1$ and $T_2$, the terms $T_3$ and $T_4$ can be bounded similarly. We use $\mu(A', \nu') := \arg\max_\mu f^{\mu, \nu'}(A')$ to denote the max player's best response strategy to $\nu'$ under the payoff matrix $A'$. Similarly one can define $\nu(A', \mu')$. One can derive the closed form expressions for the best response to $\nu_t$ under models $A$, $A_t^+$ and $\overline{A}_t$ to be $\mu_t^\dagger$, $\tilde{\mu}_t$ and $\mu_t$ respectively by solving eq. (4) to be

$$\mu_{t,i}^\dagger = \mu(A, \nu_t)_i = \arg\max_\mu f^{\mu, \nu_t}(A) = \mu_{\text{ref},i} \exp\left( \frac{A(i,:)\nu_t}{\beta} \right) \Big/ Z(A, \nu_t), \tag{37a}$$

$$\tilde{\mu}_{t,i} = \mu(A_t^+, \nu_t)_i = \arg\max_\mu f^{\mu, \nu_t}(A_t^+) = \mu_{\text{ref},i} \exp\left( \frac{A_t^+(i,:)\nu_t}{\beta} \right) \Big/ Z(A_t^+, \nu_t), \tag{37b}$$

$$\mu_{t,i} = \mu(\overline{A}_t, \nu_t)_i = \arg\max_\mu f^{\mu, \nu_t}(\overline{A}_t) = \mu_{\text{ref},i} \exp\left( \frac{\overline{A}_t(i,:)\nu_t}{\beta} \right) \Big/ Z(\overline{A}_t, \nu_t), \tag{37c}$$

where

$$Z(A', \nu') = \sum_i \mu_{\text{ref},i} \exp\left( \frac{A'(i,:)\nu'}{\beta} \right).$$

**Step 1: Bounding $T_1$** From definition of the objective function (1) we have

$$f^{\star, \nu_t}(A) - f^{\tilde{\mu}_t, \nu_t}(A) = \mathop{\mathbb{E}}_{\substack{i \sim \mu_t^\dagger \\ j \sim \nu_t}} [A(i,j)] - \beta \text{KL}(\mu_t^\dagger || \mu_{\text{ref}}) - \left( \mathop{\mathbb{E}}_{\substack{i \sim \tilde{\mu}_t \\ j \sim \nu_t}} [A(i,j)] - \beta \text{KL}(\tilde{\mu}_t || \mu_{\text{ref}}) \right) \tag{38}$$

$$= \beta \log(Z(A, \nu_t)) - \beta \log(Z(A_t^+, \nu_t)) + \tilde{\mu}_t^\top (A_t^+ - A) \nu_t \tag{39}$$

$$= \Delta(A_t^+, \nu_t) - \Delta(A, \nu_t), \tag{40}$$

where we define $\Delta(A', \nu') = -\beta \log(Z(A', \nu')) + \mu(A', \nu')^\top (A' - A)\nu'$. Eq. (39) follows from the closed form expressions for the best responses (37). Using the mean value theorem for some $\Gamma \in [0, 1]$ with $A_\Gamma = \Gamma A_t^+ + (1 - \Gamma)A$ we have

$$
\begin{aligned}
&f^{\star, \nu_t}(A) - f^{\tilde{\mu}_t, \nu_t}(A) \\
&= \Delta(A_t^+, \nu_t) - \Delta(A, \nu_t) \\
&= \sum_i \frac{\partial \Delta(A_\Gamma, \nu_t)}{\partial (A_\Gamma(i, :)\nu_t)} (A_t^+(i, :) - A(i, :))\nu_t \\
&= \sum_i \left( \beta^{-1} \mu(A_\Gamma, \nu_t)_i \left[ (A_\Gamma(i, :) - A(i, :))\nu_t \right. \right. \\
&\qquad\qquad \left. \left. - \mathop{\mathbb{E}}_{i' \sim \mu(A_\Gamma, \nu_t)} [(A_\Gamma(i', :) - A(i', :))\nu_t] \right] \right) (A_t^+(i, :) - A(i, :))\nu_t \\
&= \sum_i \left( \Gamma \beta^{-1} \mu(A_\Gamma, \nu_t)_i \left[ (A_t^+(i, :) - A(i, :))\nu_t \right. \right. \\
&\qquad\qquad \left. \left. - \mathop{\mathbb{E}}_{i' \sim \mu(A_\Gamma, \nu_t)} [(A_t^+(i', :) - A(i', :))\nu_t] \right] \right) (A_t^+(i, :) - A(i, :))\nu_t \\
&= \Gamma \beta^{-1} \left( \mathop{\mathbb{E}}_{i \sim \mu(A_\Gamma, \nu_t)} \left[ ((A_t^+(i, :) - A(i, :))\nu_t)^2 \right] - \left( \mathop{\mathbb{E}}_{i \sim \mu(A_\Gamma, \nu_t)} \left[ (A_t^+(i, :) - A(i, :)) \nu_t \right] \right)^2 \right) \\
&\leq \beta^{-1} \mathop{\mathbb{E}}_{i \sim \mu(A_\Gamma, \nu_t)} \left[ ((A_t^+(i, :) - A(i, :))\nu_t)^2 \right].
\end{aligned}
$$

(41)

(42)

Here eq. (41) follows from Lemma E.4. Let $d_t(i) = \mathbb{E}_{j \sim \nu_t} \left[ (A_t^+(i, j) - A(i, j)) \right]$, now consider the term

$$
\begin{aligned}
G_1(\Gamma) &:= \mathop{\mathbb{E}}_{i \sim \mu(A_\Gamma, \nu_t)} \left[ ((A_t^+(i, :) - A(i, :))\nu_t)^2 \right] \\
&= \sum_i \left( \mathop{\mathbb{E}}_{j \sim \nu_t} [(A_t^+(i, j) - A(i, j))] \right)^2 \mu(A_\Gamma, \nu_t)_i = \sum_i d_t(i)^2 \mu(A_\Gamma, \nu_t)_i.
\end{aligned}
$$

(43)

Under the event $\mathcal{E}_1$ (Proposition E.1), we have

$$
\begin{aligned}
&\frac{\partial G_1(\Gamma)}{\partial \Gamma} \\
&= \sum_i (d_t(i))^2 \frac{\partial \mu(A_\Gamma, \nu_t)_i}{\partial \Gamma} \\
&= \sum_i (d_t(i))^2 \left\{ \frac{\mu_{\mathrm{ref}, i} \exp \left( \beta^{-1} (A(i, :)\nu_t + \Gamma d_t(i)) \right)}{\sum_{i'} \mu_{\mathrm{ref}, i'} \exp \left( \beta^{-1} (A(i', :)\nu_t + \Gamma d_t(i')) \right)} \beta^{-1} d_t(i) \right. \\
&\quad \left. - \frac{\mu_{\mathrm{ref}, i} \exp \left( \beta^{-1} (A(i, :)\nu_t + \Gamma d_t(i)) \right) \sum_{i'} \beta^{-1} d_t(i') \mu_{\mathrm{ref}, i'} \exp \left( \beta^{-1} (A(i', :)\nu_t + \Gamma d_t(i')) \right)}{\left( \sum_{i'} \mu_{\mathrm{ref}, i'} \exp \left( \beta^{-1} (A(i', :)\nu_t + \Gamma d_t(i')) \right) \right)^2} \right\} \\
&= \beta^{-1} \left( \mathop{\mathbb{E}}_{i \sim \mu(A_\Gamma, \nu_t)} [d_t(i)^3] - \mathop{\mathbb{E}}_{i \sim \mu(A_\Gamma, \nu_t)} [d_t(i)^2] \mathop{\mathbb{E}}_{i \sim \mu(A_\Gamma, \nu_t)} [d_t(i)] \right) \\
&= \beta^{-1} \mathrm{Cov}(d_t(i), d_t(i)^2) \geq 0.
\end{aligned}
$$

(44)

Here eq. (44) follows since under the event $\mathcal{E}_1$ we have $d_t(i) \geq 0 \ \forall i$ and for any positive random variable $X$

$$
\begin{aligned}
\mathrm{Cov}(X, X^2) &= \mathbb{E}[X^3] - \mathbb{E}[X^2]\mathbb{E}[X] = \mathbb{E} \left[ (X^2)^{3/2} \right] - \mathbb{E}[X^2]\mathbb{E}[X] \\
&\geq \left( \mathbb{E} \left[ X^2 \right] \right)^{3/2} - \mathbb{E}[X^2]\mathbb{E}[X] = \mathbb{E}[X^2] \left( \sqrt{\mathbb{E}[X^2]} - \mathbb{E}[X] \right) \geq 0.
\end{aligned}
$$

(45)

Thus we have $G_1(\Gamma) \leq G_1(1)$ and using eq. (42)

$$f^{\star,\nu_t}(A) - f^{\tilde{\mu}_t,\nu_t}(A) \leq \beta^{-1}G_1(\Gamma)$$

$$\leq \beta^{-1}G_1(1) = \beta^{-1} \mathop{\mathbb{E}}_{i \sim \mu(A_t^+,\nu_t)} \left[ \left( (A_t^+(i,:) - A(i,:))\nu_t \right)^2 \right] \tag{46}$$

$$\leq 4\beta^{-1} \mathop{\mathbb{E}}_{i \sim \mu(A_t^+,\nu_t)} \left[ \left( \mathop{\mathbb{E}}_{j \sim \nu_t} [b_t(i,j)] \right)^2 \right], \tag{47}$$

where the last inequality follows from Proposition E.1 under the event $\mathcal{E}_1$.

**Step 2: Bounding $T_2$**
From the definition of the objective function (1) we have

$$f^{\tilde{\mu}_t,\nu_t}(A) - f^{\mu_t,\nu_t}(A)$$

$$= \mathop{\mathbb{E}}_{\substack{i \sim \tilde{\mu}_t \\ j \sim \nu_t}} [A(i,j)] - \beta \mathrm{KL}(\tilde{\mu}_t || \mu_{\mathrm{ref}}) - \left( \mathop{\mathbb{E}}_{\substack{i \sim \mu_t \\ j \sim \nu_t}} [A(i,j)] - \beta \mathrm{KL}(\mu_t || \mu_{\mathrm{ref}}) \right) \tag{48}$$

$$= \left( \beta \log(Z(A_t^+,\nu_t)) - \tilde{\mu}_t^\top (A_t^+ - A)\nu_t \right) - \left( \beta \log(Z(\overline{A}_t,\nu_t)) - \mu_t^\top (\overline{A}_t - A)\nu_t \right) \tag{49}$$

$$= \Delta(\overline{A}_t,\nu_t) - \Delta(A_t^+,\nu_t). \tag{50}$$

Eq. (49) follows from the closed form expressions for the best responses (37). Using the mean value theorem for some $\Gamma \in [0,1]$ with $A_\Gamma = \Gamma \overline{A}_t + (1-\Gamma)A_t^+$ we have

$$f^{\tilde{\mu}_t,\nu_t}(A) - f^{\mu_t,\nu_t}(A)$$

$$= \Delta(\overline{A}_t,\nu_t) - \Delta(A_t^+,\nu_t)$$

$$= \sum_i \frac{\partial \Delta(A_\Gamma,\nu_t)}{\partial (A_\Gamma(i,:)\nu_t)} (\overline{A}_t(i,:) - A_t^+(i,:))\nu_t$$

$$= \sum_i \left( \beta^{-1}\mu(A_\Gamma,\nu_t)_i \left[ (A_\Gamma(i,:) - A(i,:))\nu_t \right. \right.$$

$$\left. \left. - \mathop{\mathbb{E}}_{i' \sim \mu(A_\Gamma,\nu_t)} [(A_\Gamma(i',:) - A(i',:))\nu_t] \right] \right) (\overline{A}_t(i,:) - A_t^+(i,:))\nu_t$$

$$= \beta^{-1}(\mathbb{E}[XY] - \mathbb{E}[X]\mathbb{E}[Y]), \tag{51}$$

where the penultimate equality follows from Lemma E.4, and in the last line we define $X = (A_\Gamma(i,:) - A(i,:))\nu_t$, $Y = (\overline{A}_t(i,:) - A_t^+(i,:))\nu_t$, and the expectation is taken w.r.t. $i \sim \mu(A_\Gamma,\nu_t)$. Note that

$$X = \Gamma \underbrace{(\overline{A}_t(i,:) - A(i,:))\nu_t}_{:=p} + (1-\Gamma)\underbrace{(A_t^+(i,:) - A(i,:))\nu_t}_{:=q} = \Gamma(p-q) + q,$$

and

$$Y = (\overline{A}_t(i,:) - A(i,:))\nu_t - (A_t^+(i,:) - A(i,:))\nu_t = p - q.$$

Thus

$$\mathbb{E}[XY] - \mathbb{E}[X]\mathbb{E}[Y] = \mathbb{E}[\Gamma(p-q)^2 + q(p-q)] - \Gamma(\mathbb{E}[p-q])^2 - \mathbb{E}[q]\mathbb{E}[(p-q)]$$

$$= \Gamma \mathrm{var}(p-q) + \mathrm{Cov}(q, p-q)$$

$$\leq \mathbb{E}[(p-q)^2] + \max\{\mathbb{E}[q^2], \mathbb{E}[(p-q)^2]\}. \tag{52}$$

By equations (51) and (52) we know that, under the event $\mathcal{E}_1$,

$$f^{\tilde{\mu}_t,\nu_t}(A) - f^{\mu_t,\nu_t}(A)$$

$$\leq \beta^{-1} \mathop{\mathbb{E}}_{i \sim \mu(A_\Gamma, \nu_t)} [((\overline{A}_t(i,:) - A_t^+(i,:))\nu_t)^2]+$$

$$\beta^{-1} \max \left\{ \mathop{\mathbb{E}}_{i \sim \mu(A_\Gamma, \nu_t)} [((\overline{A}_t(i,:) - A_t^+(i,:))\nu_t)^2], \mathop{\mathbb{E}}_{i \sim \mu(A_\Gamma, \nu_t)} [((A_t^+(i,:) - A(i,:))\nu_t)^2] \right\}$$

$$\leq 5\beta^{-1} \mathop{\mathbb{E}}_{i \sim \mu(A_\Gamma, \nu_t)} \left[ \left( \mathop{\mathbb{E}}_{j \sim \nu_t} [b_t(i,j)] \right)^2 \right] = 5\beta^{-1} \mathop{\mathbb{E}}_{i \sim \mu(A_\Gamma, \nu_t)} \left[ (|A_t^+(i,:) - \overline{A}_t(i,:)|\nu_t)^2 \right], \tag{53}$$

where the last inequality follows from the fact that $(|A_t^+(i,:) - \overline{A}_t(i,:)|\nu_t) = \mathbb{E}_{j \sim \nu_t}[b_t(i,j)]$ and $(|A(i,:) - A_t^+(i,:)|\nu_t) \leq 2\mathbb{E}_{j \sim \nu_t}[b_t(i,j)] = 2(|A_t^+(i,:) - \overline{A}_t(i,:)|\nu_t)$ given by Proposition E.1. One can also bound the same thing slightly tighter as follows

$$\mathbb{E}[XY] - \mathbb{E}[X]\mathbb{E}[Y]$$
$$= \mathbb{E}[p(p-q) - (1-\Gamma)(q-p)^2] - \mathbb{E}[p-q]\mathbb{E}[(1-\Gamma)(q-p)] - \mathbb{E}[p-q]\mathbb{E}[p]$$
$$= \mathrm{Cov}(p, p-q) - (1-\Gamma)\mathrm{Var}(p-q) \leq \max \left\{ \mathbb{E}[p^2], \mathbb{E}[(p-q)^2] \right\}. \tag{54}$$

under the event $\mathcal{E}_1$ (c.f. Proposition E.1), using eqs. (51) and (54) we have

$$f^{\tilde{\mu}_t, \nu_t}(A) - f^{\mu_t, \nu_t}(A)$$

$$\leq \beta^{-1} \max \left\{ \mathop{\mathbb{E}}_{i \sim \mu(A_\Gamma, \nu_t)} [((\overline{A}_t(i,:) - A_t^+(i,:))\nu_t)^2], \mathop{\mathbb{E}}_{i \sim \mu(A_\Gamma, \nu_t)} [((\overline{A}_t(i,:) - A(i,:))\nu_t)^2] \right\}$$

$$\leq \beta^{-1} \mathop{\mathbb{E}}_{i \sim \mu(A_\Gamma, \nu_t)} \left[ \left( \mathop{\mathbb{E}}_{j \sim \nu_t} [b_t(i,j)] \right)^2 \right] = \beta^{-1} \mathop{\mathbb{E}}_{i \sim \mu(A_\Gamma, \nu_t)} \left[ \left( \mathop{\mathbb{E}}_{j \sim \nu_t} [A_t^+(i,j) - \overline{A}_t(i,j)] \right)^2 \right], \tag{55}$$

where the last inequality follows from Proposition E.1. Now let $\overline{d_t}(i) := \mathbb{E}_{j \sim \nu_t} \left[ A_t^+(i,j) - \overline{A}_t(i,j) \right]$ and consider the term

$$G_2(\Gamma) := \mathop{\mathbb{E}}_{i \sim \mu(A_\Gamma, \nu_t)} \left[ \left( \mathop{\mathbb{E}}_{j \sim \nu_t} \left[ A_t^+(i,j) - \overline{A}_t(i,j) \right] \right)^2 \right] = \sum_i \left( \overline{d_t}(i) \right)^2 \mu(A_\Gamma, \nu_t)_i. \tag{56}$$

Let $\check{\Gamma} = 1 - \Gamma$, then we have

$$\frac{\partial G_2(\Gamma)}{\partial \Gamma} = \sum_i \left( \overline{d_t}(i) \right)^2 \frac{\partial \mu(A_\Gamma, \nu_t)_i}{\partial \Gamma}$$

$$= \sum_i \left( \overline{d_t}(i) \right)^2 \left\{ - \frac{\mu_{\mathrm{ref},i} \exp \left( \beta^{-1} \left( \overline{A}_t(i,:)\nu_t + \check{\Gamma}\overline{d_t}(i) \right) \right)}{\sum_{i'} \mu_{\mathrm{ref},i'} \exp \left( \beta^{-1} \left( \overline{A}_t(i',:)\nu_t + \check{\Gamma}\overline{d_t}(i') \right) \right)} \beta^{-1} \overline{d_t}(i) \right.$$

$$\left. + \frac{\mu_{\mathrm{ref},i} \exp \left( \beta^{-1} \left( \overline{A}_t(i,:)\nu_t + \check{\Gamma}\overline{d_t}(i) \right) \right) \sum_{i'} \beta^{-1}\overline{d_t}(i')\mu_{\mathrm{ref},i'} \exp \left( \beta^{-1} \left( \overline{A}_t(i',:)\nu_t + \check{\Gamma}\overline{d_t}(i') \right) \right)}{\left( \sum_{i'} \mu_{\mathrm{ref},i'} \exp \left( \beta^{-1} \left( \overline{A}_t(i',:)\nu_t + \check{\Gamma}\overline{d_t}(i') \right) \right) \right)^2} \right\}$$

$$= -\beta^{-1} \left( \mathop{\mathbb{E}}_{i \sim \mu(A_\Gamma, \nu_t)} \left[ \left( \overline{d_t}(i) \right)^3 \right] - \mathop{\mathbb{E}}_{i \sim \mu(A_\Gamma, \nu_t)} \left[ \left( \overline{d_t}(i) \right)^2 \right] \mathop{\mathbb{E}}_{i \sim \mu(A_\Gamma, \nu_t)} \left[ \overline{d_t}(i) \right] \right)$$

$$= -\beta^{-1} \mathrm{Cov}(\overline{d_t}(i)^2, \overline{d_t}(i)) \leq 0, \tag{57}$$

last line follows since under the event $\mathcal{E}_1$ we have $\overline{d_t}(i) \geq 0 \; \forall i$ and for any positive random variable $X$ using eq. (45) we have $\mathrm{Cov}(X, X^2) \geq 0$ Thus the term $G_2(\Gamma) \leq G_2(0)$. Hence from eq. (55) we have

$$T_2 = f^{\tilde{\mu}_t, \nu_t}(A) - f^{\mu_t, \nu_t}(A)$$

$$\leq \beta^{-1} \mathop{\mathbb{E}}_{i \sim \mu(A_\Gamma, \nu_t)} \left[ \left( \overline{d_t}(i) \right)^2 \right] = \beta^{-1} G_2(\Gamma)$$

$$\leq \beta^{-1} G_2(0) = \beta^{-1} \mathop{\mathbb{E}}_{i \sim \mu(A_t^+, \nu_t)} \left[ \left( \overline{d_t}(i) \right)^2 \right] = \beta^{-1} \mathop{\mathbb{E}}_{i \sim \mu(A_t^+, \nu_t)} \left[ \left( \mathop{\mathbb{E}}_{j \sim \nu_t} [b_t(i,j)] \right)^2 \right]. \tag{58}$$

**Step 3: Finishing up**

From equations (47) and (58) w.p. $1 - \delta/3$ (Under event $\mathcal{E}_1$) we have

$$T_1 + T_2 \leq 5\beta^{-1} \sum_{t=1}^{T} \underset{i \sim \mu(A_t^+, \nu_t)}{\mathbb{E}} \left[ \left( \underset{j \sim \nu_t}{\mathbb{E}} [b_t(i,j)] \right)^2 \right]$$

$$\leq 5\beta^{-1} \sum_{t=1}^{T} \underset{\substack{i \sim \tilde{\mu}_t \\ j \sim \nu_t}}{\mathbb{E}} \left[ (b_t(i,j))^2 \right].$$

Similarly w.p. $1 - \delta/3$ (Under event $\mathcal{E}_1$) using the same arguments as above for the min player we have

$$T_3 + T_4 \leq 5\beta^{-1} \sum_{t=1}^{T} \underset{\substack{i \sim \mu_t \\ j \sim \tilde{\nu}_t}}{\mathbb{E}} \left[ (b_t(i,j))^2 \right].$$

Define

$$\Sigma_t^+ := \lambda \mathbf{I} + \sum_{(i,j) \in \mathcal{D}_{t-1}^+} \phi(i,j)\phi(i,j)^\top \quad \text{and} \quad \Sigma_t^- = \lambda \mathbf{I} + \sum_{(i,j) \in \mathcal{D}_{t-1}^-} \phi(i,j)\phi(i,j)^\top. \tag{59}$$

By defining the filtration $\mathcal{F}_{t-1} = \sigma \left( \left\{ (i_l^+, j_l^+, \widehat{A}(i_l^+, j_l^+)), (i_l^-, j_l^-, \widehat{A}(i_l^-, j_l^-)) \right\}_{l=1}^{t-1} \right)$, we observe that random variables $\left\| \phi\left(i_t^+, j_t^+\right) \right\|_{(\Sigma_t^+)^{-1}}^2$ and $\left\| \phi\left(i_t^-, j_t^-\right) \right\|_{(\Sigma_t^-)^{-1}}^2$ are $\mathcal{F}_t$-measurable, while the policies $\tilde{\mu}_t, \mu_t, \tilde{\nu}_t$ and $\nu_t$ are $\mathcal{F}_{t-1}$ measurable. Define the events

$$\mathcal{E}_2 = \left\{ \sum_{t=1}^{T} \underset{\substack{i \sim \tilde{\mu}_t \\ j \sim \nu_t}}{\mathbb{E}} \|\phi(i,j)\|_{(\Sigma_t^+)^{-1}}^2 \leq 2 \sum_{t=1}^{T} \|\phi(i_t^+, j_t^+)\|_{(\Sigma_t^+)^{-1}}^2 + 8 \log \left( \frac{12}{\delta} \right) \right\},$$

$$\mathcal{E}_3 = \left\{ \sum_{t=1}^{T} \underset{\substack{i \sim \mu_t \\ j \sim \tilde{\nu}_t}}{\mathbb{E}} \|\phi(i,j)\|_{(\Sigma_t^-)^{-1}}^2 \leq 2 \sum_{t=1}^{T} \|\phi(i_t^-, j_t^-)\|_{(\Sigma_t^-)^{-1}}^2 + 8 \log \left( \frac{12}{\delta} \right) \right\}.$$

Choosing $\lambda = 1$ and using Lemma D.2 with $R = 1$ (since $\|\phi(i,j)\|_{(\Sigma_t^-)^{-1}}^2 \leq 1 \, \forall \, (i,j) \in [m] \times [n]$ from assumption 2.1), we have $\mathbb{P}(\mathcal{E}_2) \geq 1 - \delta/6$ and $\mathbb{P}(\mathcal{E}_3) \geq 1 - \delta/6$. Thus from (36), under the event $\mathcal{E}_{\text{dep}}^{\text{matrix}} := \mathcal{E}_1 \cap \mathcal{E}_2 \cap \mathcal{E}_3$ we have the dual gap bounded as

$$\text{Regret(T)} = \sum_{t=1}^{T} \left( f^{\star, \nu_t}(A) - f^{\mu_t, \star}(A) \right)$$

$$= 5\beta^{-1} \sum_{t=1}^{T} \underset{\substack{i \sim \tilde{\mu}_t \\ j \sim \nu_t}}{\mathbb{E}} \left[ (b_t(i,j))^2 \right] + 5\beta^{-1} \sum_{t=1}^{T} \underset{\substack{i \sim \mu_t \\ j \sim \tilde{\nu}_t}}{\mathbb{E}} \left[ (b_t(i,j))^2 \right]$$

$$= 5\beta^{-1} \eta_T^2 \sum_{t=1}^{T} \left( \underset{\substack{i \sim \tilde{\mu}_t \\ j \sim \nu_t}}{\mathbb{E}} \|\phi(i,j)\|_{\Sigma_t^{-1}}^2 + \underset{\substack{i \sim \mu_t \\ j \sim \tilde{\nu}_t}}{\mathbb{E}} \|\phi(i,j)\|_{\Sigma_t^{-1}}^2 \right)$$

$$\leq 5\beta^{-1} \eta_T^2 \sum_{t=1}^{T} \left( \underset{\substack{i \sim \tilde{\mu}_t \\ j \sim \nu_t}}{\mathbb{E}} \|\phi(i,j)\|_{(\Sigma_t^+)^{-1}}^2 + \underset{\substack{i \sim \mu_t \\ j \sim \tilde{\nu}_t}}{\mathbb{E}} \|\phi(i,j)\|_{(\Sigma_t^-)^{-1}}^2 \right)$$

$$\leq 10\beta^{-1} \eta_T^2 \left( \sum_{t=1}^{T} \left( \|\phi(i_t^+, j_t^+)\|_{(\Sigma_t^+)^{-1}}^2 + \|\phi(i_t^-, j_t^-)\|_{(\Sigma_t^-)^{-1}}^2 \right) + 8 \log(12\delta^{-1}) \right)$$

$$= \mathcal{O} \left( \beta^{-1} \left( 1 + \sigma \sqrt{\log \left( \frac{T}{\delta} \right)} + \sigma^2 \log \left( \frac{T}{\delta} \right) \right) d^2 \log \left( \frac{T}{d} \right) \right), \tag{60}$$

where the third line follows from the fact $\Sigma_t^+ \preceq \Sigma_t$ and $\Sigma_t^- \preceq \Sigma_t$, the penultimate line comes from event $\mathcal{E}_3 \cap \mathcal{E}_3$. Where $\lambda = 1$ and we use the elliptical potential lemma (Lemma D.6) to obtain the last line.

## E.2. Proof of Theorem E.3: Regularization-Independent Bound

Using eq. (36) we have $\mathrm{Regret}(T) = T_1 + T_2 + T_3 + T_4$ and $T_3 + T_4$ can be bound similar to $T_1 + T_2$. Let $\mu_t^\dagger$ be the best response to $\nu_t$ under $A$ (c.f. (37)). We bound $T_1$ using UCB style analysis, under the event $\mathcal{E}_1$, as follows:

$$T_1 = \sum_{t=1}^{T} (f^{\mu_t^\dagger, \nu_t}(A) - f^{\tilde{\mu}_t, \nu_t}(A)) \leq \sum_{t=1}^{T} (f^{\mu_t^\dagger, \nu_t}(A_t^+) - f^{\tilde{\mu}_t, \nu_t}(A)) \tag{61}$$

$$\leq \sum_{t=1}^{T} (f^{\tilde{\mu}_t, \nu_t}(A_t^+) - f^{\tilde{\mu}_t, \nu_t}(A)) = \sum_{t=1}^{T} \mathop{\mathbb{E}}_{\substack{i \sim \tilde{\mu}_t \\ j \sim \nu_t}} [A_t^+(i,j) - A(i,j)] \tag{62}$$

$$\leq 2 \sum_{t=1}^{T} \mathop{\mathbb{E}}_{\substack{i \sim \tilde{\mu}_t \\ j \sim \nu_t}} [b_t(i,j)] = 2 \sum_{t=1}^{T} \eta_T \mathop{\mathbb{E}}_{\substack{i \sim \tilde{\mu}_t \\ j \sim \nu_t}} [\|\phi(i,j)\|_{\Sigma_t^{-1}}] \leq 2 \sum_{t=1}^{T} \eta_T \mathop{\mathbb{E}}_{\substack{i \sim \tilde{\mu}_t \\ j \sim \nu_t}} \|\phi(i,j)\|_{(\Sigma_t^+)^{-1}}. \tag{63}$$

Eq. (61) and the first inequality in (63) follow from the Proposition E.1. Here (62) follows since $\tilde{\mu}_t = \arg\max_{\mu} f^{\mu, \nu_t}(A_t^+)$. The second inequality in eq. (63) comes from the fact $\Sigma_t^+ \preceq \Sigma_t$. Similarly, under the event $\mathcal{E}_1$, we can bound $T_2$ as follows

$$T_2 = \sum_{t=1}^{T} (f^{\tilde{\mu}_t, \nu_t}(A) - f^{\mu_t, \nu_t}(A))$$

$$\leq \sum_{t=1}^{T} (f^{\tilde{\mu}_t, \nu_t}(A) - f^{\tilde{\mu}_t, \nu_t}(\overline{A}_t)) + \sum_{t=1}^{T} (f^{\mu_t, \nu_t}(\overline{A}_t) - f^{\mu_t, \nu_t}(A)) \tag{64}$$

$$\leq \sum_{t=1}^{T} \mathop{\mathbb{E}}_{\substack{i \sim \tilde{\mu}_t \\ j \sim \nu_t}} [b_t(i,j)] + \sum_{t=1}^{T} \mathop{\mathbb{E}}_{\substack{i \sim \mu_t \\ j \sim \nu_t}} [b_t(i,j)] \tag{65}$$

$$\leq 2 \sum_{t=1}^{T} \mathop{\mathbb{E}}_{\substack{i \sim \tilde{\mu}_t \\ j \sim \nu_t}} [b_t(i,j)] = 2 \sum_{t=1}^{T} \eta_T \mathop{\mathbb{E}}_{\substack{i \sim \tilde{\mu}_t \\ j \sim \nu_t}} [\|\phi(i,j)\|_{\Sigma_t^{-1}}] \tag{66}$$

$$\leq 2\eta_T \sum_{t=1}^{T} \mathop{\mathbb{E}}_{\substack{i \sim \tilde{\mu}_t \\ j \sim \nu_t}} \|\phi(i,j)\|_{(\Sigma_t^+)^{-1}}, \tag{67}$$

where (64) follows from the fact that $\mu_t = \arg\max_{\mu} f^{\mu, \nu_t}(\overline{A}_t)$, (65) follows from Proposition E.1, (66) follows since $f^{\mu_t, \nu_t}(\overline{A}_t) \geq f^{\tilde{\mu}_t, \nu_t}(\overline{A}_t)$ and

$$f^{\mu_t, \nu_t}(\overline{A}_t) + \mathop{\mathbb{E}}_{\substack{i \sim \mu_t \\ j \sim \nu_t}} [b_t(i,j)] = f^{\mu_t, \nu_t}(A_t^+) \leq f^{\tilde{\mu}_t, \nu_t}(A_t^+) = f^{\tilde{\mu}_t, \nu_t}(\overline{A}_t) + \mathop{\mathbb{E}}_{\substack{i \sim \tilde{\mu}_t \\ j \sim \nu_t}} [b_t(i,j)],$$

and (67) follows from the fact $\Sigma_t^+ \preceq \Sigma_t$. Define the filtration

$$\mathcal{F}_{t-1} = \sigma \left( \left\{ (i_l^+, j_l^+, \widehat{A}(i_l^+, j_l^+)), (i_l^-, j_l^-, \widehat{A}(i_l^-, j_l^-)) \right\}_{l=1}^{t-1} \right).$$

We have random variable $\|\phi(i_t^+, j_t^+)\|_{(\Sigma_t^+)^{-1}}$ is $\mathcal{F}_t$-measurable, while the policies $\tilde{\mu}_t, \mu_t, \tilde{\nu}_t$ and $\nu_t$ are $\mathcal{F}_{t-1}$ measurable. Define the events

$$\mathcal{E}_4 = \left\{ \sum_{t=1}^{T} \mathop{\mathbb{E}}_{\substack{i \sim \tilde{\mu}_t \\ j \sim \nu_t}} \left[ \|\phi(i,j)\|_{(\Sigma_t^+)^{-1}} \right] \leq 2 \sum_{t=1}^{T} \|\phi(i_t^+, j_t^+)\|_{(\Sigma_t^+)^{-1}} + 8 \log \left( \frac{12}{\delta} \right) \right\},$$

$$\mathcal{E}_5 = \left\{ \sum_{t=1}^{T} \mathop{\mathbb{E}}_{\substack{i \sim \mu_t \\ j \sim \tilde{\nu}_t}} \left[ \|\phi(i,j)\|_{(\Sigma_t^-)^{-1}} \right] \leq 2 \sum_{t=1}^{T} \|\phi(i_t^-, j_t^-)\|_{(\Sigma_t^-)^{-1}} + 8 \log \left( \frac{12}{\delta} \right) \right\}.$$

Choosing $\lambda = 1$ we have $\mathbb{P}(\mathcal{E}_4) \geq 1 - \delta/6$ and $\mathbb{P}(\mathcal{E}_5) \geq 1 - \delta/6$ using Lemma D.2 with $R = 1$ (since $\|\phi(i,j)\|_{(\Sigma_t^-)^{-1}} \leq 1$ $\forall\, (i,j) \in [m] \times [n]$ from assumption 2.1). Under the event $\mathcal{E}_1 \cap \mathcal{E}_4$, using equations (63) and (67) we have

$$T_1 + T_2 \leq 4\eta_T \sum_{t=1}^{T} \mathop{\mathbb{E}}_{\substack{i \sim \tilde{\mu}_t \\ j \sim \nu_t}} \left[ \|\phi(i,j)\|_{(\Sigma_t^+)^{-1}} \right]$$

$$\leq 8\eta_T \left( \sum_{t=1}^{T} \|\phi(i_t^+, j_t^+)\|_{(\Sigma_t^+)^{-1}} + 4 \log \left( \frac{12}{\delta} \right) \right) \tag{68}$$

$$\leq 8\eta_T \left( \sqrt{T \sum_{t=1}^{T} \|\phi(i_t^+, j_t^+)\|_{(\Sigma_t^+)^{-1}}^2} + 4 \log \left( \frac{12}{\delta} \right) \right) = \mathcal{O}\left( (1+\sigma)d\sqrt{T} \log \left( \frac{T}{\delta} \right) \right). \tag{69}$$

The equations (68) and (69) follow from event $\mathcal{E}_4$ and Lemma D.6 (elliptical potential lemma) respectively. Similarly one can bound $T_3 + T_4$ under the event $\mathcal{E}_1 \cap \mathcal{E}_5$ by $\mathcal{O}\left( \sigma d\sqrt{T} \log \left( \frac{T}{\delta} \right) \right)$. Thus under the event $\mathcal{E}_{\text{ind}}^{\text{matrix}} := \mathcal{E}_1 \cap \mathcal{E}_4 \cap \mathcal{E}_5$, we have

$$\text{Regret}(T) \leq \mathcal{O}\left( (1+\sigma)d\sqrt{T} \log \left( \frac{T}{\delta} \right) \right). \tag{70}$$

Finally under the event $\mathcal{E}^{\text{matrix}} = \mathcal{E}_{\text{dep}}^{\text{matrix}} \cap \mathcal{E}_{\text{ind}}^{\text{matrix}} = \mathcal{E}_1 \cap \mathcal{E}_2 \cap \mathcal{E}_3 \cap \mathcal{E}_4 \cap \mathcal{E}_5$ (w.p. atleast $1 - \delta$) equations eqs. (60) and (70) hold simultaneously which completes the proof of Theorem 2.3.

### E.3. Auxiliary Lemmas

**Lemma E.4.** *The partial derivative $\frac{\partial \Delta(A', \nu')}{\partial A'(i:)\nu'}$ is given by*

$$\frac{\partial \Delta(A', \nu')}{\partial A'(i,:)\nu'}$$
$$= \beta^{-1} \mu(A', \nu')_i (A'(i,:) - A(i,:))\nu' - \beta^{-1} \mu(A', \nu')_i \sum_{i'} \mu(A', \nu')_{i'} (A'(i',:) - A(i',:))\nu'$$
$$= \beta^{-1} \mu(A', \nu')_i \left[ (A'(i,:) - A(i,:))\nu' - \mathop{\mathbb{E}}_{i' \sim \mu(A',\nu')} [(A'(i',:) - A(i',:))\nu'] \right]. \tag{71}$$

*Proof.* The symbol $\frac{\partial}{\partial A'(i,:)\nu'}$ denotes differentiation with respect to the *scalar* quantity $A'(i,:)\nu'$. Throughout this differentiation we regard the vector $\nu'$ as constant, and keep every row of $A'$ *except* the $i^{\text{th}}$ row fixed. Because the other rows are held fixed, the cross-derivatives vanish: $\frac{\partial A'(i',:)\nu'}{\partial A'(i,:)\nu'} = 0$, $\forall i' \neq i$, so each row contributes an independent gradient term.

$$\frac{\partial \Delta(A', \nu')}{\partial A'(i,:)\nu'} = \frac{\partial \left[ -\beta \log(Z(A', \nu')) + \mu(A', \nu')(A' - A)\nu' \right]}{\partial A'(i,:)\nu'}$$
$$= -\frac{\beta}{Z(A', \nu')} \frac{\partial Z(A', \nu')}{\partial A'(i,:)\nu'} + [\mu(A', \nu')]_i + \frac{\partial ([\mu(A', \nu')]_i)}{\partial A'(i,:)\nu'} (A'(i,:) - A(i,:))\nu'$$
$$+ \sum_{i' \neq i} \frac{\partial [\mu(A', \nu')]_{i'}}{\partial A'(i,:)\nu'} (A'(i',:) - A(i',:))\nu'. \tag{72}$$

We have

$$\frac{\partial Z(A', \nu')}{\partial (A'(i,:)\nu')} = \mu_{\text{ref},i} \exp\left( \frac{A'(i,:)\nu'}{\beta} \right) \frac{1}{\beta} = \frac{Z(A', \nu')}{\beta} [\mu(A', \nu')]_i,$$

$$\frac{\partial\left([\mu(A',\nu')]_i\right)}{\partial A'(i,:)\nu'} = \frac{\partial\left(\mu_{\text{ref},i}\,\exp\left(A'(i,:)\nu'/\beta\right)/Z(A',\nu')\right)}{\partial A'(i,:)\nu'}$$

$$= \frac{\beta^{-1}\left(\mu_{\text{ref},i}\,\exp\left(A'(i,:)\nu'/\beta\right)Z(A',\nu') - \left(\mu_{\text{ref},i}\,\exp\left(A'(i,:)\nu'/\beta\right)^2\right)\right)}{Z(A',\nu')^2}$$

$$= \beta^{-1}\left([\mu(A',\nu')]_i - [\mu(A',\nu')]_i^2\right),$$

$$\frac{\partial\left([\mu(A',\nu')]_{i'}\right)}{\partial A'(i,:)\nu'} = \frac{\partial\left(\mu_{\text{ref},i'}\,\exp\left(A'(i',:)\nu'/\beta\right)/Z(A',\nu')\right)}{\partial A'(i,:)\nu'}$$

$$= \frac{-\beta^{-1}\left(\mu_{\text{ref},i}\,\exp\left(A'(i,:)\nu'/\beta\right)\mu_{\text{ref},i'}\,\exp\left(A'(i',:)\nu'/\beta\right)\right)}{Z(A',\nu')^2}$$

$$= -\beta^{-1}[\mu(A',\nu')]_i[\mu(A',\nu')]_{i'}.$$

Substituting back in eq. (72) we get the desired result. $\qquad\square$

## F. Markov Game Proofs

**Notation and Convention.** For any function $f : \mathcal{S} \to \mathbb{R}$ we define $P_h f(s,i,j) := \mathbb{E}_{s' \sim P_h(\cdot|s,i,j)}[f(s')]$. We also use the notation

$$\underset{s_{h+1}|s_h,i_h,j_h}{\mathbb{E}}(f(s_{h+1})) := \mathbb{E}_{s_{h+1}\sim P_h(\cdot|s_h,i_h,j_h)}[f(s_{h+1})] = P_h f(s_h,i_h,j_h).$$

For all $K > H$ and $(s,i,j) \in \mathcal{S} \times \mathcal{U} \times \mathcal{V}$, we set $\widehat{Q}_K(s,i,j) = 0$, $\widehat{V}_K(s) = 0$, $\text{KL}(\widehat{\mu}_{H+1}(\cdot|s)\|\mu_{\text{ref},K}(\cdot|s)) = 0$, and $\text{KL}(\widehat{\nu}_K(\cdot|s)\|\nu_{\text{ref},K}(\cdot|s)) = 0$. These conventions apply to every value function $\widehat{V}$, every $Q$-function $\widehat{Q}$ (both estimates and true values), and all feasible policies $\widehat{\mu}$ and $\widehat{\nu}$.

**Proposition F.1.** *The closed-form expressions of the best response to min-player strategy $\nu'$ under for a $Q$ function $Q'_h(s,i,j) \,\forall(s,i,j) \in \mathcal{S} \times \mathcal{U} \times \mathcal{V}, h \in [H]$ denoted by $\mu(Q',\nu')$ where $Q' := \{Q'_h\}_{h=1}^H$ is given by*

$$[\mu_{h,t}(Q',\nu')](i|s) = \frac{\mu_{\text{ref},h}(i|s)\exp\left(\mathbb{E}_{j\sim\nu'_h(\cdot|s)}[Q'(s,i,j)/\beta]\right)}{\sum_{i'\in\mathcal{U}}\mu_{\text{ref},h}(i'|s)\exp\left(\mathbb{E}_{j\sim\nu'_h(\cdot|s)}[Q'(s,i',j)/\beta]\right)}$$

*and we have $\mu_t = \mu(\overline{Q}_t,\nu_t)$, $\tilde{\mu}_t = \mu(Q_t^+,\nu_t)$ and $\mu_t^\dagger = \mu(Q^{\mu_t^\dagger,\nu_t},\nu_t)$*

*Proof.* The result is an immediate consequence of the definitions and routine calculations. $\qquad\square$

Now in order to prove our main result we note that Theorem 3.3 holds as long as for any $\delta \in [0,1]$, Theorems F.2 and F.3 can be established.

**Theorem F.2** (Regularization-dependent guarantee). *Under Assumption 3.1, for any fixed $\delta \in [0,1]$ and any $\beta > 0$, reference policies $(\mu_{\text{ref}},\nu_{\text{ref}}) = (\{\mu_{\text{ref},h}(\cdot|\cdot)\}_{h=1}^H, \{\nu_{\text{ref},h}(\cdot|\cdot)\}_{h=1}^H)$, choosing $\lambda = 1$ and $b_{h,t}^{sup}(s,i,j)$ as per eq. (23) in Algorithm 2, we have*

$$\forall\, T \in \mathbb{N}^+ : \qquad \text{Regret}(T) \leq \mathcal{O}\left(\beta^{-1}d^3H^7\log\left(\frac{dT}{\delta}\right)\log\left(\frac{dTH}{\delta\min\{1,\beta\}}\right)\right) \quad \text{w.p. } 1 - \delta/2.$$

**Theorem F.3** (Regularization-independent guarantee). *Under Assumption 3.1, for any fixed $\delta \in [0,1]$ and any $\beta \geq 0$, reference policies $(\mu_{\text{ref}},\nu_{\text{ref}}) = (\{\mu_{\text{ref},h}(\cdot|\cdot)\}_{h=1}^H, \{\nu_{\text{ref},h}(\cdot|\cdot)\}_{h=1}^H)$, choosing $\lambda = 1$ and $b_{h,t}^{sup}(s,i,j)$ as per eq. (23) in Algorithm 2, we have*

$$\forall\, T \in \mathbb{N}^+ : \qquad \text{Regret}(T) \leq \mathcal{O}\left(d^{3/2}H^3\sqrt{T}\sqrt{\log\left(\frac{dT}{\delta}\right)\log\left(\frac{dTH}{\delta\min\{1,\beta\}}\right)\cdot}\right) \quad \text{w.p. } 1 - \delta/2.$$

## F.1. Supporting Lemmas

We begin by introducing some lemmas that will be used in proving the main result. The proofs of these lemmas are deferred to Section F.4.

In Lemmas F.4, F.5 and Corollary F.6 we introduce high probability concentration events and Bellman error bounds used in proving our main results.

**Lemma F.4** (Concentration of MSE Bellman errors). *Define the Bellman error of the MSE Q function as*

$$\overline{e}_{h,t}(s,i,j) := \overline{Q}_{h,t}(s,i,j) - r_h(s,i,j) - P_h \overline{V}_{h+1}(s,i,j). \tag{73}$$

*Then under the setting in Algorithm 2, choosing $\lambda = 1$, $\forall (s,i,j) \in \mathcal{S} \times \mathcal{U} \times \mathcal{V}, h \in [H], \forall t \in [T]$ the event*

$$\mathcal{E}_6 := \left\{ |\overline{e}_{h,t}(s,i,j)| \leq \eta_1 \|\phi(s,i,j)\|_{\Sigma_{h,t}^{-1}} := b_{h,t}^{mse}(s,i,j) \right\} \tag{74}$$

*occurs with probability at least $1 - \delta/16$. Here $\eta_1 := c_1 \sqrt{d} H \sqrt{\log\left(\frac{16HT}{\delta}\right)}$, where $c_1 > 0$ is a universal constant.*

**Lemma F.5** (Concentration of Superoptimistic Bellman errors). *Under the setting in Algorithm 2, choosing $\lambda = 1$, $\forall (s,i,j) \in \mathcal{S} \times \mathcal{U} \times \mathcal{V}, h \in [H], \forall t \in [T]$ the event*

$$\mathcal{E}_7 := \left\{ \left| \left\langle \theta_{h,t}^+, \phi(s,i,j) \right\rangle - r_h(s,i,j) - P_h V_{h+1}^+(s,i,j) \right| \leq \eta_2 \|\phi(s,i,j)\|_{\Sigma_{h,t}^{-1}} := b_{h,t}(s,i,j) \right\}$$

*occurs with probability $1 - \delta/16$. Here $\eta_2 = c_2 d H^2 \sqrt{\log\left(\frac{16dTH}{\delta \min\{1,\beta\}}\right)}$ and $c_2 > 0$ is a universal constant.*

**Corollary F.6** (Bounds on Superoptimistic Bellman error w.r.t. the $Q^+$ function). *Let*

$$e_{h,t}^+(s,i,j) := Q_{h,t}^+(s,i,j) - r_h(s,i,j) - P_h V_{h+1}^+(s,i,j),$$

*then under the event $\mathcal{E}_7$, for $b_{h,t}^{sup}(s,i,j) := b_{h,t}(s,i,j) + 2b_{h,t}^{mse}(s,i,j)$, we have*

$$\left| e_{h,t}^+(s,i,j) \right| \leq 2b_{h,t}(s,i,j) + 2b_{h,t}^{mse}(s,i,j) = b_{h,t}^{sup}(s,i,j) + b_{h,t}(s,i,j).$$

For notational simplicity, while stating the next two lemmas we will omit the superscript $\nu_t$ and also the dependence on $t$. Both lemmas are valid for all $t \in [T]$. Consequently, the symbols $\mu$ and $\tilde{\mu}$ in Lemma F.7 and Lemma F.8 should be interpreted as the time-indexed policies $\mu_t$ and $\tilde{\mu}_t$, rather than an arbitrary policies.

Lemma F.7 formalizes the notion of optimism for Algorithm 2.

**Lemma F.7** (Optimism). *For the setting in Algorithm 2, under the event $\mathcal{E}_6 \cap \mathcal{E}_7$, $\forall (s_h, i_h, j_h) \in \mathcal{S} \times \mathcal{U} \times \mathcal{V}, h \in [H+1]$ and any policy $\mu'$ of the max player, we have the following equations hold:*

$$Q_h^+(s_h, i_h, j_h) \geq \overline{Q}_h(s_h, i_h, j_h), \tag{75a}$$

$$Q_h^+(s_h, i_h, j_h) \geq Q_h^{\mu'}(s_h, i_h, j_h). \tag{75b}$$

The next lemma introduces the concept of the superoptimistic gap, arising from the construction of the superoptimistic bonus term and the projection operators.

**Lemma F.8** (Super-optimistic gap). *For the setting in Algorithm 2, under the event $\mathcal{E}_6 \cap \mathcal{E}_7$, $\forall (s_h, i_h, j_h) \in \mathcal{S} \times \mathcal{U} \times \mathcal{V}, h \in [H+1]$, we have*

$$2 \left| \left( Q_h^+(s_h, i_h, j_h) - \overline{Q}_h(s_h, i_h, j_h) \right) \right| \geq \left| Q_h^+(s_h, i_h, j_h) - Q_h^\mu(s_h, i_h, j_h) \right|. \tag{76}$$

Note that this is the exact condition used in the matrix games section that we use to bound the term $T_2$ using an expectation of some function over actions sampled using the best-response policy $\tilde{\mu}$ using the first bounding method (52).

### F.2. Proof of Theorem F.2: Logarithmic Regret Bound

For simplicity we fix the initial state to $s_1$, extending the arguments to a fixed initial distribution $s_1 \sim \rho$ is trivial. One step regret is given by

$$
\begin{aligned}
\mathsf{DualGap}(\mu_t, \nu_t) &= V_1^{\star,\nu_t}(s_1) - V_1^{\mu_t,\star}(s_1) \\
&= \underbrace{V_1^{\star,\nu_t}(s_1) - V_1^{\tilde{\mu}_t,\nu_t}(s_1)}_{T_5^{(t)}} + \underbrace{V_1^{\tilde{\mu}_t,\nu_t}(s_1) - V_1^{\mu_t,\nu_t}(s_1)}_{T_6^{(t)}} \\
&\quad + \underbrace{V_1^{\mu_t,\nu_t}(s_1) - V_1^{\mu_t,\tilde{\nu}_t}(s_1)}_{T_7^{(t)}} + \underbrace{V_1^{\mu_t,\tilde{\nu}_t}(s_1) - V_1^{\mu_t,\star}(s_1)}_{T_8^{(t)}}.
\end{aligned}
\tag{77}
$$

Below we bound $T_5^{(t)}$ and $T_6^{(t)}$, and the remaining two terms can be bounded similarly.

**Step 1: Bounding $T_5^{(t)}$.** For notational simplicity we will omit the superscript $\nu_t$ here as we try to bound both $T_5$ and $T_6$. Given a fixed strategy of the minimizing player one can treat the best response computation objective as a RL policy optimization. Let $\mu_t^{\dagger}$ denote the best response to $\tilde{\nu}_t$ at $t$. We will use the following *leafing* here inspired from (Zhao et al., 2025b). Let $\mu^{(h)} := \tilde{\mu}_{1:h} \oplus \mu_{h+1:H}^{\dagger}$ denote the concatenated policy that plays $\tilde{\mu}$ for the first $h$ steps and then executes $\mu^{\dagger}$ for the remaining steps. Again we drop the subscript $t$ here for notational simplicity. Consider the term

$$
\begin{aligned}
T_5 &= V_1^{\mu^{\dagger}}(s_1) - V_1^{\tilde{\mu}}(s_1) \\
&= \sum_{h=0}^{H-1} \underbrace{V_1^{\mu^{(h)}}(s_1) - V_1^{\mu^{(h+1)}}(s_1)}_{I_{h+1}}.
\end{aligned}
$$

For any policy pair $(\mu', \nu')$, $h \in [H]$, let $d_h^{\mu',\nu'}$ denote the state distribution induced at step $h$ when following the policy $(\mu', \nu')$. Under the event $\mathcal{E}_6 \cap \mathcal{E}_7$, we can bound each $I_{h+1}$ as follows

$$
\begin{aligned}
I_{h+1} &= \mathbb{E}_{s_{h+1} \sim d_{h+1}^{\tilde{\mu},\nu}} \left[ V_{h+1}^{\mu^{(h)}}(s_{h+1}) - V_{h+1}^{\mu^{(h+1)}}(s_{h+1}) \right] \\
&= \mathbb{E}_{\substack{s_{h+1} \sim d_{h+1}^{\tilde{\mu},\nu} \\ }} \mathbb{E}_{\substack{i_{h+1} \sim \mu_{h+1}^{\dagger}(\cdot|s_{h+1}) \\ j_{h+1} \sim \nu_{h+1}(\cdot|s_{h+1})}} \left[ Q_{h+1}^{\mu^{\dagger}}(s_{h+1}, i_{h+1}, j_{h+1}) - \beta\mathrm{KL}(\mu_{h+1}^{\dagger}(\cdot|s_{h+1}) \| \mu_{\mathrm{ref},h+1}(\cdot|s_{h+1})) \right] \\
&\quad - \mathbb{E}_{\substack{s_{h+1} \sim d_{h+1}^{\tilde{\mu},\nu} \\ }} \mathbb{E}_{\substack{i_{h+1} \sim \tilde{\mu}_{h+1}(\cdot|s_{h+1}) \\ j_{h+1} \sim \nu_{h+1}(\cdot|s_{h+1})}} \left[ Q_{h+1}^{\mu^{\dagger}}(s_{h+1}, i_{h+1}, j_{h+1}) - \beta\mathrm{KL}(\tilde{\mu}_{h+1}(\cdot|s_{h+1}) \| \mu_{\mathrm{ref},h+1}(\cdot|s_{h+1})) \right] \\
\end{aligned}
\tag{78}
$$

$$
\leq \beta^{-1} \mathbb{E}_{\substack{s_{h+1} \sim d_{h+1}^{\tilde{\mu},\nu}}} \mathbb{E}_{i_{h+1} \sim \tilde{\mu}_{h+1}(\cdot|s_{h+1})} \left[ \left( \mathbb{E}_{j_{h+1} \sim \nu_{h+1}(\cdot|s_{h+1})} \left[ Q_{h+1}^{+}(s_{h+1}, i_{h+1}, j_{h+1}) - Q_{h+1}^{\mu^{\dagger}}(s_{h+1}, i_{h+1}, j_{h+1}) \right] \right)^2 \right]
\tag{79}
$$

$$
\leq \beta^{-1} \mathbb{E}_{\substack{s_{h+1} \sim d_{h+1}^{\tilde{\mu},\nu}}} \mathbb{E}_{\substack{i_{h+1} \sim \tilde{\mu}_{h+1}(\cdot|s_{h+1}) \\ j_{h+1} \sim \nu_{h+1}(\cdot|s_{h+1})}} \left[ \left( Q_{h+1}^{+}(s_{h+1}, i_{h+1}, j_{h+1}) - Q_{h+1}^{\mu^{\dagger}}(s_{h+1}, i_{h+1}, j_{h+1}) \right)^2 \right].
$$

Note that here (78) follows from the fact $Q_{h+1}^{\mu^{(h)}}(s,i,j) = Q_{h+1}^{\mu^{(h+1)}}(s,i,j) = r_{h+1}(s,i,j) + P_{h+1}V_{h+2}^{\mu^{\dagger}}(s,i,j) = Q_{h+1}^{\mu^{\dagger}}(s,i,j) \ \forall (s,i,j) \in \mathcal{S} \times \mathcal{U} \times \mathcal{V}, h \in [H]$. Eq. (79) comes from (for $Q_{h+1}^{+}(s_{h+1}, i_{h+1}, j_{h+1}) \geq Q_{h+1}^{\mu^{\dagger}}(s_{h+1}, i_{h+1}, j_{h+1})$) Lemma F.7 and the same analysis used for bounding the term $T_1$ (see eqs. (38)-(46)). Here $Q_{h+1}^{\mu^{\dagger}}(s_{h+1}, \cdot, \cdot)$ will be mapped to $A(\cdot, \cdot)$ and $Q_{h+1}^{+}(s_{h+1}, \cdot, \cdot)$ to $A^{+}(\cdot, \cdot)$ from the matrix games section. Let $a_{h+1} = (i_{h+1}, j_{h+1})$, now using Lemma F.7 we have

$$
\begin{aligned}
0 &\leq Q_{h+1}^{+}(s_{h+1}, i_{h+1}, j_{h+1}) - Q_{h+1}^{\mu^{\dagger}}(s_{h+1}, i_{h+1}, j_{h+1}) \\
&= \mathbb{E}_{s_{h+2}|s_{h+1},a_{h+1}} \left( V_{h+2}^{+}(s_{h+2}) - V_{h+2}^{\mu^{\dagger}}(s_{h+2}) \right) + e_{h+1}^{+}(s_{h+1}, i_{h+1}, j_{h+1})
\end{aligned}
$$

$$
\begin{aligned}
=\ & \mathop{\mathbb{E}}_{s_{h+2}|s_{h+1},a_{h+1}} \mathop{\mathbb{E}}_{\substack{i_{h+2}\sim\tilde{\mu}_{h+2}(\cdot|s_{h+2})\\ j_{h+2}\sim\nu_{h+2}(\cdot|s_{h+2})}} \Big(Q_{h+2}^{+}(s_{h+2},i_{h+2},j_{h+2}) - \beta\mathrm{KL}(\tilde{\mu}_{h+2}(\cdot|s_{h+2})\|\mu_{\mathrm{ref},h+2}(\cdot|s_{h+2})) \\
& \quad + \beta\mathrm{KL}(\nu_{h+2}(\cdot|s_{h+2})\|\nu_{\mathrm{ref},h+2}(\cdot|s_{h+2}))\Big) \\
-\ & \mathop{\mathbb{E}}_{s_{h+2}|s_{h+1},a_{h+1}} \mathop{\mathbb{E}}_{\substack{i_{h+2}\sim\mu_{h+2}^{\dagger}(\cdot|s_{h+2})\\ j_{h+2}\sim\nu_{h+2}(\cdot|s_{h+2})}} \Big(Q_{h+2}^{\mu^{\dagger}}(s_{h+2},i_{h+2},j_{h+2}) - \beta\mathrm{KL}(\mu_{h+2}^{\dagger}(\cdot|s_{h+2})\|\mu_{\mathrm{ref},h+2}(\cdot|s_{h+2})) \\
& \quad + \beta\mathrm{KL}(\nu_{h+2}(\cdot|s_{h+2})\|\nu_{\mathrm{ref},h+2}(\cdot|s_{h+2}))\Big) + e_{h+1}^{+}(s_{h+1},i_{h+1},j_{h+1}) \\
\leq\ & \mathop{\mathbb{E}}_{s_{h+2}|s_{h+1},a_{h+1}} \mathop{\mathbb{E}}_{\substack{i_{h+2}\sim\tilde{\mu}_{h+2}(\cdot|s_{h+2})\\ j_{h+2}\sim\nu_{h+2}(\cdot|s_{h+2})}} \Big[Q_{h+2}^{+}(s_{h+2},i_{h+2},j_{h+2}) - Q_{h+2}^{\mu^{\dagger}}(s_{h+2},i_{h+2},j_{h+2})\Big] + e_{h+1}^{+}(s_{h+1},i_{h+1},j_{h+1})
\end{aligned}
$$

$$(80)$$

$$
\begin{aligned}
\leq\ & \cdots \\
\leq\ & \mathbb{E}_{\cdot|s_{h+1},a_{h+1}}^{\tilde{\mu},\nu}\left[\sum_{k=h+1}^{H} e_{k}^{+}(s_{k},i_{k},j_{k})\right].
\end{aligned}
$$

Here $\mathbb{E}_{\cdot|s_{h+1},a_{h+1}}^{\tilde{\mu},\nu}$ denotes expectation with respect to the law of $s_k \sim \tilde{\mu},\nu|s_{h+1},a_{h+1}$, that is, the distribution of $s_k$ induced by policy $(\tilde{\mu},\nu)$ when starting from state $s_{h+1}$, taking action $a_{h+1}$ at step $h+1$, $i_k \sim \tilde{\mu}_k(\cdot|s_k)$ and $j_k \sim \nu_k(\cdot|s_k)$ for $k > h+1$. Here $e_h^{+}(s_h,i_h,j_h)$ is the Bellman error of the optimistic Q function and the Bellman error of $Q^{\mu^{\dagger}}(s_h,i_h,j_h) = r_h(s_h,i_h,j_h) + P_h V_{h+1}^{\mu^{\dagger}}(s_h,i_h,j_h)$ is zero. Eq. (80) follows by lower bounding the second term by swapping $\mu_{h+2}^{\dagger}(\cdot|s_{h+2})$ to the policy $\tilde{\mu}_{h+2}(\cdot|s_{h+2})$ since

$$
\mu_{h+2}^{\dagger}(\cdot|s_{h+2}) = \arg\max_{\mu'_{h+2}(\cdot|s_{h+2})} \mathop{\mathbb{E}}_{\substack{i_{h+2}\sim\mu'_{h+2}(\cdot|s_{h+2})\\ j_{h+2}\sim\nu_{h+2}(\cdot|s_{h+2})}} \Big(Q_{h+2}^{\mu^{\dagger}}(s_{h+2},i_{h+2},j_{h+2}) - \beta\mathrm{KL}(\mu'_{h+2}(\cdot|s_{h+2})\|\mu_{\mathrm{ref},h+2})\Big).
$$

Thus we have

$$
\begin{aligned}
I_{h+1} \leq\ & \beta^{-1} \mathop{\mathbb{E}}_{s_{h+1}\sim d_{h+1}^{\tilde{\mu},\nu}} \mathop{\mathbb{E}}_{\substack{i_{h+1}\sim\tilde{\mu}_{h+1}(\cdot|s_{h+1})\\ j_{h+1}\sim\nu_{h+1}(\cdot|s_{h+1})}} \left[\left(\mathbb{E}_{\cdot|s_{h+1},a_{h+1}}^{\tilde{\mu},\nu} \sum_{k=h+1}^{H} e_{k}^{+}(s_{k},i_{k},j_{k})\right)^{2}\right] \\
\leq\ & \beta^{-1}\mathbb{E}^{\tilde{\mu},\nu}\left[\left(\sum_{k=h+1}^{H} e_{k}^{+}(s_{k},i_{k},j_{k})\right)^{2}\right].
\end{aligned}
$$

Here $\mathbb{E}^{\tilde{\mu},\nu}$ is used to denote $s_k \sim d_k^{\tilde{\mu},\nu}$, $i_k \sim \tilde{\mu}_k(\cdot|s_k)$ and $j_k \sim \nu_k(\cdot|s_k)$. Thus we have

$$
T_5 = \sum_{h=0}^{H-1} I_{h+1} \leq \beta^{-1}\sum_{h=0}^{H-1}\mathbb{E}^{\tilde{\mu},\nu}\left[\left(\sum_{k=h+1}^{H} e_{k}^{+}(s_{k},i_{k},j_{k})\right)^{2}\right]. \tag{81}
$$

**Step 2: Bounding $T_6^{(t)}$.** Similar to bounding $T_5$ we *leaf* the policy in the following. Let $\mu^{(h)} = \tilde{\mu}_{1:h} \oplus \mu_{h+1:H}$, we have

$$
\begin{aligned}
T_6 &= V_1^{\tilde{\mu}}(s_1) - V_1^{\mu}(s_1) \\
&= \sum_{h=0}^{H-1} \underbrace{V_1^{\mu^{(H-h)}}(s_1) - V_1^{\mu^{(H-h-1)}}(s_1)}_{J_{H-h-1}}.
\end{aligned} \tag{82}
$$

We can write $J_h$ $(h = 0,\cdots,H-1)$ as follows

$$
J_h = \mathop{\mathbb{E}}_{s_{h+1}\sim d_{h+1}^{\tilde{\mu},\nu}} \left[V_{h+1}^{\mu^{(h+1)}}(s_{h+1}) - V_{h+1}^{\mu^{(h)}}(s_{h+1})\right]
$$

$$
\begin{aligned}
= \; & \mathop{\mathbb{E}}_{\substack{s_{h+1}\sim d_{h+1}^{\tilde{\mu},\nu}}} \mathop{\mathbb{E}}_{\substack{i_{h+1}\sim\tilde{\mu}_{h+1}(\cdot|s_{h+1}) \\ j_{h+1}\sim\nu_{h+1}(\cdot|s_{h+1})}} \left[ Q_{h+1}^{\mu}(s_{h+1},i_{h+1},j_{h+1}) - \beta\mathrm{KL}(\tilde{\mu}_{h+1}(\cdot|s_{h+1})\|\mu_{\mathrm{ref},h+1}(\cdot|s_{h+1})) \right] \\
- \; & \mathop{\mathbb{E}}_{\substack{s_{h+1}\sim d_{h+1}^{\tilde{\mu},\nu}}} \mathop{\mathbb{E}}_{\substack{i_{h+1}\sim\mu_{h+1}(\cdot|s_{h+1}) \\ j_{h+1}\sim\nu_{h+1}(\cdot|s_{h+1})}} \left[ Q_{h+1}^{\mu}(s_{h+1},i_{h+1},j_{h+1}) - \beta\mathrm{KL}(\mu_{h+1}(\cdot|s_{h+1})\|\mu_{\mathrm{ref},h+1}(\cdot|s_{h+1})) \right].
\end{aligned}
\tag{83}
$$

Note that here eq. (83) follows from the fact $Q_{h+1}^{\mu^{(h)}}(s,i,j) = Q_{h+1}^{\mu^{(h+1)}}(s,i,j) = r_{h+1}(s,i,j) + P_{h+1}V_{h+2}^{\mu}(s,i,j) = Q_{h+1}^{\mu}(s,i,j) \; \forall (s,i,j) \in \mathcal{S}\times\mathcal{U}\times\mathcal{V}, h\in[H]$. Now under the event $\mathcal{E}_6 \cap \mathcal{E}_7, \exists \; \Gamma \in [0,1]$ such that, for

$$
g_1(s_{h+1}) := \beta^{-1} \mathop{\mathbb{E}}_{i_{h+1}\sim\mu_{h+1}^{\Gamma}(\cdot|s_{h+1})} \left[ \left( \mathop{\mathbb{E}}_{j_{h+1}\sim\nu_{h+1}(\cdot|s_{h+1})} \left[ Q_{h+1}^{+}(s_{h+1},i_{h+1},j_{h+1}) - \overline{Q}_{h+1}(s_{h+1},i_{h+1},j_{h+1}) \right] \right)^2 \right],
$$

and

$$
g_2(s_{h+1}) := \beta^{-1} \mathop{\mathbb{E}}_{i_{h+1}\sim\mu_{h+1}^{\Gamma}(\cdot|s_{h+1})} \left[ \left( \mathop{\mathbb{E}}_{j_{h+1}\sim\nu_{h+1}(\cdot|s_{h+1})} \left[ Q_{h+1}^{+}(s_{h+1},i_{h+1},j_{h+1}) - Q_{h+1}^{\mu}(s_{h+1},i_{h+1},j_{h+1}) \right] \right)^2 \right].
$$

we have

$$
J_h \leq \mathop{\mathbb{E}}_{s_{h+1}\sim d_{h+1}^{\tilde{\mu},\nu}} \left[ g_1(s_{h+1}) + \max\{g_1(s_{h+1}), g_2(s_{h+1})\} \right].
\tag{84}
$$

Here eq.. (84) is obtained using the same arguments as the matrix games section, specifically the first way of bounding $T_2$ (see eqs.(48)-(52)). Here we can map eq. (83) to the eq. (48) specifically $Q_{h+1}^{\mu}(s_{h+1},\cdot,\cdot)$ can be mapped to $A(\cdot,\cdot)$, $Q_{h+1}^{+}(s_{h+1},\cdot,\cdot)$ to $A^{+}(\cdot,\cdot)$ and $\overline{Q}_{h+1}(s_{h+1},\cdot,\cdot)$ to $\overline{A}(\cdot,\cdot)$ from the matrix games section. The policy $\mu_{h+1}^{\Gamma}(\cdot|s_{h+1})$ is the optimal best response to $\nu_{h+1}(\cdot|s_{h+1})$ under the reward model $Q_{h+1}^{\Gamma}(\cdot|s_{h+1})$ ($\mu_{h+1}^{\Gamma} := \mu(Q^{\Gamma},\nu)$, see Proposition F.1) where

$$
\begin{aligned}
Q_{h+1}^{\Gamma}(s_{h+1},i_{h+1},j_{h+1}) &= \Gamma\overline{Q}_{h+1}(s_{h+1},i_{h+1},j_{h+1}) + (1-\Gamma)Q_{h+1}^{+}(s_{h+1},i_{h+1},j_{h+1}) \\
&= \overline{Q}_{h+1}(s_{h+1},i_{h+1},j_{h+1}) + (1-\Gamma)\left( Q_{h+1}^{+}(s_{h+1},i_{h+1},j_{h+1}) - \overline{Q}_{h+1}(s_{h+1},i_{h+1},j_{h+1}) \right)
\end{aligned}
$$

Now using Lemma F.8 we have

$$
g_2(s_{h+1}) \leq 4\beta^{-1} \mathop{\mathbb{E}}_{i_{h+1}\sim\mu_{h+1}^{\Gamma}(\cdot|s_{h+1})} \left[ \left( \mathop{\mathbb{E}}_{j_{h+1}\sim\nu_{h+1}(\cdot|s_{h+1})} \left[ Q_{h+1}^{+}(s_{h+1},i_{h+1},j_{h+1}) - \overline{Q}_{h+1}(s_{h+1},i_{h+1},j_{h+1}) \right] \right)^2 \right].
$$

and thus

$$
\begin{aligned}
J_h &\leq 5\beta^{-1} \mathop{\mathbb{E}}_{s_{h+1}\sim d_{h+1}^{\tilde{\mu},\nu}} \mathop{\mathbb{E}}_{i_{h+1}\sim\mu_{h+1}^{\Gamma}(\cdot|s_{h+1})} \left[ \left( \mathop{\mathbb{E}}_{j_{h+1}\sim\nu_{h+1}(\cdot|s_{h+1})} \left[ Q_{h+1}^{+}(s_{h+1},i_{h+1},j_{h+1}) - \overline{Q}_{h+1}(s_{h+1},i_{h+1},j_{h+1}) \right] \right)^2 \right] \\
&\leq 5\beta^{-1} \mathop{\mathbb{E}}_{\substack{s_{h+1}\sim d_{h+1}^{\tilde{\mu},\nu}}} \mathop{\mathbb{E}}_{\substack{i_{h+1}\sim\mu_{h+1}^{\Gamma}(\cdot|s_{h+1}) \\ j_{h+1}\sim\nu_{h+1}(\cdot|s_{h+1})}} \left[ \left( Q_{h+1}^{+}(s_{h+1},i_{h+1},j_{h+1}) - \overline{Q}_{h+1}(s_{h+1},i_{h+1},j_{h+1}) \right)^2 \right].
\end{aligned}
$$

Note that this is the exact form we obtain while bounding the term $T_2$ and using the same arguments (56)-(58) one can show that the term is maximized at $\Gamma = 0$ and we have $\mu_{h+1}^0 = \tilde{\mu}_{h+1}$, specifically $Q_{h+1}^{\mu}(s_{h+1},\cdot,\cdot)$ will be mapped to $A(\cdot,\cdot)$, $Q_{h+1}^{+}(s_{h+1},\cdot,\cdot)$ to $A^{+}(\cdot,\cdot)$, $Q_{h+1}^{\Gamma}(s_{h+1},\cdot,\cdot)$ will be mapped to $A_{\Gamma}(\cdot,\cdot)$ and $\overline{Q}_{h+1}(s_{h+1},\cdot,\cdot)$ to $\overline{A}(\cdot,\cdot)$.

$$
J_h \leq 5\beta^{-1} \mathop{\mathbb{E}}_{\substack{s_{h+1}\sim d_{h+1}^{\tilde{\mu},\nu}}} \mathop{\mathbb{E}}_{\substack{i_{h+1}\sim\tilde{\mu}_{h+1}(\cdot|s_{h+1}) \\ j_{h+1}\sim\nu_{h+1}(\cdot|s_{h+1})}} \left[ \left( Q_{h+1}^{+}(s_{h+1},i_{h+1},j_{h+1}) - \overline{Q}_{h+1}(s_{h+1},i_{h+1},j_{h+1}) \right)^2 \right].
\tag{85}
$$

Let $a_{h+1} = (i_{h+1}, j_{h+1})$, using Lemma F.7 we have

$$
0 \leq Q_{h+1}^{+}(s_{h+1},i_{h+1},j_{h+1}) - \overline{Q}_{h+1}(s_{h+1},i_{h+1},j_{h+1})
\tag{86}
$$

$$= \mathop{\mathbb{E}}_{s_{h+2}|s_{h+1},a_{h+1}} \left( V_{h+2}^+(s_{h+2}) - \overline{V}_{h+2}(s_{h+2}) \right) + e_{h+1}^+(s_{h+1},i_{h+1},j_{h+1}) - \overline{e}_{h+1}(s_{h+1},i_{h+1},j_{h+1})$$

$$= \mathop{\mathbb{E}}_{s_{h+2}|s_{h+1},a_{h+1}} \mathop{\mathbb{E}}_{\substack{i_{h+2}\sim\tilde{\mu}_{h+2}(\cdot|s_{h+2})\\ j_{h+2}\sim\nu_{h+2}(\cdot|s_{h+2})}} \left( Q_{h+2}^+(s_{h+2},i_{h+2},j_{h+2}) - \beta\mathrm{KL}(\tilde{\mu}_{h+2}(\cdot|s_{h+2})\|\mu_{\mathrm{ref},h+2}(\cdot|s_{h+2})) \right.$$

$$\left. + \beta\mathrm{KL}(\nu_{h+2}(\cdot|s_{h+2})\|\nu_{\mathrm{ref},h+2}(\cdot|s_{h+2})) \right)$$

$$- \mathop{\mathbb{E}}_{s_{h+2}|s_{h+1},a_{h+1}} \mathop{\mathbb{E}}_{\substack{i_{h+2}\sim\mu_{h+2}(\cdot|s_{h+2})\\ j_{h+2}\sim\nu_{h+2}(\cdot|s_{h+2})}} \left( \overline{Q}_{h+2}(s_{h+2},i_{h+2},j_{h+2}) - \beta\mathrm{KL}(\mu_{h+2}(\cdot|s_{h+2})\|\mu_{\mathrm{ref},h+2}(\cdot|s_{h+2})) \right.$$

$$\left. + \beta\mathrm{KL}(\nu_{h+2}(\cdot|s_{h+2})\|\nu_{\mathrm{ref},h+2}(\cdot|s_{h+2})) \right) + e_{h+1}^+(s_{h+1},i_{h+1},j_{h+1}) - \overline{e}_{h+1}(s_{h+1},i_{h+1},j_{h+1})$$

$$\leq \mathop{\mathbb{E}}_{s_{h+2}|s_{h+1},a_{h+1}} \mathop{\mathbb{E}}_{\substack{i_{h+2}\sim\tilde{\mu}_{h+2}(\cdot|s_{h+2})\\ j_{h+2}\sim\nu_{h+2}(\cdot|s_{h+2})}} \left[ Q_{h+2}^+(s_{h+2},i_{h+2},j_{h+2}) - \overline{Q}_{h+2}(s_{h+2},i_{h+2},j_{h+2}) \right]$$

$$+ e_{h+1}^+(s_{h+1},i_{h+1},j_{h+1}) - \overline{e}_{h+1}(s_{h+1},i_{h+1},j_{h+1}) \tag{87}$$

$$\leq \cdots$$

$$\leq \mathbb{E}^{\tilde{\mu},\nu}_{\cdot|s_{h+1},a_{h+1}} \left[ \sum_{k=h+1}^{H} e_k^+(s_k,i_k,j_k) - \overline{e}_k(s_k,i_k,j_k) \right]$$

$$\leq \left( \mathbb{E}^{\tilde{\mu},\nu}_{\cdot|s_{h+1},a_{h+1}} \left[ \sum_{k=h+1}^{H} \left| e_k^+(s_k,i_k,j_k) \right| \right] + \mathbb{E}^{\tilde{\mu},\nu}_{\cdot|s_{h+1},a_{h+1}} \left[ \sum_{k=h+1}^{H} \left| \overline{e}_k(s_k,i_k,j_k) \right| \right] \right). \tag{88}$$

Here eq. (87) follows from lower bounding the second term by swapping the policy $\mu$ by $\tilde{\mu}$ since $\mu$ is the maximizer under $\overline{Q}(\cdot|s_{h+2})$

$$\mu_{h+2}(\cdot|s_{h+2}) = \arg\max_{\mu'_{h+2}(\cdot|s_{h+2})} \mathop{\mathbb{E}}_{\substack{i_{h+2}\sim\mu'_{h+2}(\cdot|s_{h+2})\\ j_{h+2}\sim\nu_{h+2}(\cdot|s_{h+2})}} \left( \overline{Q}_{h+2}(s_{h+2},i_{h+2},j_{h+2}) - \beta\mathrm{KL}(\mu'_{h+2}(\cdot|s_{h+2})\|\mu_{\mathrm{ref},h+2}(\cdot|s_{h+2})) \right)$$

Thus combining equations (85) and (88) we have

$$J_h \leq 5\beta^{-1} \mathop{\mathbb{E}}_{\substack{s_{h+1}\sim d_{h+1}^{\tilde{\mu},\nu}}} \mathop{\mathbb{E}}_{\substack{i_{h+1}\sim\tilde{\mu}_{h+1}(\cdot|s_{h+1})\\ j_{h+1}\sim\nu_{h+1}(\cdot|s_{h+1})}} \left[ \left( \mathbb{E}^{\tilde{\mu},\nu}_{\cdot|s_{h+1},a_{h+1}} \left[ \sum_{k=h+1}^{H} \left| e_k^+(s_k,i_k,j_k) \right| + \sum_{k=h+1}^{H} \left| \overline{e}_k(s_k,i_k,j_k) \right| \right] \right)^2 \right]$$

$$\leq 5\beta^{-1} \mathbb{E}^{\tilde{\mu},\nu} \left[ \left( \sum_{k=h}^{H} \left| e_k^+(s_k,i_k,j_k) \right| + \left| \overline{e}_k(s_k,i_k,j_k) \right| \right)^2 \right]. \tag{89}$$

Here $\mathbb{E}^{\tilde{\mu},\nu}$ is used to denote $s_k \sim d_k^{\mu,\nu}$, $i_k \sim \tilde{\mu}_k(\cdot|s_k)$ and $j_k \sim \nu_k(\cdot|s_k)$.

**Step 3: Finishing up.** Define

$$\Sigma_{h,t}^+ := \lambda\mathbf{I} + \sum_{\tau\in\mathcal{D}_{t-1}^+} \phi(s_h^\tau,i_h^\tau,j_h^\tau)\phi(s_h^\tau,i_h^\tau,j_h^\tau)^\top \quad \text{and} \quad \Sigma_{h,t}^- := \lambda\mathbf{I} + \sum_{\tau\in\mathcal{D}_{t-1}^-} \phi(s_h^\tau,i_h^\tau,j_h^\tau)\phi(s_h^\tau,i_h^\tau,j_h^\tau)^\top.$$

By defining the filtration $\mathcal{F}_{t-1} = \sigma\left(\{\tau_l^+,\tau_l^-\}_{l=1}^{t-1}\right)$, where $\tau_t^+ = \left\{(s_{h,t}^+,i_{h,t}^+,j_{h,t}^+,r_{h,t}^+,s_{h+1,t}^+)\right\}_{h=1}^{H}$ and $\tau_t^- = \left\{(s_{h,t}^-,i_{h,t}^-,j_{h,t}^-,r_{h,t}^-,s_{h+1,t}^-)\right\}_{h=1}^{H}$ as defined in Algorithm 2, we observe that the random variable $\sum_{h=1}^{H} \left\|\phi\left(s_{h,t}^+,i_{h,t}^+,j_{h,t}^+\right)\right\|_{(\Sigma_{h,t}^+)^{-1}}^2$ is $\mathcal{F}_t$ measurable while the policies $\tilde{\mu}_t$ and $\nu_t$ are $\mathcal{F}_{t-1}$ measurable. Now let $\mathcal{E}_8$ denote the event

$$\mathcal{E}_8 = \left\{ \sum_{t=1}^{T} \mathbb{E}^{\tilde{\mu}_t,\nu_t} \left[ \sum_{h=1}^{H} \|\phi(s_h,i_h,j_h)\|_{(\Sigma_{h,t}^+)^{-1}}^2 \right] \leq 2\sum_{t=1}^{T}\sum_{h=1}^{H} \left\|\phi\left(s_{h,t}^+,i_{h,t}^+,j_{h,t}^+\right)\right\|_{(\Sigma_{h,t}^+)^{-1}}^2 + 8H\log\left(\frac{16}{\delta}\right) \right\}.$$

Then choosing $\lambda = 1$, $\mathbb{P}(\mathcal{E}_8) \geq 1 - \delta/8$ using Lemma D.2 with $R = H$ since $\sum_{h=1}^{H} \|\phi(s_h, i_h, j_h)\|^2_{(\Sigma_{h,t}^+)^{-1}} \leq H$ by assumption 3.1. Now under the event $\mathcal{E}_6 \cap \mathcal{E}_7 \cap \mathcal{E}_8$ (w.p. at least $1 - \delta/4$), combining equations (81), (82),(89) and bringing back the $t$ in the superscript we have

$$\sum_{t=1}^{T}(T_5^{(t)} + T_6^{(t)})$$

$$\leq \beta^{-1} \sum_{t=1}^{T} \sum_{h=1}^{H} \left( 5\mathbb{E}^{\tilde{\mu}_t, \nu_t} \left[ \left( \sum_{k=h}^{H} \left| e_{k,t}^+(s_k, i_k, j_k) \right| + |\bar{e}_{k,t}(s_k, i_k, j_k)| \right)^2 \right] + \mathbb{E}^{\tilde{\mu}_t, \nu_t} \left[ \left( \sum_{k=h}^{H} \left| e_{k,t}^+(s_k, i_k, j_k) \right| \right)^2 \right] \right)$$

$$\leq \beta^{-1} \sum_{t=1}^{T} \sum_{h=1}^{H} \left( 5\mathbb{E}^{\tilde{\mu}_t, \nu_t} \left[ \left( \sum_{k=h}^{H} 2b_{k,t}(s_k, i_k, j_k) + 3b_{k,t}^{\mathrm{mse}}(s_k, i_k, j_k) \right)^2 \right] + \mathbb{E}^{\tilde{\mu}_t, \nu_t} \left[ \left( \sum_{k=h}^{H} b_{k,t}(s_k, i_k, j_k) \right)^2 \right] \right) \quad (90)$$

$$\leq \beta^{-1} H^2 \sum_{t=1}^{T} \sum_{h=1}^{H} \left( 5\mathbb{E}^{\tilde{\mu}_t, \nu_t} \left[ (2b_{h,t}(s_h, i_h, j_h) + 3b_{h,t}^{\mathrm{mse}}(s_h, i_h, j_h))^2 \right] + \mathbb{E}^{\tilde{\mu}_t, \nu_t} \left[ (b_{h,t}(s_h, i_h, j_h))^2 \right] \right)$$

$$\leq c_3 \beta^{-1} d^2 H^6 \log \left( \frac{16dTH}{\delta \min\{1, \beta\}} \right) \sum_{t=1}^{T} \sum_{h=1}^{H} \mathbb{E}^{\tilde{\mu}_t, \nu_t} \left[ \|\phi(s_h, i_h, j_h)\|^2_{\Sigma_{h,t}^{-1}} \right] \quad (91)$$

$$\leq c_3 \beta^{-1} d^2 H^6 \log \left( \frac{16dTH}{\delta \min\{1, \beta\}} \right) \sum_{t=1}^{T} \sum_{h=1}^{H} \mathbb{E}^{\tilde{\mu}_t, \nu_t} \left[ \|\phi(s_h, i_h, j_h)\|^2_{(\Sigma_{h,t}^+)^{-1}} \right] \quad (92)$$

$$\leq 2c_3 \beta^{-1} d^2 H^6 \log \left( \frac{16dTH}{\delta \min\{1, \beta\}} \right) \left( \sum_{t=1}^{T} \left( \sum_{h=1}^{H} \left\| \phi\left(s_{h,t}^+, i_{h,t}^+, j_{h,t}^+\right) \right\|^2_{(\Sigma_{h,t}^+)^{-1}} \right) + 4H \log \left( \frac{16}{\delta} \right) \right) \quad (93)$$

$$\leq c_3' \beta^{-1} d^3 H^7 \log \left( \frac{16dTH}{\delta \min\{1, \beta\}} \right) \log \left( \frac{T+1}{\delta} \right). \quad (94)$$

Here we use Corollary F.6 and Lemma F.4 to obtain eq. (90). Eq. (91) can be derived for some universal constant $c_3$ by substituting the expressions for $b_{h,t}^{\mathrm{mse}}(s_h, i_h, j_h)$ and $b_{h,t}(s_h, i_h, j_h)$. Eq. (92) relies on the identity $\Sigma_{h,t} = \Sigma_{h,t}^+ + \Sigma_{h,t}^-$, which implies that $\Sigma_{h,t}^{-1} \preceq \left( \Sigma_{h,t}^+ \right)^{-1}$. Eq. (93) from event $\mathcal{E}_8$. Eq. (94) follows from the elliptical potential lemma (Lemma D.6). One can similarly bound the term $\sum_{t=1}^{T} \left( T_7^{(t)} + T_8^{(t)} \right)$ (w.p. $1 - \delta/4$) to obtain

$$\mathsf{Regret}(T) = \sum_{t=1}^{T} \mathsf{DualGap}(\mu_t, \nu_t) \leq \mathcal{O}\left( \beta^{-1} d^3 H^7 \log \left( \frac{dT}{\delta} \right) \log \left( \frac{dTH}{\delta \min\{1, \beta\}} \right) \right) \quad \text{w.p. } (1 - \delta/2).$$

## F.3. Proof of Theorem F.3: Traditional $\sqrt{T}$ Bound

### F.3.1. $\beta > 0$: REGULARIZED SETTING

For simplicity we again fix the initial state to $s_1$, extending the arguments to a fixed initial distribution $s_1 \sim \rho$ is trivial. Recall the dual gap can be decomposed as $\mathsf{DualGap}(\mu_t, \nu_t) = T_5^{(t)} + T_6^{(t)} + T_7^{(t)} + T_8^{(t)}$ as per equation (77). We will bound the terms $T_5^{(t)}$ and $T_6^{(t)}$ and the remaining terms can be bounded similarly.

**Step 1: Bounding $T_5^{(t)}$.** Let $\mu_t^\dagger$ denote the best response to $\nu_t$ at time $t$. We shall omit $\nu_t$ in the superscript of $Q$ for notational simplicity. Then under the event $\mathcal{E}_6 \cap \mathcal{E}_7$ we have

$$T_5^{(t)} = V_1^{\star, \nu_t}(s_1) - V_1^{\tilde{\mu}_t, \nu_t}(s_1)$$

$$= \mathop{\mathbb{E}}_{\substack{i_1 \sim \mu_{1,t}^\dagger(\cdot|s_1) \\ j_i \sim \nu_{1,t}(\cdot|s_1)}} \left[ Q_1^{\mu_t^\dagger}(s_1, i_1, j_1) \right] - \beta \mathsf{KL}(\mu_{1,t}^\dagger(\cdot\|s_1)\|\mu_{\mathrm{ref},1}(\cdot\|s_1))$$

$$- \left( \underset{\substack{i_1 \sim \tilde{\mu}_{1,t}(\cdot|s_1) \\ j_i \sim \nu_{1,t}(\cdot|s_1)}}{\mathbb{E}} \left[ Q_1^{\tilde{\mu}_t}(s_1, i_1, j_1) \right] - \beta \mathrm{KL}(\tilde{\mu}_{1,t}(\cdot\|s_1) \| \mu_{\mathrm{ref},1}(\cdot\|s_1)) \right)$$

$$\leq \underset{\substack{i_1 \sim \mu_{1,t}^\dagger(\cdot|s_1) \\ j_i \sim \nu_{1,t}(\cdot|s_1)}}{\mathbb{E}} \left[ Q_{1,t}^+(s_1, i_1, j_1) \right] - \beta \mathrm{KL}(\mu_{1,t}^\dagger(\cdot\|s_1) \| \mu_{\mathrm{ref},1}(\cdot\|s_1))$$

$$- \left( \underset{\substack{i_1 \sim \tilde{\mu}_{1,t}(\cdot|s_1) \\ j_i \sim \nu_{1,t}(\cdot|s_1)}}{\mathbb{E}} \left[ Q_1^{\tilde{\mu}_t}(s_1, i_1, j_1) \right] - \beta \mathrm{KL}(\tilde{\mu}_{1,t}(\cdot\|s_1) \| \mu_{\mathrm{ref},1}(\cdot\|s_1)) \right) \tag{95}$$

$$\leq \underset{\substack{i_1 \sim \tilde{\mu}_{1,t}(\cdot|s_1) \\ j_i \sim \nu_{1,t}(\cdot|s_1)}}{\mathbb{E}} \left[ Q_{1,t}^+(s_1, i_1, j_1) \right] - \underset{\substack{i_1 \sim \tilde{\mu}_{1,t}(\cdot|s_1) \\ j_i \sim \nu_{1,t}(\cdot|s_1)}}{\mathbb{E}} \left[ Q_1^{\tilde{\mu}_t}(s_1, i_1, j_1) \right] \tag{96}$$

$$= \underset{\substack{i_1 \sim \tilde{\mu}_{1,t}(\cdot|s_1) \\ j_i \sim \nu_{1,t}(\cdot|s_1)}}{\mathbb{E}} \left[ P_1 V_{2,t}^+(s_1, i_1, j_1) \right] - \underset{\substack{i_1 \sim \tilde{\mu}_{1,t}(\cdot|s_1) \\ j_i \sim \nu_{1,t}(\cdot|s_1)}}{\mathbb{E}} \left[ P_1 V_{2,t}^{\tilde{\mu}_t}(s_1, i_1, j_1) \right] + \underset{\substack{i_1 \sim \tilde{\mu}_{1,t}(\cdot|s_1) \\ j_i \sim \nu_{1,t}(\cdot|s_1)}}{\mathbb{E}} \left[ e_{1,t}^+(s_1, i_1, j_1) \right]$$

$$= \mathbb{E}^{\tilde{\mu}_t, \nu_t} \left[ V_{2,t}^+(s_2) - V_{2,t}^{\tilde{\mu}_t}(s_2) \right] + \mathbb{E}^{\tilde{\mu}_t, \nu_t} \left[ e_{1,t}^+(s_1, i_1, j_1) \right]$$

$$= \cdots$$

$$= \mathbb{E}^{\tilde{\mu}_t, \nu_t} \left[ \sum_{h=1}^H e_{h,t}^+(s_h, i_h, j_h) \right]. \tag{97}$$

Here eq. (95) follows from optimism (Lemma F.7) and eq. (96) follows since $\tilde{\mu}_{1,t}(\cdot|s_1)$ is the optimal policy under $Q_1^+(s_1, \cdot, \cdot)$.

**Step 2: Bounding $T_6^{(t)}$.** We have

$$T_6^{(t)} = V_1^{\tilde{\mu}_t, \nu_t}(s_1) - V_1^{\mu_t, \nu_t}(s_1)$$

$$= \underbrace{V_1^{\tilde{\mu}_t, \nu_t}(s_1) - \overline{V}_{1,t}(s_1)}_{T_{6a}^{(t)}} + \underbrace{\overline{V}_{1,t}(s_1) - V_1^{\mu_t, \nu_t}(s_1)}_{T_{6b}^{(t)}}.$$

Here we again omit $\nu_t$ in the superscript for notational simplicity. Under the event $\mathcal{E}_6 \cap \mathcal{E}_7$, the term $T_{6a}^{(t)}$ can be bounded as follows

$$T_{6a}^{(t)} = V_1^{\tilde{\mu}_t, \nu_t}(s_1) - \overline{V}_{1,t}(s_1)$$

$$= \underset{\substack{i_1 \sim \tilde{\mu}_{1,t}(\cdot|s_1) \\ j_i \sim \nu_{1,t}(\cdot|s_1)}}{\mathbb{E}} \left[ Q_1^{\tilde{\mu}_t}(s_1, i_1, j_1) \right] - \beta \mathrm{KL}(\tilde{\mu}_{1,t}(\cdot\|s_1) \| \mu_{\mathrm{ref},1}(\cdot\|s_1))$$

$$- \left( \underset{\substack{i_1 \sim \mu_{1,t}(\cdot|s_1) \\ j_i \sim \nu_{1,t}(\cdot|s_1)}}{\mathbb{E}} \left[ \overline{Q}_{1,t}(s_1, i_1, j_1) \right] - \beta \mathrm{KL}(\mu_{1,t}(\cdot\|s_1) \| \mu_{\mathrm{ref},1}(\cdot\|s_1)) \right)$$

$$\leq \underset{\substack{i_1 \sim \tilde{\mu}_{1,t}(\cdot|s_1) \\ j_i \sim \nu_{1,t}(\cdot|s_1)}}{\mathbb{E}} \left[ Q_{1,t}^+(s_1, i_1, j_1) \right] - \beta \mathrm{KL}(\tilde{\mu}_{1,t}(\cdot\|s_1) \| \mu_{\mathrm{ref},1}(\cdot\|s_1)) -$$

$$\left( \underset{\substack{i_1 \sim \tilde{\mu}_{1,t}(\cdot|s_1) \\ j_i \sim \nu_{1,t}(\cdot|s_1)}}{\mathbb{E}} \left[ \overline{Q}_{1,t}(s_1, i_1, j_1) \right] - \beta \mathrm{KL}(\tilde{\mu}_t(\cdot\|s_1) \| \mu_{\mathrm{ref}}(\cdot\|s_1)) \right) \tag{98}$$

$$= \underset{\substack{i_1 \sim \tilde{\mu}_{1,t}(\cdot|s_1) \\ j_i \sim \nu_{1,t}(\cdot|s_1)}}{\mathbb{E}} \left[ Q_{1,t}^+(s_1, i_1, j_1) - \overline{Q}_{1,t}(s_1, i_1, j_1) \right]. \tag{99}$$

Here eq. (98) follows by upper bounding $Q_1^{\tilde{\mu}_t}$ by $Q_{1,t}^+$ using optimism (Lemma F.7) in the first (positive) term and lower bounding the second (negative) term by switching the max players policy to $\tilde{\mu}_{1,t}(\cdot|s_1)$ since

$$\mu_{1,t}(\cdot|s_1) = \arg\max_{\mu_1'(\cdot|s_1)} \left( \mathbb{E}_{\substack{i_1 \sim \mu_{1,t}'(\cdot|s_1) \\ j_i \sim \nu_{1,t}(\cdot|s_1)}} \left[ \overline{Q}_{1,t}(s_1, i_1, j_1) \right] - \beta \mathrm{KL}(\mu_1'(\cdot\|s_1)\|\mu_{\mathrm{ref},1}(\cdot\|s_1)) \right)$$

is the optimal policy under $\overline{Q}_{1,t}$. Under the event $\mathcal{E}_6 \cap \mathcal{E}_7$, we bound $T_{6b}^{(t)}$ as follows

$$
\begin{aligned}
T_{6b}^{(t)} &= \overline{V}_{1,t}(s_1) - V_1^{\mu_t, \nu_t}(s_1) \\
&= \mathbb{E}_{\substack{i_1 \sim \mu_{1,t}(\cdot|s_1) \\ j_1 \sim \nu_{1,t}(\cdot|s_1)}} \left[ \overline{Q}_{1,t}(s_1, i_1, j_1) - Q_1^{\mu_t}(s_1, i_1, j_1) \right] \\
&\le \mathbb{E}_{\substack{i_1 \sim \mu_{1,t}(\cdot|s_1) \\ j_1 \sim \nu_{1,t}(\cdot|s_1)}} \left[ Q_{1,t}^+(s_1, i_1, j_1) - \overline{Q}_{1,t}(s_1, i_1, j_1) \right] \qquad (100) \\
&= \mathbb{E}_{\substack{i_1 \sim \mu_{1,t}(\cdot|s_1) \\ j_1 \sim \nu_{1,t}(\cdot|s_1)}} \left[ Q_{1,t}^+(s_1, i_1, j_1) \right] - \beta \mathrm{KL}(\mu_{1,t}(\cdot\|s_1)\|\mu_{\mathrm{ref},1}(\cdot\|s_1)) \\
&\quad - \left( \mathbb{E}_{\substack{i_1 \sim \mu_{1,t}(\cdot|s_1) \\ j_1 \sim \nu_{1,t}(\cdot|s_1)}} \left[ \overline{Q}_{1,t}(s_1, i_1, j_1) \right] - \beta \mathrm{KL}(\mu_{1,t}(\cdot\|s_1)\|\mu_{\mathrm{ref},1}(\cdot\|s_1)) \right) \\
&\le \mathbb{E}_{\substack{i_1 \sim \tilde{\mu}_{1,t}(\cdot|s_1) \\ j_1 \sim \nu_{1,t}(\cdot|s_1)}} \left[ Q_{1,t}^+(s_1, i_1, j_1) \right] - \beta \mathrm{KL}(\tilde{\mu}_t(\cdot\|s_1)\|\mu_{\mathrm{ref}}(\cdot\|s_1)) \\
&\quad - \left( \mathbb{E}_{\substack{i_1 \sim \tilde{\mu}_{1,t}(\cdot|s_1) \\ j_1 \sim \nu_{1,t}(\cdot|s_1)}} \left[ \overline{Q}_{1,t}(s_1, i_1, j_1) \right] - \beta \mathrm{KL}(\tilde{\mu}_{1,t}(\cdot\|s_1)\|\mu_{\mathrm{ref},1}(\cdot\|s_1)) \right) \qquad (101) \\
&= \mathbb{E}_{\substack{i_1 \sim \tilde{\mu}_{1,t}(\cdot|s_1) \\ j_1 \sim \nu_{1,t}(\cdot|s_1)}} \left[ Q_{1,t}^+(s_1, i_1, j_1) - \overline{Q}_{1,t}(s_1, i_1, j_1) \right]. \qquad (102)
\end{aligned}
$$

Here eq. (100) follows from Lemma F.8 and Lemma F.7. Eq. (101) follows by upper bounding the first term and lower bounding the second term by swapping policy $\mu_t(\cdot\|s_1)$ by $\tilde{\mu}_t(\cdot\|s_1)$ since $\tilde{\mu}_{1,t}(\cdot\|s_1)$ is the optimal policy under $Q_{1,t}^+(s_1, \cdot, \cdot)$ and $\mu_t(\cdot\|s_1)$ is the optimal policy under $\overline{Q}_{1,t}(s_1, \cdot, \cdot)$. From equations (99) and (102) under the event $\mathcal{E}_6 \cap \mathcal{E}_7$, we have

$$
\begin{aligned}
T_6^{(t)} &\le 2 \mathbb{E}_{\substack{i_1 \sim \tilde{\mu}_{1,t}(\cdot|s_1) \\ j_1 \sim \nu_{1,t}(\cdot|s_1)}} \left[ Q_{1,t}^+(s_1, i_1, j_1) - \overline{Q}_{1,t}(s_1, i_1, j_1) \right] \\
&\le 2 \left( \mathbb{E}^{\tilde{\mu}_t, \nu_t} \left[ \sum_{k=1}^H \left| e_{h,t}^+(s_h, i_h, j_h) \right| \right] + \mathbb{E}^{\tilde{\mu}_t, \nu_t} \left[ \sum_{h=1}^H |\overline{e}_{h,t}(s_h, i_h, j_h)| \right] \right). \qquad (103)
\end{aligned}
$$

Here eq. (103) can be obtained using the same steps used in obtaining equations (86)-(88).

**Step 3: Finishing up.** By defining the filtration $\mathcal{F}_{t-1} = \sigma\left(\{\tau_l^+, \tau_l^-\}_{l=1}^{t-1}\right)$, we observe that the random variable $\sum_{h=1}^H \left\| \phi\left(s_{h,t}^+, i_{h,t}^+, j_{h,t}^+\right) \right\|_{(\Sigma_{h,t}^+)^{-1}}$ is $\mathcal{F}_t$ measurable while the policies $\tilde{\mu}_t$ and $\nu_t$ are $\mathcal{F}_{t-1}$ measurable. Now let $\mathcal{E}_9$ denote the event

$$\mathcal{E}_9 = \left\{ \sum_{t=1}^T \mathbb{E}^{\tilde{\mu}_t, \nu_t} \left[ \sum_{h=1}^H \| \phi(s_h, i_h, j_h) \|_{(\Sigma_{h,t}^+)^{-1}} \right] \le 2 \sum_{t=1}^T \sum_{h=1}^H \left\| \phi\left(s_{h,t}^+, i_{h,t}^+, j_{h,t}^+\right) \right\|_{(\Sigma_{h,t}^+)^{-1}} + 8H \log\left(\frac{16}{\delta}\right) \right\}.$$

Then choosing $\lambda = 1$, $\mathbb{P}(\mathcal{E}_9) \geq 1 - \delta/8$ by Lemma D.2 with $R = H$ since $\sum_{h=1}^{H} \|\phi(s_h, i_h, j_h)\|_{(\Sigma_{h,t}^+)^{-1}} \leq H$ by assumption 3.1. Now using equations (97) and (103) under the event $\mathcal{E}_6 \cap \mathcal{E}_7 \cap \mathcal{E}_9$ (w.p. $1 - \delta/4$) we have

$$\sum_{t=1}^{T} \left( T_5^{(t)} + T_6^{(t)} \right) \leq \sum_{t=1}^{T} \left( 3\mathbb{E}^{\tilde{\mu}_t, \nu_t} \left[ \sum_{h=1}^{H} \left| e_{h,t}^+(s_h, i_h, j_h) \right| \right] + 2\mathbb{E}^{\tilde{\mu}_t, \nu_t} \left[ \sum_{h=1}^{H} |\overline{e}_{h,t}(s_h, i_h, j_h)| \right] \right)$$

$$\leq \sum_{t=1}^{T} \left( 3\mathbb{E}^{\tilde{\mu}_t, \nu_t} \left[ \sum_{h=1}^{H} \left( 2b_{h,t}(s_h, i_h, j_h) + 2b_{h,t}^{\mathrm{mse}}(s_h, i_h, j_h) \right) \right] + 2\mathbb{E}^{\tilde{\mu}_t, \nu_t} \left[ \sum_{h=1}^{H} b_{h,t}^{\mathrm{mse}}(s_h, i_h, j_h) \right] \right) \tag{104}$$

$$\leq c_4 dH^2 \sqrt{\log \left( \frac{16dTH}{\delta \min\{1, \beta\}} \right)} \sum_{t=1}^{T} \sum_{h=1}^{H} \mathbb{E}^{\tilde{\mu}_t, \nu_t} \left[ \|\phi(s_h, i_h, j_h)\|_{\Sigma_{h,t}^{-1}} \right] \tag{105}$$

$$\leq c_4 dH^2 \sqrt{\log \left( \frac{16dTH}{\delta \min\{1, \beta\}} \right)} \sum_{t=1}^{T} \sum_{h=1}^{H} \mathbb{E}^{\tilde{\mu}_t, \nu_t} \left[ \|\phi(s_h, i_h, j_h)\|_{(\Sigma_{h,t}^+)^{-1}} \right] \tag{106}$$

$$\leq 2c_4 dH^2 \sqrt{\log \left( \frac{16dTH}{\delta \min\{1, \beta\}} \right)} \left( \sum_{t=1}^{T} \sum_{h=1}^{H} \left\| \phi\left( s_{h,t}^+, i_{h,t}^+, j_{h,t}^+ \right) \right\|_{(\Sigma_{h,t}^+)^{-1}} + 4H \log \left( \frac{16}{\delta} \right) \right) \tag{107}$$

$$\leq 2c_4 dH^2 \sqrt{\log \left( \frac{16dTH}{\delta \min\{1, \beta\}} \right)} \left( \sum_{h=1}^{H} \sqrt{T \sum_{t=1}^{T} \left\| \phi\left( s_{h,t}^+, i_{h,t}^+, j_{h,t}^+ \right) \right\|_{(\Sigma_{h,t}^+)^{-1}}^2} + 4H \log \left( \frac{16}{\delta} \right) \right)$$

$$\leq c_4' dH^3 \sqrt{\log \left( \frac{16dTH}{\delta \min\{1, \beta\}} \right)} \left( \sqrt{dT \log(T+1)} + 4 \log \left( \frac{16}{\delta} \right) \right). \tag{108}$$

Here we use Corollary F.6 and Lemma F.4 to obtain eq. (104). Eq. (105) can be derived for some universal constant $c_4$ by substituting the expressions for $b_{h,t}^{\mathrm{mse}}(s_h, i_h, j_h)$ and $b_{h,t}(s_h, i_h, j_h)$. Eq. (106) uses the fact $\Sigma_{h,t} \succeq \Sigma_{h,t}^+$. The bound in (107) follows from event $\mathcal{E}_9$. Eq. (108) follows from the elliptical potential lemma (Lemma D.6). One can similarly bound the term $\sum_{t=1}^{T} \left( T_7^{(t)} + T_8^{(t)} \right)$ (w.p. $1 - \delta/4$) to obtain

$$\mathrm{Regret}(T) = \sum_{t=1}^{T} \mathsf{DualGap}(\mu_t, \nu_t) \leq \mathcal{O} \left( d^{3/2} H^3 \sqrt{T} \sqrt{\log \left( \frac{dT}{\delta} \right) \log \left( \frac{dTH}{\delta \min\{1, \beta\}} \right)} \cdot \right) \quad \text{w.p. } (1 - \delta/2).$$

## F.4. Proofs of Supporting Lemmas

### F.4.1. PROOF OF LEMMA F.4

Using Lemma D.8, with the covering number bound in Lemma F.14, $B_1 = H$ (from Lemma F.10), $L = 2H\sqrt{2dt/\lambda}$ (from Lemma F.13), $B_3 = 0$, we have with probability at least $1 - \delta/16H \; \forall \, t \in [T]$,

$$\left\| \sum_{\tau \in \mathcal{D}_{t-1}} \phi_{h,t} \left[ \overline{V}_{h+1,t}\left( s_{h+1}^\tau \right) - P_h \overline{V}_{h+1,t}(s_h^\tau, i_h^\tau, j_h^\tau) \right] \right\|_{\Sigma_{h,t}^{-1}}^2$$

$$\leq 4H^2 \left[ \frac{d}{2} \log \left( \frac{2t + \lambda}{\lambda} \right) + d \log \left( 1 + \frac{8H\sqrt{2dt}}{\varepsilon\sqrt{\lambda}} \right) + \log \left( \frac{16H}{\delta} \right) \right] + \frac{32t^2\varepsilon^2}{\lambda}.$$

Choosing $\lambda = 1$ and $\varepsilon = \sqrt{d}H/t$, we have $\forall \, h \in [H]$ and $\forall \, t \in [T]$

$$\left\| \sum_{\tau \in \mathcal{D}_{t-1}} \phi_{h,t} \left[ \overline{V}_{h+1,t}\left( s_{h+1}^\tau \right) - P_h \overline{V}_{h+1,t}(s_h^\tau, i_h^\tau, j_h^\tau) \right] \right\|_{\Sigma_{h,t}^{-1}} \leq C_1 \sqrt{d}H \sqrt{\log \left( \frac{16HT}{\delta} \right)}, \tag{109}$$

for some universal constant $C_1 > 0$. Note that we have to divide the $\delta/16$ budget across the horizon since at every $h \in [H]$ the function class used while applying lemma D.8 is different as a result of the projection operation varying with $h$. Since $r_h(s, i, j) + P_h \overline{V}_{h+1}(s, i, j) \in [0, H - h + 1]$ from Lemma F.10, and $\overline{Q}_{h,t}(s, i, j) = \Pi_h(\langle \overline{\theta}_{h,t}, \phi(s, i, j) \rangle)$, we have

$$\left| \overline{Q}_{h,t}(s, i, j) - r_h(s, i, j) - P_h \overline{V}_{h+1}(s, i, j) \right| \leq \left| \langle \overline{\theta}_{h,t}, \phi(s, i, j) \rangle - r_h(s, i, j) - P_h \overline{V}_{h+1}(s, i, j) \right|. \tag{110}$$

Now let $\pi^\star = (\mu^\star, \nu^\star)$ be the nash equilibrium policy of the true MDP, and $\theta_h^{\pi^\star}$ be its corresponding parameter, whose existence is guaranteed by Lemma F.12, we have

$$\theta_h^{\pi^\star} = \Sigma_{h,t}^{-1} \left( \sum_{\tau \in \mathcal{D}_{t-1}} \phi_{h,\tau} \phi_{h,\tau}^\top + \lambda \mathbf{I} \right) \theta_h^{\pi^\star} = \Sigma_{h,t}^{-1} \left( \sum_{\tau \in \mathcal{D}_{t-1}} \phi_{h,\tau} (r_{h,\tau} + P_h V_{h+1,t}^{\pi^\star}) + \lambda \theta_h^{\pi^\star} \right). \tag{111}$$

Also recall

$$\overline{\theta}_{h,t} = \Sigma_{h,t}^{-1} \sum_{\tau \in \mathcal{D}_{t-1}} \phi_{h,\tau} \left[ r_{h,\tau} + \overline{V}_{h+1,t}(s_{h+1}^\tau) \right].$$

Using the above two equations we have

$$\begin{aligned}
\overline{\theta}_{h,t} - \theta_h^{\pi^\star} &= \Sigma_{h,t}^{-1} \left\{ \sum_{\tau \in \mathcal{D}_{t-1}} \phi_{h,\tau} \left[ \overline{V}_{h+1,t}(s_{h+1}^\tau) - P_h V_{h+1}^{\pi^\star}(s_h^\tau, i_h^\tau, j_h^\tau) \right] - \lambda \theta_h^{\pi^\star} \right\} \\
&= \underbrace{-\lambda \Sigma_{h,t}^{-1} \theta_h^{\pi^\star}}_{p_1} + \underbrace{\Sigma_{h,t}^{-1} \sum_{\tau \in \mathcal{D}_{t-1}} \phi_{h,\tau} \left[ \overline{V}_{h+1,t}(s_{h+1}^\tau) - P_h \overline{V}_{h+1,t}(s_h^\tau, i_h^\tau, j_h^\tau) \right]}_{p_2} \\
&\quad + \underbrace{\Sigma_{h,t}^{-1} \sum_{\tau \in \mathcal{D}_{t-1}} \phi_{h,\tau} \left[ P_h \left( \overline{V}_{h+1,t}(s_h^\tau, i_h^\tau, j_h^\tau) - V_{h+1}^{\pi^\star}(s_h^\tau, i_h^\tau, j_h^\tau) \right) \right]}_{p_3}.
\end{aligned} \tag{112}$$

Assuming eq. (109) holds (w.p. $1 - \delta/16$), one can bound the terms as follows:

$$|\langle \phi(s, i, j), p_1 \rangle| = \left| \langle \phi(s, i, j), \lambda \Sigma_{h,t}^{-1} \theta_h^{\pi^\star} \rangle \right| \leq \lambda \left\| \theta_h^{\pi^\star} \right\|_{\Sigma_{h,t}^{-1}} \| \phi(s, i, j) \|_{\Sigma_{h,t}^{-1}} \leq 2H\sqrt{d\lambda} \| \phi(s, i, j) \|_{\Sigma_{h,t}^{-1}}, \tag{113a}$$

$$|\langle \phi(s, i, j), p_2 \rangle| \leq C_1 \sqrt{d} H \sqrt{\log \left( \frac{16HT}{\delta} \right)} \| \phi(s, i, j) \|_{\Sigma_{h,t}^{-1}}. \tag{113b}$$

Here eq. (113a) follows from Lemma F.12. We use the result from eq. (109) to obtain upper bound in eq. (113b). Lastly we have

$$\begin{aligned}
\langle \phi(s, i, j), p_3 \rangle &= \left\langle \phi(s, i, j), \Sigma_{h,t}^{-1} \sum_{\tau \in \mathcal{D}_{t-1}} \phi_{h,\tau} \left[ P_h \left( \overline{V}_{h+1,t}(s_h^\tau, i_h^\tau, j_h^\tau) - V_{h+1}^{\pi^\star}(s_h^\tau, i_h^\tau, j_h^\tau) \right) \right] \right\rangle \\
&= \left\langle \phi(s, i, j), \Sigma_{h,t}^{-1} \sum_{\tau \in \mathcal{D}_{t-1}} \phi_{h,\tau} (\phi_{h,\tau})^\top \left[ \int \left( \overline{V}_{h+1,t}(s') - V_{h+1}^{\pi^\star}(s') \right) d\psi(s') \right] \right\rangle \\
&= \left\langle \phi(s, i, j), \int \left( \overline{V}_{h+1,t}(s') - V_{h+1}^{\pi^\star}(s') \right) d\psi(s') \right\rangle \\
&\quad - \lambda \left\langle \phi(s, i, j), \Sigma_{h,t}^{-1} \int \left( \overline{V}_{h+1,t}(s') - V_{h+1}^{\pi^\star}(s') \right) d\psi(s') \right\rangle \\
&= P_h \left( \overline{V}_{h+1,t} - V_{h+1}^{\pi^\star} \right) (s, i, j) - \lambda \left\langle \phi(s, i, j), \Sigma_{h,t}^{-1} \int \left( \overline{V}_{h+1,t}(s') - V_{h+1}^{\pi^\star}(s') \right) d\psi(s') \right\rangle.
\end{aligned}$$

Thus

$$\left| \langle \phi(s,i,j), p_3 \rangle - P_h \left( \overline{V}_{h+1,t} - V_{h+1}^{\pi^\star} \right)(s,i,j) \right| = \left| -\lambda \left\langle \phi(s,i,j), \Sigma_{h,t}^{-1} \int \left( \overline{V}_{h+1,t}(s') - V_{h+1}^{\pi^\star}(s') \right) d\psi(s') \right\rangle \right|$$

$$\leq 2H\sqrt{d\lambda} \left\| \phi(s,i,j) \right\|_{\Sigma_{h,t}^{-1}} \tag{113c}$$

Here eq. (113c) follows from Lemma F.10 and Lemma F.9. Now

$$\langle \overline{\theta}_{h,t}, \phi(s,i,j) \rangle - r_h(s,i,j) - P_h \overline{V}_{h+1}(s,i,j)$$

$$= \langle \overline{\theta}_{h,t}, \phi(s,i,j) \rangle - Q_h^{\pi^\star}(s,i,j) - P_h \left( \overline{V}_{h+1,t} - V_{h+1}^{\pi^\star} \right)(s,i,j)$$

$$= \left\langle \phi(s,i,j), \overline{\theta}_{h,t} - \theta_h^{\pi^\star} \right\rangle - P_h \left( \overline{V}_{h+1,t} - V_{h+1}^{\pi^\star} \right)(s,i,j)$$

$$\overset{(112)}{=} \langle \phi(s,i,j), p_1 \rangle + \langle \phi(s,i,j), p_2 \rangle + \langle \phi(s,i,j), p_3 \rangle - P_h \left( \overline{V}_{h+1,t} - V_{h+1}^{\pi^\star} \right)(s,i,j). \tag{114}$$

Using the equations (113a),(113b), (113c), (114) we have

$$\left| \langle \overline{\theta}_{h,t}, \phi(s,i,j) \rangle - r_h(s,i,j) - P_h \overline{V}_{h+1}(s,i,j) \right| \leq c_1 \sqrt{d} H \sqrt{\log\left( \frac{16HT}{\delta} \right)} \left\| \phi(s,i,j) \right\|_{\Sigma_{h,t}^{-1}}$$

for some universal constant $c_1 > 0$. Using eq. (110) completes the proof

$$\left| \overline{Q}_{h,t}(s,i,j) - r_h(s,i,j) - P_h \overline{V}_{h+1}(s,i,j) \right| \leq \left| \langle \overline{\theta}_{h,t}, \phi(s,i,j) \rangle - r_h(s,i,j) - P_h \overline{V}_{h+1}(s,i,j) \right|$$

$$\leq c_1 \sqrt{d} H \sqrt{\log\left( \frac{16HT}{\delta} \right)} \left\| \phi(s,i,j) \right\|_{\Sigma_{h,t}^{-1}}.$$

### F.4.2. PROOF OF LEMMA F.5

Using Lemma D.8 with with the covering number bound in Lemma F.15, $B_1 = 4H^2$ (from Lemma F.11), $L = 2H\sqrt{2dt/\lambda}$ (from Lemma F.13), $L^+ = 4H^2\sqrt{2dt/\lambda}$ (from Lemma F.13), $B_2^+ + B_2 - b_2 \leq 4H^2$ (from equation (20)) and $B_3 = \eta_2 + 2\eta_1$ we have with probability at least $1 - \delta/16H \; \forall \; t \in [T]$

$$\left\| \sum_{\tau \in \mathcal{D}_{t-1}} \phi_{h,t} \left[ V_{h+1,t}^+ \left( s_{h+1}^\tau \right) - P_h V_{h+1,t}^+(s_h^\tau, i_h^\tau, j_h^\tau) \right] \right\|_{\Sigma_{h,t}^{-1}}^2$$

$$\leq 64H^4 \left[ \frac{d}{2} \log\left( \frac{2t+\lambda}{\lambda} \right) + d\log\left( 1 + \frac{48H^2\sqrt{2dt}}{\varepsilon\sqrt{\lambda}} \right) + d^2 \log\left( 1 + \frac{72\sqrt{d}(\eta_2 + 2\eta_1)^2}{\lambda\varepsilon^2} \right) \right.$$

$$\left. + d\log\left( 1 + \frac{48H\sqrt{2dt}}{\varepsilon\sqrt{\lambda}} \right) + d\log\left( 1 + \frac{4608H^5\sqrt{2dt}}{\varepsilon^2\beta\sqrt{\lambda}} \right) + \log\left( \frac{16H}{\delta} \right) \right] + \frac{32t^2\varepsilon^2}{\lambda},$$

Setting $\lambda = 1$ and $\eta_1 = c_1\sqrt{d}H\sqrt{\log\left( \frac{16TH}{\delta} \right)}$, $\varepsilon = dH^2/T$ and $\eta_2 = c_2 dH^2 \sqrt{\log\left( \frac{16dTH}{\min\{1,\beta\}\delta} \right)}$, we have

$$\left\| \sum_{\tau \in \mathcal{D}_{t-1}} \phi_{h,t} \left[ V_{h+1,t}^+ \left( s_{h+1}^\tau \right) - P_h V_{h+1,t}^+(s_h^\tau, i_h^\tau, j_h^\tau) \right] \right\|_{\Sigma_{h,t}^{-1}} \leq C_2 dH^2 \sqrt{\log\left( \frac{16((c_2 + 2c_1) + 1)dTH}{\delta\min\{1,\beta\}} \right)}, \tag{115}$$

for some universal constant $C_2 > 0$. Using the same steps as used in the proof of Lemma F.4 we have $\forall \; h \in [H]$ and $t \in [T]$

$$\theta_{h,t}^+ - \theta_h^{\pi^\star} = \underbrace{-\lambda \Sigma_{h,t}^{-1} \theta_h^{\pi^\star}}_{p_4} + \underbrace{\Sigma_{h,t}^{-1} \sum_{\tau \in \mathcal{D}_{t-1}} \phi_{h,\tau} \left[ V_{h+1,t}^+(s_{h+1}^\tau) - P_h V_{h+1,t}^+(s_h^\tau, i_h^\tau, j_h^\tau) \right]}_{p_5}$$

$$+ \Sigma_{h,t}^{-1} \sum_{\tau \in \mathcal{D}_{t-1}} \phi_{h,\tau} \Big[ P_h \Big( V_{h+1,t}^+(s_h^\tau, i_h^\tau, j_h^\tau) - V_{h+1}^{\pi^\star}(s_h^\tau, i_h^\tau, j_h^\tau) \Big) \Big] .$$
$$\underbrace{\hspace{9cm}}_{p_6}$$

Assuming eq. (115) holds (w.p. $1 - \delta/16$), one can bound the terms as follows

$$|\langle \phi(s,i,j), p_4 \rangle| = \left| \phi(s,i,j), \lambda \Sigma_{h,t}^{-1} \theta_h^{\pi^\star} \right| \leq \lambda \left\| \theta_h^{\pi^\star} \right\|_{\Sigma_{h,t}^{-1}} \| \phi(s,i,j) \|_{\Sigma_{h,t}^{-1}} \leq 2H\sqrt{d\lambda} \, \| \phi(s,i,j) \|_{\Sigma_{h,t}^{-1}}, \tag{116a}$$

$$|\langle \phi(s,i,j), p_5 \rangle| \leq C_2 dH^2 \sqrt{\log \left( \frac{16((c_2 + 2c_1) + 1)dTH}{\delta \min\{1,\beta\}} \right)} \, \| \phi(s,i,j) \|_{\Sigma_{h,t}^{-1}}. \tag{116b}$$

Here eq. (116a) follows from Lemma F.12. We use the result from eq. (115) to obtain upper bound in eq. (116b) Lastly using similar arguments as Lemma (F.4) we have

$$\langle \phi(s,i,j), p_6 \rangle = \left\langle \phi(s,i,j), \Sigma_{h,t}^{-1} \sum_{\tau \in \mathcal{D}_{t-1}} \phi_{h,\tau} \Big[ P_h \Big( V_{h+1,t}^+(s_h^\tau, i_h^\tau, j_h^\tau) - V_{h+1}^{\pi^\star}(s_h^\tau, i_h^\tau, j_h^\tau) \Big) \Big] \right\rangle$$

$$= P_h \left( V_{h+1,t}^+ - V_{h+1}^{\pi^\star} \right)(s,i,j) - \lambda \left\langle \phi(s,i,j), \Sigma_{h,t}^{-1} \int \left( V_{h+1,t}^+(s') - V_{h+1}^{\pi^\star}(s') \right) d\psi(s') \right\rangle.$$

Thus

$$\left| \langle \phi(s,i,j), p_6 \rangle - P_h \left( V_{h+1,t}^+ - V_{h+1}^{\pi^\star} \right)(s,i,j) \right| = \left| -\lambda \left\langle \phi(s,i,j), \Sigma_{h,t}^{-1} \int \left( V_{h+1,t}^+(s') - V_{h+1}^{\pi^\star}(s') \right) d\psi(s') \right\rangle \right|$$

$$\leq 6H^2 \sqrt{d\lambda} \, \| \phi(s,i,j) \|_{\Sigma_{h,t}^{-1}} \tag{116c}$$

Here eq. (116c) follows from Lemma (F.11) and Lemma (F.9). Using the equations (116a),(116b), (116c), and the fact $\left\langle \phi(s,i,j), \theta_{h,t}^+ \right\rangle - Q_h^{\pi^\star}(s,i,j) = \left\langle \phi(s,i,j), \theta_{h,t}^+ - \theta_h^{\pi^\star} \right\rangle = \langle \phi(s,i,j), p_4 \rangle + \langle \phi(s,i,j), p_5 \rangle + \langle \phi(s,i,j), p_6 \rangle$ for $\lambda = 1$, using similar arguments to Lemma F.4, we have

$$\left| \langle \theta_{h,t}^+, \phi(s,i,j) \rangle - r_h(s,i,j) - P_h V_{h+1}^+(s,i,j) \right| \leq c' dH^2 \sqrt{\log \left( \frac{16dTH}{\delta \min\{1,\beta\}} \right) + \log(1 + c_2 + 2c_1)} \, \| \phi(s,i,j) \|_{\Sigma_{h,t}^{-1}}$$

for some universal constant $c'$ which is independent of $c_1, c_2$. Since $dT/\delta > 1$ and $c_1$ is a fixed universal constant from Lemma F.4, choosing a large enough $c_2 > c'$ we have

$$\left| \langle \theta_{h,t}^+, \phi(s,i,j) \rangle - r_h(s,i,j) - P_h V_{h+1}^+(s,i,j) \right| \leq c_2 dH^2 \sqrt{\log \left( \frac{16dTH}{\delta \min\{1,\beta\}} \right)} \, \| \phi(s,i,j) \|_{\Sigma_{h,t}^{-1}}.$$

This completes the proof of Lemma F.5.

### F.4.3. PROOF OF COROLLARY F.6

From the definition of $Q_{h,t}^+(s,i,j) = \Pi_h^+ \left( \langle \theta_{h,t}^+, \phi(s,i,j) \rangle + b_{h,t}^{\mathrm{sup}}(s,i,j) \right)$, under event $\mathcal{E}_7$, we have

$$\left| Q_{h,t}^+(s,i,j) - r_h(s,i,j) - P_h V_{h+1}^+(s,i,j) \right| = \left| \Pi_h^+ \left( \langle \theta_{h,t}^+, \phi(s,i,j) \rangle + b_{h,t}^{\mathrm{sup}}(s,i,j) \right) - r_h(s,i,j) - P_h V_{h+1}^+(s,i,j) \right|$$

$$\leq \left| \langle \theta_{h,t}^+, \phi(s,i,j) \rangle + b_{h,t}^{\mathrm{sup}}(s,i,j) - r_h(s,i,j) - P_h V_{h+1}^+(s,i,j) \right| \tag{117}$$

$$\leq b_{h,t}^{\mathrm{sup}}(s,i,j) + b_{h,t}(s,i,j) = 2b_{h,t}(s,i,j) + 2b_{h,t}^{\mathrm{mse}}(s,i,j) \tag{118}$$

Here eq. (117) follows since $r_h(s,i,j) + P_h V_{h+1}^+(s,i,j) \in [0, 3(H-h+1)^2]$ (Lemma F.11) and the projection operator $\Pi_h^+$ whose output $\Pi_h^+(\cdot) \in [0, 3(H-h+1)^2]$ is a non-expansive map. Eq. (118) follows from Lemma F.5. This concludes the proof.

### F.4.4. PROOF OF LEMMA F.7

Firstly we note that whenever $Q_h^+(s_h, i_h, j_h) = 3(H - h + 1)^2$ attains the maximum possible clipped value, the lemma holds trivially since $Q_h^{\mu'}(s_h, i_h, j_h) \leq (H - h + 1)^2$ (from Lemma F.11) and $\overline{Q}_h(s_h, i_h, j_h) \leq H - h + 1$ (from the design of the projection operator (20a)). By convention, we know eq. (75a) holds trivially when $h = H + 1$ Assume the statement is true for $h + 1$, then under $\mathcal{E}_6 \cap \mathcal{E}_7$,

$$
\begin{aligned}
& Q_h^+(s_h, i_h, j_h) - \overline{Q}_h(s_h, i_h, j_h) \\
& \overset{(22)}{=} \langle \theta_h^+, \phi(s_h, i_h, j_h) \rangle - r_h(s_h, i_h, j_h) - P_h V_{h+1}^+(s_h, i_h, j_h) + b_h(s_h, i_h, j_h) + 2b_h^{\mathrm{mse}}(s_h, i_h, j_h) \\
& \qquad + P_h \left( V_{h+1}^+(s_h, i_h, j_h) - \overline{V}_{h+1}(s_h, i_h, j_h) \right) - \overline{e}_h(s_h, i_h, j_h) \\
& \geq b_h^{\mathrm{mse}}(s_h, i_h, j_h) + P_h \left( V_{h+1}^+(s_h, i_h, j_h) - \overline{V}_{h+1}(s_h, i_h, j_h) \right) \\
& = b_h^{\mathrm{mse}}(s_h, i_h, j_h) +
\end{aligned}
\tag{119}
$$

$$
\begin{aligned}
& \underset{s_{h+1}|s_h,i_h,j_h}{\mathbb{E}} \left( \underset{\substack{i_{h+1} \sim \tilde{\mu}_{h+1}(\cdot|s_{h+1}) \\ j_{h+1} \sim \nu_{h+1}(\cdot|s_{h+1})}}{\mathbb{E}} \left[ Q_{h+1}^+(s_{h+1}, i_{h+1}, j_{h+1}) \right] - \beta \mathrm{KL}(\tilde{\mu}_{h+1}(\cdot|s_{h+1}) \| \mu_{\mathrm{ref},h+1}(\cdot|s_{h+1})) \right) \\
& \quad - \underset{s_{h+1}|s_h,i_h,j_h}{\mathbb{E}} \left( \underset{\substack{i_{h+1} \sim \mu_{h+1}(\cdot|s_{h+1}) \\ j_{h+1} \sim \nu_{h+1}(\cdot|s_{h+1})}}{\mathbb{E}} \left[ \overline{Q}_{h+1}(s_{h+1}, i_{h+1}, j_{h+1}) \right] - \beta \mathrm{KL}(\mu_{h+1}(\cdot|s_{h+1}) \| \mu_{\mathrm{ref},h+1}(\cdot|s_{h+1})) \right)
\end{aligned}
\tag{120}
$$

$$
\geq b_h^{\mathrm{mse}}(s_h, i_h, j_h) + \underset{s_{h+1}|s_h,i_h,j_h}{\mathbb{E}} \left( \underset{\substack{i_{h+1} \sim \mu_{h+1}(\cdot|s_{h+1}) \\ j_{h+1} \sim \nu_{h+1}(\cdot|s_{h+1})}}{\mathbb{E}} \left[ Q_{h+1}^+(s_{h+1}, i_{h+1}, j_{h+1}) - \overline{Q}_{h+1}(s_{h+1}, i_{h+1}, j_{h+1}) \right] \right) \geq 0,
\tag{121}
$$

where $\overline{e}_h$ is defined in (73), eq. (119) follows from Lemma F.4 and Lemma F.5, we omit the KL terms corresponding to the min player policy ($\nu_{h+1}(\cdot|s_{h+1})$) since it is the same for both $V_{h+1}^+$ and $\overline{V}_{h+1}$ in eq. (120), and we swap $\tilde{\mu}_{h+1}(\cdot|s_{h+1})$ by $\mu_{h+1}(\cdot|s_{h+1})$ in the first term of eq. (121) and the inequality follows from the optimality of the superoptimistic best response policy $\tilde{\mu}_{h+1}(\cdot|s_{h+1})$ under $Q_{h+1}^+(s_{h+1}, \cdot, \cdot)$ and $\nu_{h+1}$, and the induction hypothesis gives the last inequality. Using similar arguments, we have

$$
\begin{aligned}
& Q_h^+(s_h, i_h, j_h) - Q_h^{\mu'}(s_h, i_h, j_h) \\
& = \langle \theta_h^+, \phi(s_h, i_h, j_h) \rangle - r_h(s_h, i_h, j_h) - P_h V_{h+1}^+(s_h, i_h, j_h) + b_h(s_h, i_h, j_h) + 2b_h^{\mathrm{mse}}(s_h, i_h, j_h) \\
& \qquad + P_h \left( V_{h+1}^+(s_h, i_h, j_h) - V_{h+1}^{\mu'}(s_h, i_h, j_h) \right) \\
& \geq 2b_h^{\mathrm{mse}}(s_h, i_h, j_h)
\end{aligned}
$$

$$
\begin{aligned}
& + \underset{s_{h+1}|s_h,i_h,j_h}{\mathbb{E}} \left( \underset{\substack{i_{h+1} \sim \tilde{\mu}_{h+1}(\cdot|s_{h+1}) \\ j_{h+1} \sim \nu_{h+1}(\cdot|s_{h+1})}}{\mathbb{E}} \left[ Q_{h+1}^+(s_{h+1}, i_{h+1}, j_{h+1}) \right] - \beta \mathrm{KL}(\tilde{\mu}_{h+1}(\cdot|s_{h+1}) \| \mu_{\mathrm{ref},h+1}(\cdot|s_{h+1})) \right) \\
& \quad - \underset{s_{h+1}|s_h,i_h,j_h}{\mathbb{E}} \left( \underset{\substack{i_{h+1} \sim \mu_{h+1}'(\cdot|s_{h+1}) \\ j_{h+1} \sim \nu_{h+1}(\cdot|s_{h+1})}}{\mathbb{E}} \left[ Q_{h+1}^{\mu'}(s_{h+1}, i_{h+1}, j_{h+1}) \right] - \beta \mathrm{KL}(\mu_{h+1}'(\cdot|s_{h+1}) \| \mu_{\mathrm{ref},h+1}(\cdot|s_{h+1})) \right)
\end{aligned}
\tag{122}
$$

$$
\geq 2b_h^{\mathrm{mse}}(s_h, i_h, j_h) + \underset{s_{h+1}|s_h,i_h,j_h}{\mathbb{E}} \left( \underset{\substack{i_{h+1} \sim \mu_{h+1}'(\cdot|s_{h+1}) \\ j_{h+1} \sim \nu_{h+1}(\cdot|s_{h+1})}}{\mathbb{E}} \left[ Q_{h+1}^+(s_{h+1}, i_{h+1}, j_{h+1}) - Q_{h+1}^{\mu'}(s_{h+1}, i_{h+1}, j_{h+1}) \right] \right) \geq 0.
\tag{123}
$$

Here eq. (122) follows from Lemma F.5, Eq. (123) follows from the optimality of the superoptimistic best response policy $\tilde{\mu}_{h+1}(\cdot|s_{h+1})$ under $Q_{h+1}^+(s_{h+1}, \cdot, \cdot)$ and $\nu_{h+1}$ and the induction hypothesis implies the penultimate expression is positive.

### F.4.5. PROOF OF LEMMA F.8

From Lemma F.7 we have $Q_h^+(s_h, i_h, j_h) \geq \overline{Q}_h(s_h, i_h, j_h)$ and $Q_h^+(s_h, i_h, j_h) \geq Q_h^\mu(s_h, i_h, j_h)$. Note that whenever we have an underestimate of $Q^\mu$, i.e, $Q_h^\mu(s_h, i_h, j_h) \geq \overline{Q}_h(s_h, i_h, j_h)$ we have eq. (76) hold automatically even without the 2x multiplier hence we will only concern ourselves with the case where we overestimate $Q^\mu$, i.e., $Q_h^\mu(s_h, i_h, j_h) \leq \overline{Q}_h(s_h, i_h, j_h)$. We also note that when $Q_h^+(s_h, i_h, j_h) = 3(H - h + 1)^2$ attains the maximum possible clipped value the statement holds trivially again since $\overline{Q}_h(s_h, i_h, j_h) \leq (H - h + 1)$ (from the design of the projection operator (20a)) and $Q_h^\mu(s_h, i_h, j_h) \geq -(H - h + 1)^2 \ \forall \ (s_h, i_h, j_h)$ (from Lemma F.11). Since (by Lemma F.5)

$$\langle \theta_h^+, \phi(s_h, i_h, j_h) \rangle + b_h^{\mathrm{sup}}(s_h, i_h, j_h) \geq r_h(s_h, i_h, j_h) + P_h V_{h+1}^+(s_h, i_h, j_h) + 2b_h^{\mathrm{mse}}(s_h, i_h, j_h) > 0,$$

we only need to prove the equation in the overestimation case where

$$0 < Q_h^+(s_h, i_h, j_h) = \langle \theta_{h,t}^+, \phi(s, i, j) \rangle + b_{h,t}^{\mathrm{sup}}(s, i, j) < 3(H - h + 1)^2,$$

where eq. (76) (by Lemma F.7) is equivalent to

$$Q_h^+(s_h, i_h, j_h) - \overline{Q}_h(s_h, i_h, j_h) \geq \overline{Q}_h(s_h, i_h, j_h) - Q_h^\mu(s_h, i_h, j_h),$$

which we do via an induction argument. We know that eq. (76) holds trivially for $h = H + 1$. Assume it holds for $h + 1$. We will show that it also holds for $h$.

$$
\begin{aligned}
&Q_h^+(s_h, i_h, j_h) - \overline{Q}_h(s_h, i_h, j_h) \\
&= \langle \theta_h^+, \phi(s_h, i_h, j_h) \rangle - r_h(s_h, i_h, j_h) - P_h V_{h+1}^+(s_h, i_h, j_h) + b_h(s_h, i_h, j_h) + 2b_h^{\mathrm{mse}}(s_h, i_h, j_h) \\
&\quad + P_h \left( V_{h+1}^+(s_h, i_h, j_h) - \overline{V}_{h+1}(s_h, i_h, j_h) \right) - \overline{e}_h(s_h, i_h, j_h) \\
&\geq b_h^{\mathrm{mse}}(s_h, i_h, j_h) + P_h \left( V_{h+1}^+(s_h, i_h, j_h) - \overline{V}_{h+1}(s_h, i_h, j_h) \right) \quad\quad (124) \\
&= b_h^{\mathrm{mse}}(s_h, i_h, j_h)
\end{aligned}
$$

$$
+ \mathop{\mathbb{E}}_{s_{h+1}|s_h, i_h, j_h} \left( \mathop{\mathbb{E}}_{\substack{i_{h+1} \sim \tilde{\mu}_{h+1}(\cdot|s_{h+1}) \\ j_{h+1} \sim \nu_{h+1}(\cdot|s_{h+1})}} \left[ Q_{h+1}^+(s_{h+1}, i_{h+1}, j_{h+1}) \right] - \beta \mathrm{KL}(\tilde{\mu}_{h+1}(\cdot|s_{h+1}) \| \mu_{\mathrm{ref}, h+1}(\cdot|s_{h+1})) \right)
$$

$$
- \mathop{\mathbb{E}}_{s_{h+1}|s_h, i_h, j_h} \left( \mathop{\mathbb{E}}_{\substack{i_{h+1} \sim \mu_{h+1}(\cdot|s_{h+1}) \\ j_{h+1} \sim \nu_{h+1}(\cdot|s_{h+1})}} \left[ \overline{Q}_{h+1}(s_{h+1}, i_{h+1}, j_{h+1}) \right] - \beta \mathrm{KL}(\mu_{h+1}(\cdot|s_{h+1}) \| \mu_{\mathrm{ref}, h+1}(\cdot|s_{h+1})) \right)
$$

$$
\geq b_h^{\mathrm{mse}}(s_h, i_h, j_h) + \mathop{\mathbb{E}}_{s_{h+1}|s_h, i_h, j_h} \left( \mathop{\mathbb{E}}_{\substack{i_{h+1} \sim \mu_{h+1}(\cdot|s_{h+1}) \\ j_{h+1} \sim \nu_{h+1}(\cdot|s_{h+1})}} \left[ Q_{h+1}^+(s_{h+1}, i_{h+1}, j_{h+1}) - \overline{Q}_{h+1}(s_{h+1}, i_{h+1}, j_{h+1}) \right] \right) \quad (125)
$$

$$
\geq b_h^{\mathrm{mse}}(s_h, i_h, j_h) + \mathop{\mathbb{E}}_{s_{h+1}|s_h, i_h, j_h} \left( \mathop{\mathbb{E}}_{\substack{i_{h+1} \sim \mu_{h+1}(\cdot|s_{h+1}) \\ j_{h+1} \sim \nu_{h+1}(\cdot|s_{h+1})}} \left[ \overline{Q}_{h+1}(s_{h+1}, i_{h+1}, j_{h+1}) - Q_{h+1}^\mu(s_{h+1}, i_{h+1}, j_{h+1}) \right] \right) \quad (126)
$$

$$
= b_h^{\mathrm{mse}}(s_h, i_h, j_h) + \mathop{\mathbb{E}}_{s_{h+1}|s_h, i_h, j_h} \left( \overline{V}_{h+1}(s_{h+1}) - V_{h+1}^\mu(s_{h+1}) \right)
$$

$$
= b_h^{\mathrm{mse}}(s_h, i_h, j_h) + \overline{Q}_h(s_h, i_h, j_h) - Q_h^\mu(s_h, i_h, j_h) - \overline{e}(s_h, i_h, j_h)
$$

$$
\geq \overline{Q}_h(s_h, i_h, j_h) - Q_h^\mu(s_h, i_h, j_h). \quad (127)
$$

Here eq. (124) follows from Lemma F.4 and Lemma F.5. Eq. (125) swaps $\tilde{\mu}_{h+1}(\cdot|s_{h+1})$ by $\mu_{h+1}(\cdot|s_{h+1})$ in the first term and the inequality follows since the optimality of policy $\tilde{\mu}(\cdot|s_{h+1})$ under $Q^+(s_{h+1}, \cdot, \cdot)$ and eq. (126) follows from the induction hypothesis $\left( 2 \left| (Q_{h+1}^+(s, i, j) - \overline{Q}_{h+1}(s, i, j)) \right| \geq \left| Q_{h+1}^+(s, i, j) - Q_{h+1}^\mu(s, i, j) \right| \right)$ alongside the optimism lemma (Lemma F.7) implies $Q_{h+1}^+(s, i, j) - \overline{Q}_{h+1}(s, i, j) \geq \overline{Q}_{h+1}(s, i, j) - Q_{h+1}^\mu(s, i, j)$. Eq. (127) follows from Lemma F.4.

### F.5. Auxiliary Lemmas

**Lemma F.9.** *If $(\mu', \nu') := (\mu'_h, \nu'_h)_{h=1}^H$ is the Nash Equilibrium of a KL regularized Markov Game where $0 \le r'_h(s_h, i_h, j_h) \le 1$. Let $V_h^{\mu',\nu'}(s) := \mathbb{E}^{\mu',\nu'} \left[ \sum_{k=h}^H r'_k(s_k, i, j) - \beta \log \frac{\mu'_k(i|s_k)}{\mu_{ref,k}(i|s_k)} + \beta \log \frac{\nu'_k(j|s_k)}{\nu_{ref,k}(j|s_k)} \Big| s_h = s \right]$ and $Q_h^{\mu',\nu'}(s,i,j) := r'_h(s,i,j) + \mathbb{E}_{s' \sim P_h(\cdot|s,i,j)} \left[ V_{h+1}^{\mu',\nu'}(s') \right]$ be the value and Q functions under this game. Then $\forall (s,i,j) \in \mathcal{S} \times \mathcal{U} \times \mathcal{V}, h \in [H], \beta > 0$ we have*

$$Q_h^{\mu',\nu'}(s_h, i, j) \in [0, H - h + 1],$$
$$V_h^{\mu',\nu'}(s_h) \in [0, H - h + 1],$$
$$\beta KL\left(\mu'_h(\cdot|s_h)\|\mu_{ref,h}(\cdot|s_h)\right) \in [0, H - h + 1],$$
$$\beta KL\left(\nu'_h(\cdot|s_h)\|\nu_{ref,h}(\cdot|s_h)\right) \in [0, H - h + 1].$$

*Proof.* We prove the proposition using induction. The statement is true trivially for $h = H + 1$. Assume the statement is true for $h + 1$ then we have

$$Q_h^{\mu',\nu'}(s_h, i, j) = r'_h(s_h, i, j) + \mathbb{E}_{s' \sim P_h(\cdot|s_h,i,j)} \left[ V_{h+1}^{\mu',\nu'}(s') \right].$$

Since $V_{h+1}^{\mu',\nu'}(s') \in [0, H - h]$ and $r'_h(s_h, i, j) \in [0, 1]$, we have $Q_h^{\mu',\nu'}(s_h, i, j) \in [0, H - h + 1]$. In addition,

$$V_h^{\mu',\nu'}(s_h) = \mathbb{E}_{\substack{i \sim \mu'_h(\cdot|s_h) \\ j \sim \nu'_h(\cdot|s_h)}} \left[ Q_h^{\mu',\nu'}(s_h, i, j) \right] - \beta KL\left(\mu'_h(\cdot|s_h)\|\mu_{ref}(\cdot|s_h)\right) + \beta KL\left(\nu'_h(\cdot|s_h)\|\nu_{ref}(\cdot|s_h)\right).$$

Using the closed form expression for $\mu'_h(\cdot \mid s_h)$ we have

$$
\begin{aligned}
V_h^{\mu',\nu'}(s_h) &= \beta \log \left( \sum_i \mu_{ref,h}(i|s_h) \exp \left( \mathbb{E}_{j \sim \nu'(\cdot|s_h)} \left[ Q_h^{\mu',\nu'}(s_h, i, j) \right] / \beta \right) \right) + \beta KL\left(\nu'_h(\cdot|s_h)\|\nu_{ref,h}(\cdot|s_h)\right) \\
&\ge \mathbb{E}_{\substack{i \sim \mu_{ref,h}(\cdot|s_h) \\ j \sim \nu'_h(\cdot|s_h)}} \left[ Q_h^{\mu',\nu'}(s_h, i, j) \right] + \beta KL\left(\nu'_h(\cdot|s_h)\|\nu_{ref,h}(\cdot|s_h)\right) \\
&\ge 0.
\end{aligned}
$$

Here the second line uses $\log\left(\mathbb{E}[X]\right) \ge \mathbb{E}\left[\log(X)\right]$ (Jensen's inequality). Similarly, using the closed form expression for $\nu'_h(\cdot \mid s_h)$ we have

$$
\begin{aligned}
V_h^{\mu',\nu'}(s_h) &= -\beta \log \left( \sum_j \nu_{ref,h}(i|s_h) \exp \left( -\mathbb{E}_{i \sim \mu'_h(\cdot|s_h)} \left[ Q_h^{\mu',\nu'}(s_h, i, j) \right] / \beta \right) \right) - \beta KL\left(\mu'_h(\cdot|s_h)\|\mu_{ref,h}(\cdot|s_h)\right) \\
&\le \mathbb{E}_{\substack{i \sim \mu'(\cdot|s_h) \\ j \sim \nu_{ref,h}(\cdot|s_h)}} \left[ Q_h^{\mu',\nu'}(s_h, i, j) \right] - \beta KL\left(\mu'_h(\cdot|s_h)\|\mu_{ref,h}(\cdot|s_h)\right) \\
&\le H - h + 1.
\end{aligned}
$$

Lastly, note that since $\mu'_h(\cdot|s_h)$ is the Nash equilibrium point, for a fixed $\nu'_h$ we have

$$\mathbb{E}_{\substack{i \sim \mu'_h(\cdot|s_h) \\ j \sim \nu'_h(\cdot|s_h)}} \left[ Q_h^{\mu',\nu'}(s_h, i, j) \right] - \beta KL(\mu'_h(\cdot|s_h)\|\mu_{ref,h}(\cdot|s_h)) \ge \mathbb{E}_{\substack{i \sim \mu_{ref,h}(\cdot|s_h) \\ j \sim \nu'_h(\cdot|s_h)}} \left[ Q_h^{\mu',\nu'}(s_h, i, j) \right],$$

which gives

$$\beta KL(\mu'_h(\cdot|s_h)\|\mu_{ref,h}(\cdot|s_h)) \le \mathbb{E}_{\substack{i \sim \mu'_h(\cdot|s_h) \\ j \sim \nu'_h(\cdot|s_h)}} \left[ Q_h^{\mu',\nu'}(s_h, i, j) \right] - \mathbb{E}_{\substack{i \sim \mu_{ref,h}(\cdot|s_h) \\ j \sim \nu'_h(\cdot|s_h)}} \left[ Q_h^{\mu',\nu'}(s_h, i, j) \right] \le H - h + 1.$$

Similar argument using the min player can be used to obtain $\beta KL\left(\nu'_h(\cdot|s_h)\|\nu_{ref,h}(\cdot|s_h)\right) \in [0, H - h + 1]$. $\qquad \square$

**Lemma F.10.** *Let $(\mu_t, \nu_t) := (\mu_{h,t}, \nu_{h,t})_{h=1}^H$ be the estimated stagewise Nash Equilibrium policies of a KL regularized Matrix Game as defined in eq. (17) of Algorithm 2. Then $\forall (s, i, j) \in \mathcal{S} \times \mathcal{U} \times \mathcal{V}, h \in [H], \beta > 0$, we have*

$$\overline{Q}_{h,t}(s_h, i, j) \in [0, H - h + 1], \tag{128a}$$

$$\overline{V}_{h,t}(s_h) \in [0, H - h + 1], \tag{128b}$$

$$\beta KL\left(\mu_{h,t}(\cdot|s_h)\|\mu_{ref,h}(\cdot|s_h)\right) \in [0, H - h + 1], \tag{128c}$$

$$\beta KL\left(\nu_{h,t}(\cdot|s_h)\|\nu_{ref,h}(\cdot|s_h)\right) \in [0, H - h + 1]. \tag{128d}$$

*Proof.* We know $\overline{Q}_{h,t}(s_h, i, j) \in [0, H - h + 1]$ by the design of the projection operator $\Pi_h$. And since

$$(\mu_{h,t}(\cdot|s), \nu_{h,t}(\cdot|s)) \leftarrow \text{KL reg Nash Zero-sum}(\overline{Q}_{h,t}(s, \cdot, \cdot)),$$

using the same arguments as Lemma F.9 one can prove equations (128b)-(128d). $\qquad\square$

The next lemma provides upper and lower bounds on the functions $Q$ and $V$, which will be used in our analysis. We provide loose bounds on some of these terms for simplicity.

**Lemma F.11** (Range of Q, V functions). *Under the setting in Algorithm 2, for any $t \in [T]$, we have the following ranges for the Bellman target, value and Q functions for all $\forall (s, i, j) \in \mathcal{S} \times \mathcal{U} \times \mathcal{V}, h \in [H]$ and $\beta > 0$:*

$$V_{h+1,t}^+(s) \in [0, 3(H - h)^2 + (H - h)],$$

$$r_h(s, i, j) + P_h V_{h+1,t}^+(s, i, j) \in [0, 3(H - h + 1)^2],$$

$$Q_h^{\mu_t, \nu_t}(s, i, j) \in [-(H - h + 1)^2, (H - h + 1)^2],$$

$$V_h^{\mu_t, \nu_t}(s) \in [-(H - h + 1)^2, (H - h + 1)^2 + (H - h + 1)].$$

*We also have for any policy $\mu'$:*

$$Q_h^{\mu', \nu_t}(s, i, j) \leq (H - h + 1)^2,$$

$$V_h^{\mu', \nu_t}(s) \leq (H - h + 1)^2 + (H - h + 1).$$

*Proof.* Here we omit the subscript $t$ for notational simplicity while proving the first two statements. We have $Q_{h+1}^+(s, i, j) \in [0, 3(H - h)^2], \forall (s, i, j) \in \mathcal{S} \times \mathcal{U} \times \mathcal{V}, h \in [H]$ by definition of the projection operator $\Pi_h^+$ (see eq. (20b)). We have

$$V_{h+1}^+(s) = \mathbb{E}_{\substack{i \sim \tilde{\mu}_{h+1}(\cdot|s) \\ j \sim \nu_{h+1}(\cdot|s)}} \left[Q_{h+1}^+(s, i, j)\right] - \beta KL(\tilde{\mu}_{h+1}(\cdot|s)\|\mu_{\text{ref},h+1}(\cdot|s)) + \beta KL(\nu_{h+1}(\cdot|s)\|\nu_{\text{ref},h+1}(\cdot|s))$$

$$\leq \mathbb{E}_{\substack{i \sim \tilde{\mu}_{h+1}(\cdot|s) \\ j \sim \nu_{h+1}(\cdot|s)}} \left[Q_{h+1}^+(s, i, j)\right] + \beta KL(\nu_{h+1}(\cdot|s)\|\nu_{\text{ref},h+1}(\cdot|s)) \leq 3(H - h)^2 + (H - h), \tag{129}$$

where the last inequality follows from Lemma F.10 and (20). Thus $\forall (s, i, j) \in \mathcal{S} \times \mathcal{U} \times \mathcal{V}, h \in [H]$ we also have the target for the Bellman update

$$r_h(s, i, j) + P_h V_{h+1,t}^+(s, i, j) \leq 1 + 3(H - h)^2 + (H - h) \leq 3(H - h + 1)^2,$$

and

$$V_{h+1}^+(s) = \mathbb{E}_{\substack{i \sim \tilde{\mu}_{h+1}(\cdot|s) \\ j \sim \nu_{h+1}(\cdot|s)}} \left[Q_{h+1}^+(s, i, j)\right] - \beta KL(\tilde{\mu}_{h+1}(\cdot|s)\|\mu_{\text{ref},h+1}(\cdot|s)) + \beta KL(\nu_{h+1}(\cdot|s)\|\nu_{\text{ref},h+1}(\cdot|s))$$

$$\geq \mathbb{E}_{\substack{i \sim \mu_{\text{ref},h+1}(\cdot|s) \\ j \sim \nu_{h+1}(\cdot|s)}} \left[Q_{h+1}^+(s, i, j)\right] + \beta KL(\nu_{h+1}(\cdot|s)\|\nu_{\text{ref},h+1}(\cdot|s)) \geq 0.$$

Therefore, $\forall (s, i, j) \in \mathcal{S} \times \mathcal{U} \times \mathcal{V}, h \in [H]$, we have

$$r_h(s, i, j) + P_h V_{h+1,t}^+(s, i, j) \geq 0.$$

One can rewrite eq. (9) at step $h + 1$ as

$$V_{h+1}^{\mu',\nu_t}(s) = \mathbb{E}^{\mu',\nu_t}\left[\sum_{k=h+1}^{H} r_k(s_k,i,j) - \beta\mathrm{KL}\left(\mu'_k(\cdot|s_k)\|\mu_{\mathrm{ref},k}(\cdot|s_k)\right) + \beta\mathrm{KL}\left(\nu_{k,t}(\cdot|s_k)\|\nu_{\mathrm{ref},k}(\cdot|s_k)\right)\bigg|s_h = s\right]$$

$$\leq \mathbb{E}^{\mu',\nu_t}\left[\sum_{k=h+1}^{H} r_k(s_k,i,j) + \beta\mathrm{KL}\left(\nu_{k,t}(\cdot|s_k)\|\nu_{\mathrm{ref},k}(\cdot|s_k)\right)\bigg|s_h = s\right] \leq (H-h)^2 + (H-h), \qquad (130a)$$

where the last inequality is due to Lemma F.10. Thus for any policy $\mu'$ we have

$$Q_h^{\mu',\nu_t}(s,i,j) = r_h(s,i,j) + P_h V_{h+1}^{\mu',\nu_t}(s,i,j) \leq (H-h+1)^2.$$

Similarly, we have for any $s \in \mathcal{S}$, $h \in [H]$:

$$V_{h+1}^{\mu_t,\nu_t}(s) = \mathbb{E}^{\mu_t,\nu_t}\left[\sum_{k=h+1}^{H} r_k(s_k,i,j) - \beta\mathrm{KL}\left(\mu_{k,t}(\cdot|s_k)\|\mu_{\mathrm{ref},k}(\cdot|s_k)\right) + \beta\mathrm{KL}\left(\nu_{k,t}(\cdot|s_k)\|\nu_{\mathrm{ref},k}(\cdot|s_k)\right)\bigg|s_h = s\right]$$

$$\geq \mathbb{E}^{\mu_t,\nu_t}\left[\sum_{k=h+1}^{H} -\beta\mathrm{KL}\left(\mu_{k,t}(\cdot|s_k)\|\mu_{\mathrm{ref},k}(\cdot|s_k)\right)\bigg|s_h = s\right] \geq -(H-h)^2. \qquad (130b)$$

Since

$$Q_h^{\mu_t,\nu_t}(s,i,j) = r_h(s,i,j) + P_h V_{h+1}^{\mu_t,\nu_t}(s,i,j)$$

and $r_h(s,i,j) \in [0,1]$, using (130a) and (130b), we have

$$Q_h^{\mu_t,\nu_t}(s,i,j) \in [-(H-h+1)^2, (H-h+1)^2].$$

$\square$

This following lemma is a consequence of the linear MDP, similar results can be found in (Jin et al., 2020) (Lemma B.1) and (Xie et al., 2023) (Lemma 7).

**Lemma F.12** (Linearity of the Q function). *Let $(\mu_t,\nu_t) := (\mu_{h,t},\nu_{h,t})_{h=1}^{H}$ be the estimated stagewise Nash Equilibrium policies as defined in eq. (17) of Algorithm 2, then under the linear MDP (Assumption 3.1) there exist weights $\{\theta_h^{\mu_t,\nu_t}\}_{h=1}^{H}$ such that*

$$Q_h^{\mu_t,\nu_t}(s,i,j) = \langle\phi(s,i,j),\theta_h^{\mu_t,\nu_t}\rangle \qquad \text{and} \qquad \|\theta_h^{\mu_t,\nu_t}\| \leq 3H^2\sqrt{d}, \qquad \forall(s,i,j) \in \mathcal{S}\times\mathcal{U}\times\mathcal{V}, h \in [H].$$

*Similarly for the Nash equilibrium policy $(\mu^\star,\nu^\star) = (\mu_h^\star,\nu_h^\star)_{h=1}^{H}$ then there exist weights $\{\theta_h^{\mu^\star,\nu^\star}\}_{h=1}^{H}$ such that*

$$Q_h^{\mu^\star,\nu^\star}(s,i,j) = \left\langle\phi(s,i,j),\theta_h^{\mu^\star,\nu^\star}\right\rangle \qquad \text{and} \qquad \left\|\theta_h^{\mu^\star,\nu^\star}\right\| \leq 2H\sqrt{d}, \qquad \forall(s,i,j) \in \mathcal{S}\times\mathcal{U}\times\mathcal{V}, h \in [H].$$

*Proof.* From the Bellman eq. (10) we have

$$Q_h^{\mu_t,\nu_t}(s,i,j) := r_h(s,i,j) + \mathbb{E}_{s'\sim P_h(\cdot|s_h,i,j)}\left[V_{h+1}^{\mu_t,\nu_t}(s')\right].$$

From the definition of linear MDP (c.f. Assumption 3.1) we know that can set

$$\theta_h^{\mu_t,\nu_t} = \omega_h + \int V_{h+1}^{\mu_t,\nu_t}(s')d\psi(s') \leq 3H^2\sqrt{d}.$$

since $\|\omega_h\| \leq \sqrt{d}$ and $\left\|\int V_{h+1}^{\mu_t,\nu_t}(s')d\psi(s')\right\| \leq 2H^2\sqrt{d}$ (from Lemma F.11). Similarly, we have

$$\theta_h^{\mu^\star,\nu^\star} = \omega_h + \int V_{h+1}^{\mu^\star,\nu^\star}(s')d\psi(s'). \qquad (131)$$

Using $\left\|\int V_{h+1}^{\mu^\star,\nu^\star}(s')d\psi(s')\right\| \leq H\sqrt{d}$ (from Lemma F.9) we have $\|\theta_h^{\mu^\star,\nu^\star}\| \leq 2H\sqrt{d}$. $\square$

Note that the proof of Proposition 3.2 is contained in the proofs of Lemma F.9 and F.12. The following lemma bounds the $L_2$ norms of the estimated parameters ($\overline{\theta}_{h,t}$ and $\theta_{h,t}^+$) and is similar to (Jin et al., 2020) (Lemma B.2) and (Xie et al., 2023) (Lemma 8)

**Lemma F.13** ($L_2$ norm bounds). *For all $h \in [H], t \in [T]$, we have the following bounds on the $L_2$ norms:*

$$\|\overline{\theta}_{h,t}\| \leq 2H\sqrt{2dt/\lambda} \quad and \quad \|\theta_{h,t}^+\| \leq 4H^2\sqrt{2dt/\lambda}.$$

*Proof.* We have

$$\max_{\|\mathbf{x}\|=1} \left|\mathbf{x}^\top \overline{\theta}_{h,t}\right| = \left|\mathbf{x}^\top \Sigma_{h,t}^{-1} \sum_{\tau \in \mathcal{D}_{t-1}} \phi_{h,\tau} \left[r_{h,\tau} + \overline{V}_{h+1,t}(s_{h+1}^\tau)\right]\right|$$

$$\leq 2H \sum_{\tau \in \mathcal{D}_{t-1}} \left|\mathbf{x}^\top \Sigma_{h,t}^{-1} \phi_{h,\tau}\right| \leq 2H \sum_{\tau \in \mathcal{D}_{t-1}} |\mathbf{x}|_{\Sigma_{h,t}^{-1}} |\phi_{h,\tau}|_{\Sigma_{h,t}^{-1}}$$

$$\leq 2H \sqrt{\left[\sum_{\tau \in \mathcal{D}_{t-1}} \mathbf{x}^\top \Sigma_{h,t}^{-1} \mathbf{x}\right]\left[\sum_{\tau \in \mathcal{D}_{t-1}} \phi_{h,\tau}^\top \Sigma_{h,t}^{-1} \phi_{h,\tau}\right]} \leq 2H\sqrt{2dt/\lambda}.$$

where the first inequality follows from Lemma F.10 and the last inequality follows from Lemma D.7. Similarly, we have

$$\max_{\|\mathbf{x}\|=1} \left|\mathbf{x}^\top \theta_{h,t}^+\right| = \left|\mathbf{x}^\top \Sigma_{h,t}^{-1} \sum_{\tau \in \mathcal{D}_{t-1}} \phi_{h,\tau} \left[r_{h,\tau} + V_{h+1,t}^+(s_{h+1}^\tau)\right]\right|$$

$$\leq 4H^2 \sqrt{\left[\sum_{\tau \in \mathcal{D}_{t-1}} \mathbf{x}^\top \Sigma_{h,t}^{-1} \mathbf{x}\right]\left[\sum_{\tau \in \mathcal{D}_{t-1}} \phi_{h,\tau}^\top \Sigma_{h,t}^{-1} \phi_{h,\tau}\right]} \leq 4H^2\sqrt{2dt/\lambda}.$$

here the first inequality follows from Lemma F.11 and the last inequality follows from Lemma D.7. $\square$

The following lemma provides an upper bound on the covering number of the value functions induced by the $Q$-function estimates in Algorithm 2 when $\beta > 0$. The original result for the unregularized setting appears in (Jin et al., 2020) (Lemma D.6).

**Lemma F.14** (Covering number of the MSE Value function class in Algorithm 2). *For some $\beta > 0$, let $\mathcal{V}^{mse}$ denote the function class on the state space $\mathcal{S}$ with the parametric form*

$$V(s) := \mathbb{E}_{\substack{i \sim \mu_Q(\cdot|s) \\ j \sim \nu_Q(\cdot|s)}} [Q(s,i,j)] - \beta KL\left(\mu_Q(\cdot|s)\|\mu_{ref}(\cdot|s)\right) + \beta KL\left(\nu_Q(\cdot|s)\|\nu_{ref}(\cdot|s)\right),$$

*for fixed policies $\nu_{ref}, \mu_{ref}$ (we omit the index $h \in [H]$), where $Q(s,i,j) \in \mathcal{Q}(s,i,j)$ and $\mu_Q(\cdot|s)$ and $\nu_Q(\cdot|s)$ being the Nash equilibrium policies for the KL regularized game with the payoff matrix $Q(s,\cdot,\cdot)$. $\mathcal{Q}$ is a function class on the space $\mathcal{S} \times \mathcal{U} \times \mathcal{V}$ with the parametric form*

$$Q(s,i,j) = \Pi_{(b_2,B_2)}\left(\boldsymbol{\theta}^\top \phi(s,i,j) + \eta\sqrt{\phi(s,i,j)^\top \boldsymbol{\Sigma}^{-1} \phi(s,i,j)}\right)$$

*with function parameters $\|\boldsymbol{\theta}\| \leq L$, $\lambda_{\min}(\boldsymbol{\Sigma}) \geq \lambda$ and $0 \leq \eta \leq B_3$, and we define $\Pi_{(b_2,B_2)}(\cdot) = \min\{\max\{\cdot, b_2\}, B_2\}$ where $b_2 \leq B_2$ are function class parameters (fixed for a given $h \in [H]$). Then the covering number of the class $\mathcal{V}^{mse}$ w.r.t the $L_\infty$-norm $dist(V_1, V_2) = \sup_s |V_1(s) - V_2(s)|$ can be upper bounded as*

$$\log \mathcal{N}_\varepsilon^{mse} \leq d\log(1 + 4L/\varepsilon) + d^2 \log\left(1 + 8d^{1/2}B_3^2/(\lambda\varepsilon^2)\right). \tag{132}$$

*Proof.* We can reparameterize any function $Q \in \mathcal{Q}$ as follows:

$$Q(s, i, j) = \Pi_{(b_2, B_2)} \left( \boldsymbol{\theta}^\top \phi(s, i, j) + \sqrt{\phi(s, i, j)^\top \mathbf{A} \phi(s, i, j)} \right),$$

for the positive semi-definite matrix $\mathbf{A} = \eta^2 \boldsymbol{\Sigma}^{-1}$ with the spectral norm $\|\mathbf{A}\| \leq B_3^2/\lambda$ (which implies $\|\mathbf{A}\|_F \leq d^{1/2} B_3^2/\lambda$) Let $V_1(\cdot)$ and $V_2(\cdot)$ be the value functions induced by $Q_1(\cdot, \cdot, \cdot)$ (parameterized by $\boldsymbol{\theta_1}$, $\mathbf{A}_1$, optimal policies $\mu_1, \nu_1$) and $Q_2(\cdot, \cdot, \cdot)$ (parameterized by $\boldsymbol{\theta_2}$, $\mathbf{A_2}$, optimal policies $\mu_2, \nu_2$) respectively. Define

$$V_1(s, \mu, \nu) := \mathbb{E}_{\substack{i \sim \mu(\cdot|s) \\ j \sim \nu(\cdot|s)}} [Q_1(s, i, j)] - \beta \mathrm{KL} \left( \mu(\cdot|s) \| \mu_{\mathrm{ref}}(\cdot|s) \right) + \beta \mathrm{KL} \left( \nu(\cdot|s) \| \nu_{\mathrm{ref}}(\cdot|s) \right),$$

then we have $V_1(s, \mu_1, \nu_1) = V_1(s)$, similarly define $V_2(s, \mu, \nu)$, then we have $\qquad\qquad\square$

$$\begin{aligned}
\mathrm{dist}(V_1, V_2) &= \sup_s |V_1(s) - V_2(s)| \\
&= \sup_s |V_1(s, \mu_1, \nu_1) - V_2(s, \mu_2, \nu_2)| \\
&= \sup_s \max \left\{ V_1(s, \mu_1, \nu_1) - V_2(s, \mu_2, \nu_2), V_2(s, \mu_2, \nu_2) - V_1(s, \mu_1, \nu_1) \right\} \\
&\leq \sup_s \max \left\{ |V_1(s, \mu_1, \nu_2) - V_2(s, \mu_1, \nu_2)|, |V_2(s, \mu_2, \nu_1) - V_1(s, \mu_2, \nu_1)| \right\} \qquad (133) \\
&= \sup_s \max \left\{ \left| \mathbb{E}_{\substack{i \sim \mu_1(\cdot|s) \\ j \sim \nu_2(\cdot|s)}} [Q_1(s, i, j) - Q_2(s, i, j)] \right|, \left| \mathbb{E}_{\substack{i \sim \mu_2(\cdot|s) \\ j \sim \nu_1(\cdot|s)}} [Q_2(s, i, j) - Q_1(s, i, j)] \right| \right\} \\
&\leq \sup_{s, i, j} |Q_1(s, i, j) - Q_2(s, i, j)| \qquad (134) \\
&\leq \sup_{\|\phi\| \leq 1} \left| \left( \boldsymbol{\theta_1}^\top \phi + \sqrt{\phi^\top \mathbf{A_1} \phi} \right) - \left( \boldsymbol{\theta_2}^\top \phi + \sqrt{\phi^\top \mathbf{A_2} \phi} \right) \right| \qquad (135) \\
&\leq \|\boldsymbol{\theta_1} - \boldsymbol{\theta_2}\| + \sqrt{\|\mathbf{A_1} - \mathbf{A_2}\|} \\
&\leq \|\boldsymbol{\theta_1} - \boldsymbol{\theta_2}\| + \sqrt{\|\mathbf{A_1} - \mathbf{A_2}\|_F},
\end{aligned}$$

where eq. (133) follows from the fact $V_1(s, \mu_1, \nu_1) \leq V_1(s, \mu_1, \nu') \; \forall \nu'$ and $V_2(s, \mu_2, \nu_2) \geq V_2(s, \mu', \nu_2) \; \forall \mu'$ from the properties of Nash equilibrium resulting in

$$V_1(s, \mu_1, \nu_1) - V_2(s, \mu_2, \nu_2) \leq V_1(s, \mu_1, \nu_2) - V_2(s, \mu_1, \nu_2) \leq |V_1(s, \mu_1, \nu_2) - V_2(s, \mu_1, \nu_2)|,$$

when $V_1(s) \geq V_2(s)$. The other term is similarly obtained when $V_1(s) \leq V_2(s)$ and eq. (135) follows since $\Pi_{(b_2, B_2)}(\cdot) = \min\{\max\{\cdot, b_2\}, B_2\}$ is non-expansive, the penultimate line uses the fact

$$|\sqrt{x} - \sqrt{y}| \leq \sqrt{|x - y|},$$

giving us

$$\sup_{\|\phi\| \leq 1} \left| \sqrt{\phi^\top \mathbf{A_1} \phi} - \sqrt{\phi^\top \mathbf{A_2} \phi} \right| \leq \sup_{\|\phi\| \leq 1} \sqrt{|\phi^\top (\mathbf{A_1} - \mathbf{A_2}) \phi|} \leq \sqrt{\|\mathbf{A_1} - \mathbf{A_2}\|}.$$

Applying Lemma D.1 to upper bound the cardinality of the $\mathcal{C}_\theta$ : the $\varepsilon/2$ cover of $\{\boldsymbol{\theta} \in \mathbb{R}^d | \|\boldsymbol{\theta}\| \leq L\}$ and $\mathcal{C}_A$ : the $\varepsilon^2/4$ cover of $\{\mathbf{A} \in \mathbb{R}^{d \times d} \mid \|\mathbf{A}\|_F \leq d^{1/2} B_3^2 \lambda^{-1}\}$ with respect to the Frobenius norm, we obtain

$$\log \mathcal{N}_\varepsilon^{\mathrm{mse}} \leq \log |\mathcal{C}_\theta| + \log |\mathcal{C}_A| \leq d \log(1 + 4L/\varepsilon) + d^2 \log \left[ 1 + 8 d^{1/2} B_3^2 / (\lambda \varepsilon^2) \right]. \qquad (136)$$

Note that for the MSE function class in algorithm 2 has $\eta = 0$, we presented the loose bound above to reuse the steps in a later proof.

**Lemma F.15** (Covering number of the superoptimistic value function class in Algorithm 2)**.** *For some $\beta > 0$, let $\mathcal{V}^{sup}$ denote the function class on the state space $\mathcal{S}$ with the parametric form*

$$V(s) := \mathbb{E}_{\substack{i \sim \tilde{\mu}(\cdot|s) \\ j \sim \nu_{\overline{Q}}(\cdot|s)}} \left[ Q^+(s,i,j) \right] - \beta KL \left( \tilde{\mu}(\cdot|s) \| \mu_{ref}(\cdot|s) \right) + \beta KL \left( \nu_{\overline{Q}}(\cdot|s) \| \nu_{ref}(\cdot|s) \right),$$

*for fixed policies $\nu_{ref}, \mu_{ref}$ (we omit the index $h \in [H]$), where $\overline{Q}(s,i,j) \in \overline{\mathcal{Q}}(s,i,j)$, $Q^+(s,i,j) \in \mathcal{Q}^+(s,i,j)$, $\nu_{\overline{Q}}(\cdot|s)$ is the min player KL regularized Nash equilibrium policy for a payoff matrix $\overline{Q}(s,\cdot,\cdot)$ and $\tilde{\mu}(\cdot|s)$ is the max player's KL regularized best response to min player policy $\nu_{\overline{Q}}(\cdot|s)$ under the payoff matrix $Q^+(s,\cdot,\cdot)$. Hence $V$ is fully parameterized by $Q^+, \overline{Q}$. The functions $Q^+$ and $\overline{Q}$ have the following parametric forms:*

$$\overline{Q}(s,i,j) = \Pi_{(b_2, B_2)} \left( \overline{\boldsymbol{\theta}}^{\top} \phi(s,i,j) \right),$$

$$Q^+(s,i,j) = \Pi_{(b_2^+, B_2^+)} \left( \left( \boldsymbol{\theta}^+ \right)^{\top} \phi(s,i,j) + \eta \sqrt{\phi(s,i,j)^{\top} \boldsymbol{\Sigma}^{-1} \phi(s,i,j)} \right),$$

*with function parameters $\|\overline{\boldsymbol{\theta}}\| \leq L$, $\|\boldsymbol{\theta}^+\| \leq L^+$, $\lambda_{\min}(\boldsymbol{\Sigma}) \geq \lambda$ and $0 \leq \eta \leq B_3$, and we define $\Pi_{(a,b)}(\cdot) = \min\{\max\{\cdot, a\}, b\}$ and $b_2 \leq B_2$, $b_2^+ < B_2^+$ are function class parameters (fixed for a given $h \in [H]$). Then the covering number of the class $\mathcal{V}^{sup}$ w.r.t the $L_\infty$-norm $dist(V_1, V_2) = \sup_s |V_1(s) - V_2(s)|$ can be upper bounded as*

$$\mathcal{N}_\varepsilon^{sup} \leq d \log(1 + 12L^+/\varepsilon) + d^2 \log \left( 1 + 72 d^{1/2}(B_3)^2 / (\lambda \varepsilon^2) \right) + d \log(1 + 24L/\varepsilon)$$
$$+ d \log(1 + 144L(B_2^+ + B_2 - b_2)^2/\varepsilon^2 \beta)$$

*Proof.* We can reparameterize any function $Q^+ \in \mathcal{Q}^+$ as follows:

$$Q^+(s,i,j) = \Pi_{(b_2^+, B_2^+)} \left( \left( \boldsymbol{\theta}^+ \right)^{\top} \phi(s,i,j) + \sqrt{\phi(s,i,j)^{\top} \mathbf{A} \phi(s,i,j)} \right),$$

for the positive semi-definite matrix $A = \eta^2 \Sigma^{-1}$ with the spectral norm $\|\mathbf{A}\| \leq B_3^2/\lambda$ (which implies $\|\mathbf{A}\|_F \leq d^{1/2} B_3^2/\lambda$) Let $V_1(\cdot)$ and $V_2(\cdot)$ be the value functions induced by $Q_1^+(\cdot,\cdot,\cdot)$, $\overline{Q}_1(\cdot,\cdot,\cdot)$ (parameterized by $\overline{\boldsymbol{\theta}}_1, \boldsymbol{\theta}_1^+, \mathbf{A}_1$, induced policies $\tilde{\mu}_1$ and $\nu_1$) and $Q_2^+(\cdot,\cdot,\cdot)$, $\overline{Q}_2(\cdot,\cdot,\cdot)$ (parameterized by $\overline{\boldsymbol{\theta}}_2, \boldsymbol{\theta}_2^+, \mathbf{A}_2$, induced policies $\tilde{\mu}_2$ and $\nu_2$) respectively. Define

$$V_1(s,\mu,\nu) := \mathbb{E}_{\substack{i \sim \mu(\cdot|s) \\ j \sim \nu(\cdot|s)}} \left[ Q_1^+(s,i,j) \right] - \beta KL \left( \mu(\cdot|s) \| \mu_{ref}(\cdot|s) \right) + \beta KL \left( \nu(\cdot|s) \| \nu_{ref}(\cdot|s) \right),$$

then we have $V_1(s, \tilde{\mu}_1, \nu_1) = V_1(s)$, similarly define $V_2(s,\mu,\nu)$, then we have

$$\begin{aligned}
dist(V_1, V_2) &= \sup_s |V_1(s) - V_2(s)| \\
&= \sup_s |V_1(s, \tilde{\mu}_1, \nu_1) - V_2(s, \tilde{\mu}_2, \nu_2)| \\
&= \sup_s \max \left\{ V_1(s, \tilde{\mu}_1, \nu_1) - V_2(s, \tilde{\mu}_2, \nu_2), V_2(s, \tilde{\mu}_2, \nu_2) - V_1(s, \tilde{\mu}_1, \nu_1) \right\} \\
&\leq \sup_s \max \left\{ |V_1(s, \tilde{\mu}_1, \nu_1) - V_2(s, \tilde{\mu}_1, \nu_2)|, |V_2(s, \tilde{\mu}_2, \nu_2) - V_1(s, \tilde{\mu}_2, \nu_1)| \right\} \qquad (137) \\
&\leq \sup_s \max \left\{ |V_1(s, \tilde{\mu}_1, \nu_1) - V_2(s, \tilde{\mu}_1, \nu_2)|, |V_2(s, \tilde{\mu}_2, \nu_2) - V_1(s, \tilde{\mu}_2, \nu_1)| \right\} \\
&\leq \sup_s \left\{ \underbrace{\left| \mathbb{E}_{\substack{i \sim \tilde{\mu}_1(\cdot|s) \\ j \sim \nu_1(\cdot|s)}} \left[ Q_1^+(s,i,j) \right] - \mathbb{E}_{\substack{i \sim \tilde{\mu}_1(\cdot|s) \\ j \sim \nu_2(\cdot|s)}} \left[ Q_2^+(s,i,j) \right] \right|, \left| \mathbb{E}_{\substack{i \sim \tilde{\mu}_2(\cdot|s) \\ j \sim \nu_1(\cdot|s)}} \left[ Q_1^+(s,i,j) \right] - \mathbb{E}_{\substack{i \sim \tilde{\mu}_2(\cdot|s) \\ j \sim \nu_2(\cdot|s)}} \left[ Q_2^+(s,i,j) \right] \right|}_{T_Q(s)} \right. \\
&\quad \left. + \beta \underbrace{\left| KL(\nu_1(\cdot|s) \| \nu_{ref}(\cdot|s)) - KL(\nu_2(\cdot|s) \| \nu_{ref}(\cdot|s)) \right|}_{T_{KL}(s)} \right\}
\end{aligned}$$

$$\leq \sup_s T_Q(s) + \sup_s T_{\text{KL}}(s) \tag{138}$$

where we bound $T_Q(s)$ as follows. Consider the term

$$\left| \underset{\substack{i\sim\tilde{\mu}_1(\cdot|s)\\j\sim\nu_1(\cdot|s)}}{\mathbb{E}} \left[Q_1^+(s,i,j)\right] - \underset{\substack{i\sim\tilde{\mu}_1(\cdot|s)\\j\sim\nu_2(\cdot|s)}}{\mathbb{E}} \left[Q_2^+(s,i,j)\right] \right| \tag{139}$$

$$\leq \left| \underset{\substack{i\sim\tilde{\mu}_1(\cdot|s)\\j\sim\nu_1(\cdot|s)}}{\mathbb{E}} \left[Q_1^+(s,i,j)\right] - \underset{\substack{i\sim\tilde{\mu}_1(\cdot|s)\\j\sim\nu_1(\cdot|s)}}{\mathbb{E}} \left[Q_2^+(s,i,j)\right] \right| + \left| \underset{\substack{i\sim\tilde{\mu}_1(\cdot|s)\\j\sim\nu_1(\cdot|s)}}{\mathbb{E}} \left[Q_2^+(s,i,j)\right] - \underset{\substack{i\sim\tilde{\mu}_1(\cdot|s)\\j\sim\nu_2(\cdot|s)}}{\mathbb{E}} \left[Q_2^+(s,i,j)\right] \right|$$

$$\leq \sup_{i,j} |Q_1^+(s,i,j) - Q_2^+(s,i,j)| + \|\mathbb{E}_{i\sim\tilde{\mu}_1}[Q_2^+(s,i,\cdot)]\|_\infty \|\nu_1(\cdot|s) - \nu_2(\cdot|s)\|_1 \tag{140}$$

$$\leq \sup_{i,j} |Q_1^+(s,i,j) - Q_2^+(s,i,j)| + B_2^+ \|\nu_1(\cdot|s) - \nu_2(\cdot|s)\|_1 \tag{141}$$

Using the same argument (eq. 139-141) for the other term in $T_Q$ we obtain the same bound and thus

$$\sup_s T_Q(s) \leq \sup_{s,i,j} |Q_1^+(s,i,j) - Q_2^+(s,i,j)| + \sup_s B_2^+ \|\nu_1(\cdot|s) - \nu_2(\cdot|s)\|_1 \tag{142}$$

Since $\nu_1(\cdot|s)$ and $\nu_2(\cdot|s)$ are the min-player NE policies under the payoff matrices $\overline{Q}_1$ and $\overline{Q}_2$ respectively and $\overline{Q}_1, \overline{Q}_2 \in [b_2, B_2]$, using lemma F.17, equations (138) and (142) we have

$$\text{dist}(V_1, V_2) \leq \sup_{s,i,j} |Q_1^+(s,i,j) - Q_2^+(s,i,j)| + 2\sup_s B_2^+ \sqrt{\frac{\sup_{i,j} |\overline{Q}_1(s,i,j) - \overline{Q}_2(s,i,j)|}{\beta}}$$

$$+ 2\sup_s \left( \sup_{i,j} |\overline{Q}_1(s,i,j) - \overline{Q}_2(s,i,j)| + (B_2 - b_2)\sqrt{\frac{\sup_{i,j} |\overline{Q}_1(s,i,j) - \overline{Q}_2(s,i,j)|}{\beta}} \right)$$

$$\leq \sup_{s,i,j} |Q_1^+(s,i,j) - Q_2^+(s,i,j)| + 2\sup_{s,i,j} |\overline{Q}_1(s,i,j) - \overline{Q}_2(s,i,j)|$$

$$+ 2(B_2^+ + B_2 - b_2)\sqrt{\frac{\sup_{s,i,j} |\overline{Q}_1(s,i,j) - \overline{Q}_2(s,i,j)|}{\beta}} \tag{143}$$

Thus to get an $\varepsilon$ cover we need an $\varepsilon/3$-cover for $\sup_{s,i,j} |Q_1^+(s,i,j) - Q_2^+(s,i,j)|$ ($\mathcal{N}_{\varepsilon/3}^+$) and a $\min\left\{\frac{\varepsilon}{6}, \frac{\beta\varepsilon^2}{36(B_2^+ + B_2 - b_2)^2}\right\}$-cover for $\sup_{s,i,j} |\overline{Q}_1(s,i,j) - \overline{Q}_2(s,i,j)|$ ($\overline{\mathcal{N}}_{\varepsilon/3}$). Using the same steps as the equations eqs (134)-(136) we have

$$\log \mathcal{N}_{\varepsilon/3}^+ \leq d\log(1 + 12L^+/\varepsilon) + d^2 \log\left(1 + 72d^{1/2}(B_3)^2/(\lambda\varepsilon^2)\right)$$

$$\log \overline{\mathcal{N}}_{\varepsilon/3} \leq \min\left\{d\log(1 + 144L(B_2^+ + B_2 - b_2)^2/\varepsilon^2\beta), \; d\log(1 + 24L/\varepsilon)\right\}$$

$$\leq d\log(1 + 144L(B_2^+ + B_2 - b_2)^2/\varepsilon^2\beta) + d\log(1 + 24L/\varepsilon)$$

The last line follows since $B_3 = 0$ for the function class $\overline{\mathcal{Q}}$ and thus we have

$$\mathcal{N}_\varepsilon^{\text{sup}} \leq d\log(1 + 12L^+/\varepsilon) + d^2 \log\left(1 + 72d^{1/2}(B_3)^2/(\lambda\varepsilon^2)\right) + d\log(1 + 24L/\varepsilon)$$

$$+ d\log(1 + 144L(B_2^+ + B_2 - b_2)^2/\varepsilon^2\beta)$$

$\square$

**Lemma F.16** (KL three-point identity). *Let $\beta > 0$ and let $\nu_{ref} \in \Delta_n$ and $\mu_{ref} \in \Delta_m$ have full support. Fix vectors $c \in \mathbb{R}^n$, $d \in \mathbb{R}^m$ and define, for $\mu \in \Delta_m$, $\nu \in \Delta_n$,*

$$F(\nu) := \langle c, \nu\rangle + \beta \, KL(\nu\|\nu_{ref}), \qquad \text{and} \qquad G(\mu) := \langle d, \mu\rangle - \beta \, KL(\mu\|\mu_{ref}).$$

*Let $\mu^\star$ and $\nu^\star$ be the maximizer and minimizer of $G$ and $F$ respectively. Then for all $\mu \in \Delta_m$, $\nu \in \Delta_n$, we have*

$$F(\nu) = F(\nu^\star) + \beta \, KL(\nu\|\nu^\star), \qquad \text{and} \qquad G(\mu) = G(\mu^\star) - \beta \, KL(\mu\|\mu^\star).$$

*Proof.* The maximizer $\mu^\star \in \arg\max_{\mu \in \Delta_m} G(\mu)$ is unique and satisfies

$$\mu^\star(i) \;=\; \frac{\mu_{\text{ref}}(i) \exp\big(d(i)/\beta\big)}{\sum_{k=1}^m \mu_{\text{ref}}(k) \exp\big(d(k)/\beta\big)} \qquad \forall i \in [m].$$

Define $Z := \sum_{k=1}^m \mu_{\text{ref}}(k) \exp(d(k)/\beta)$, using $d(i) = \beta \log \frac{\mu^\star(i)}{\mu_{\text{ref}}(i)} + \beta \log Z$ we have

$$
\begin{aligned}
G(\mu) - G(\mu^\star) &= \langle d, \mu - \mu^\star \rangle - \beta \sum_{i=1}^m \Big( \mu(i) \log \frac{\mu(i)}{\mu_{\text{ref}}(i)} - \mu^\star(i) \log \frac{\mu^\star(i)}{\mu_{\text{ref}}(i)} \Big). \\
&= \beta \sum_{i=1}^m (\mu(i) - \mu^\star(i)) \log \frac{\mu^\star(i)}{\mu_{\text{ref}}(i)} + \beta \log Z \sum_{i=1}^m (\mu(i) - \mu^\star(i)) - \beta \sum_{i=1}^m \Big( \mu(i) \log \frac{\mu(i)}{\mu_{\text{ref}}(i)} - \mu^\star(i) \log \frac{\mu^\star(i)}{\mu_{\text{ref}}(i)} \Big) \\
&= \beta \sum_{i=1}^m (\mu(i) - \mu^\star(i)) \log \frac{\mu^\star(i)}{\mu_{\text{ref}}(i)} - \beta \sum_{i=1}^m \Big( \mu(i) \log \frac{\mu(i)}{\mu_{\text{ref}}(i)} - \mu^\star(i) \log \frac{\mu^\star(i)}{\mu_{\text{ref}}(i)} \Big) \\
&= -\beta \sum_{i=1}^m \mu(i) \log \frac{\mu(i)}{\mu^\star(i)} \;=\; -\beta \, \text{KL}(\mu \| \mu^\star).
\end{aligned}
\tag{144}
$$

The $\log Z$ term in eq (144) vanishes since $\sum_i \mu(i) = \sum_i \mu^\star(i) = 1$. Now similarly the minimizer $\nu^\star \in \arg\min_{\nu \in \Delta_n} F(\nu)$ is unique and satisfies

$$\nu^\star(j) \;=\; \frac{\nu_{\text{ref}}(j) \exp\big(-c(j)/\beta\big)}{\sum_{k=1}^n \nu_{\text{ref}}(k) \exp\big(-c(k)/\beta\big)} \qquad \forall j \in [n].$$

Define $Z := \sum_{k=1}^n \nu_{\text{ref}}(k) \exp(-c(k)/\beta)$, now using $c(j) = -\beta \log \frac{\nu^\star(j)}{\nu_{\text{ref}}(j)} - \beta \log Z$.

$$
\begin{aligned}
F(\nu) - F(\nu^\star) &= \langle c, \nu - \nu^\star \rangle + \beta \sum_{j=1}^n \Big( \nu(j) \log \frac{\nu(j)}{\nu_{\text{ref}}(j)} - \nu^\star(j) \log \frac{\nu^\star(j)}{\nu_{\text{ref}}(j)} \Big). \\
&= -\beta \sum_{j=1}^n (\nu(j) - \nu^\star(j)) \log \frac{\nu^\star(j)}{\nu_{\text{ref}}(j)} - \beta \log Z \sum_{j=1}^n (\nu(j) - \nu^\star(j)) + \beta \sum_{j=1}^n \Big( \nu(j) \log \frac{\nu(j)}{\nu_{\text{ref}}(j)} - \nu^\star(j) \log \frac{\nu^\star(j)}{\nu_{\text{ref}}(j)} \Big) \\
&= -\beta \sum_{j=1}^n (\nu(j) - \nu^\star(j)) \log \frac{\nu^\star(j)}{\nu_{\text{ref}}(j)} + \beta \sum_{j=1}^n \Big( \nu(j) \log \frac{\nu(j)}{\nu_{\text{ref}}(j)} - \nu^\star(j) \log \frac{\nu^\star(j)}{\nu_{\text{ref}}(j)} \Big) \\
&= \beta \sum_{j=1}^n \nu(j) \log \frac{\nu(j)}{\nu^\star(j)} \;=\; \beta \, \text{KL}(\nu \| \nu^\star).
\end{aligned}
\tag{145}
$$

The $\log Z$ term in eq. (145) vanishes since $\sum_j \nu(j) = \sum_j \nu^\star(j) = 1$. $\qquad\square$

**Lemma F.17** (Stability under payoff perturbations). *Let $A \in \mathbb{R}^{m \times n}$ and $A' \in \mathbb{R}^{m \times n}$ be two payoff matrices with*

$$A - A' = \delta, \qquad \|\delta\|_{\max} := \sup_{i \in [m], \, j \in [n]} |\delta(i,j)| = \sup_{i,j} \big| A(i,j) - A'(i,j) \big|.$$

*The reference policies $\mu_{ref} \in \Delta_m$ and $\nu_{ref} \in \Delta_n$ have full support and fix $\beta > 0$. Let $(\mu, \nu)$ and $(\mu', \nu')$ denote the unique Nash equilibrium policies of the KL-regularized zero-sum matrix games induced by $A$ and $A'$, respectively, with reference policies $(\mu_{ref}, \nu_{ref})$ and regularization strength $\beta$ for both players. If all the entries of $A$ and $A'$ lie in $[A_{\min}, A_{\max}]$. Then the following statements hold*

$$\big| KL(\nu \| \nu_{ref}) - KL(\nu' \| \nu_{ref}) \big| \;\leq\; \frac{2}{\beta} \|\delta\|_{\max} \;+\; \frac{2\,(A_{\max} - A_{\min})}{\beta} \sqrt{\frac{\|\delta\|_{\max}}{\beta}},$$

$$\|\nu - \nu'\|_1 \;\leq\; 2\sqrt{\frac{\|\delta\|_{\max}}{\beta}}.$$
$$\tag{146}$$

*Proof.* Define, for any payoff matrix $A \in \mathbb{R}^{m \times n}$,

$$f_A(\mu, \nu) := \mu^\top A \nu - \beta \operatorname{KL}(\mu \| \mu_{\mathrm{ref}}) + \beta \operatorname{KL}(\nu \| \nu_{\mathrm{ref}}), \qquad \mu \in \Delta_m, \ \nu \in \Delta_n.$$

Let $(\mu, \nu)$ and $(\mu', \nu')$ denote the KL regularized Nash equilibrium policies for $A$ and $A'$, respectively, and recall that $\delta = A - A'$. By Lemma F.16 (minimization form), since $\nu \in \arg\min_{\tilde{\nu} \in \Delta_n} f_A(\mu, \tilde{\nu})$, we have

$$f_A(\mu, \nu') = f_A(\mu, \nu) + \beta \operatorname{KL}(\nu' \| \nu), \tag{147}$$

using Lemma F.16 (maximization form), since $\mu \in \arg\max_{\tilde{\mu} \in \Delta_m} f_A(\tilde{\mu}, \nu)$,

$$f_A(\mu', \nu) = f_A(\mu, \nu) - \beta \operatorname{KL}(\mu' \| \mu). \tag{148}$$

Subtracting (148) from (147) yields

$$f_A(\mu, \nu') - f_A(\mu', \nu) = \beta \Big( \operatorname{KL}(\nu' \| \nu) + \operatorname{KL}(\mu' \| \mu) \Big). \tag{149}$$

Similarly, under $A'$ we have $\nu' \in \arg\min_{\tilde{\nu} \in \Delta_n} f_{A'}(\mu', \tilde{\nu})$ and $\mu' \in \arg\max_{\tilde{\mu} \in \Delta_m} f_{A'}(\tilde{\mu}, \nu')$, hence

$$f_{A'}(\mu', \nu) = f_{A'}(\mu', \nu') + \beta \operatorname{KL}(\nu \| \nu'), \tag{150}$$
$$f_{A'}(\mu, \nu') = f_{A'}(\mu', \nu') - \beta \operatorname{KL}(\mu \| \mu'). \tag{151}$$

Subtracting (151) from (150) yields

$$f_{A'}(\mu', \nu) - f_{A'}(\mu, \nu') = \beta \Big( \operatorname{KL}(\nu \| \nu') + \operatorname{KL}(\mu \| \mu') \Big). \tag{152}$$

Adding (149) and (152) gives

$$\beta \Big( \operatorname{KL}(\nu' \| \nu) + \operatorname{KL}(\nu \| \nu') + \operatorname{KL}(\mu' \| \mu) + \operatorname{KL}(\mu \| \mu') \Big) = \Big( f_A(\mu, \nu') - f_A(\mu', \nu) \Big) + \Big( f_{A'}(\mu', \nu) - f_{A'}(\mu, \nu') \Big). \tag{153}$$

Using $A' = A - \delta$, we have $f_{A'}(\tilde{\mu}, \tilde{\nu}) = f_A(\tilde{\mu}, \tilde{\nu}) - \tilde{\mu}^\top \delta \tilde{\nu}$. for any $(\tilde{\mu}, \tilde{\nu})$. Substituting this into the RHS of eq 153 yields the exact identity

$$\beta \Big( \operatorname{KL}(\nu' \| \nu) + \operatorname{KL}(\nu \| \nu') + \operatorname{KL}(\mu' \| \mu) + \operatorname{KL}(\mu \| \mu') \Big) = -\mu'^\top \delta \nu + \mu^\top \delta \nu'. \tag{154}$$

By eq: (154) and nonnegativity of KL, we have

$$\operatorname{KL}(\nu \| \nu') \leq \frac{1}{\beta} \left( \mu'^\top \delta \nu - \mu^\top \delta \nu' \right) \leq \frac{1}{\beta} \left( \left| \mu'^\top \delta \nu \right| + \left| \mu^\top \delta \nu' \right| \right) \leq \frac{2}{\beta} \| \delta \|_{\max}. \tag{155}$$

Note that the same bound also applies to $\operatorname{KL}(\nu' \| \nu)$. Now by Pinsker's inequality and (155), we have the second part of the lemma

$$\| \nu - \nu' \|_1 \leq \sqrt{2 \operatorname{KL}(\nu \| \nu')} \leq 2 \sqrt{\frac{\| \delta \|_{\max}}{\beta}}. \tag{156}$$

For any $p, q, r$ with the same support

$$\operatorname{KL}(p \| r) - \operatorname{KL}(q \| r) = \operatorname{KL}(p \| q) + \langle p - q, \log(q/r) \rangle.$$

Applying this with $(p, q, r) = (\nu, \nu', \nu_{\mathrm{ref}})$ and taking absolute values gives

$$\left| \operatorname{KL}(\nu \| \nu_{\mathrm{ref}}) - \operatorname{KL}(\nu' \| \nu_{\mathrm{ref}}) \right| \leq \operatorname{KL}(\nu \| \nu') + \| \nu - \nu' \|_1 \cdot \left\| \log \frac{\nu'}{\nu_{\mathrm{ref}}} \right\|_\infty. \tag{157}$$

Since $\nu' \in \arg\min_{\nu' \in \Delta_n} f_{A'}(\mu', \nu')$, it is the KL-regularized best response to $\mu'$ under $A'$, hence

$$\nu'(j) = \frac{\nu_{\mathrm{ref}}(j)\exp\big(-(A'^{\top}\mu')(j)/\beta\big)}{\sum_{k=1}^{n} \nu_{\mathrm{ref}}(k)\exp\big(-(A'^{\top}\mu')(k)/\beta\big)} \qquad \forall j \in [n]. \tag{158}$$

Let $Z := \sum_{k=1}^{n} \nu_{\mathrm{ref}}(k)\exp\big(-(A'^{\top}\mu')(k)/\beta\big)$. Then

$$\log\frac{\nu'(j)}{\nu_{\mathrm{ref}}(j)} = -\frac{(A'^{\top}\mu')(j)}{\beta} - \log Z. \tag{159}$$

Now for every $j \in [n]$,

$$(A'^{\top}\mu')(j) = \sum_{i=1}^{m} \mu'(i)A'(i,j) \in [A_{\min}, A_{\max}], \qquad \text{and} \qquad Z \in \left[e^{-A_{\max}/\beta},\ e^{-A_{\min}/\beta}\right]$$

which implies $\log Z \in \left[-\frac{A_{\max}}{\beta}, -\frac{A_{\min}}{\beta}\right]$ and from (159) we have, for all $j$,

$$\left|\log\frac{\nu'(j)}{\nu_{\mathrm{ref}}(j)}\right| \le \frac{A_{\max} - A_{\min}}{\beta}, \qquad \text{hence} \qquad \left\|\log\frac{\nu'}{\nu_{\mathrm{ref}}}\right\|_{\infty} \le \frac{A_{\max} - A_{\min}}{\beta}. \tag{160}$$

Combining equations (155),(156), (157), (160) we have

$$\big|\mathrm{KL}(\nu\|\nu_{\mathrm{ref}}) - \mathrm{KL}(\nu'\|\nu_{\mathrm{ref}})\big| \le \frac{2}{\beta}\|\delta\|_{\max} + 2\sqrt{\frac{\|\delta\|_{\max}}{\beta}} \cdot \frac{A_{\max} - A_{\min}}{\beta},$$

$$\square$$

# G. Additional discussion

## G.1. Single agent settings

Both OMG and SOMG can be used in the single agent setting for Bandits and RL respectively by setting the action set (and hence even the reference policy) of the min player to a singleton. For Matrix games this results in the same bound as Theorem 2.3.

However, in the RL setting we can obtain a tighter dependence on $H$. Using a smaller bonus term and a projection operator with linear dependence on $H$, we achieve improved regret guarantees. When specialized to the single-agent RL setting, this gives a regret bound of $\min\big\{\widetilde{\mathcal{O}}\big(d^{3/2}H^2\sqrt{T}\big),\ \mathcal{O}\big(\beta^{-1}d^3H^5\log^2(T/\delta)\big)\big\}$.

The value function in game theoretic setting is given by

$$V_h^{\mu,\nu}(s) := \mathbb{E}\left[\sum_{k=h}^{H} r_k(s_k, i, j) - \beta\mathrm{KL}(\mu_k(\cdot|s_k)\|\mu_{\mathrm{ref},k}(\cdot|s_k)) + \beta\mathrm{KL}(\nu_k(\cdot|s_k)\|\nu_{\mathrm{ref},k}(\cdot|s_k))\ \middle|\ s_h = s\right],$$

This design of bonus terms and projection operators is possible due to the fact that when the min player action set is restricted to singleton the positive KL terms disappear and the value function (and hence the $Q$ functions) will now be bounded between $(-\infty, H]$ instead of $(-\infty, \infty)$. Specifically for a policy $\pi$ the value function in KL regularized RL is given by

$$V_h^{\pi}(s) := \mathbb{E}\left[\sum_{k=h}^{H} r_k(s_k, i, j) - \beta\mathrm{KL}(\pi_k(\cdot|s_k)\|\pi_{\mathrm{ref},k}(\cdot|s_k))\ \middle|\ s_h = s\right],$$

Thus the projection for best response $Q$ function can now use $\mathcal{O}(H)$ ceiling in equation (20b). We donot need a $Q^-$ ((15c),(16c)) in SOMG since the min player makes no decisions (action set is singleton) as shown Algorithm. As a result of this the we get a $H$ dependence in bonus term and hence a $\min\big\{\widetilde{\mathcal{O}}\big(d^{3/2}H^2\sqrt{T}\big),\ \mathcal{O}\big(\beta^{-1}d^3H^5\log^2(T/\delta)\big)\big\}$ regret. This

matches the best known regret bound obtained by (Zhao et al., 2025b)[5] in single agent KL regularized RL. Furthermore as a result of simpler function classes SOMG can also be directly be used without a discretization argument for single agent RL for the unregularized $\beta = 0$ case.

### G.2. Extension to general function approximation

SOMG can be extended beyond the linear MDPs to RKHS/General function approximation for the $Q$ dunction class with local (state-action wise) optimism using standard arguments from the literature. For example to extend SOMG to general function approximation we additionally need a standard realizability assumption (Zhao et al., 2025b; Ye et al., 2023) on the value functions class induced by SOMG in equation 19 which we get for free in Linear MDP (From Assumption 3.1 and lemma F.12) and a bounded log covering number assumption.

Beyond this the only parts of the SOMG proof that are specific to linear MDP are lemmas F.4, F.5 which define bonuses $b_{h,t}^{\mathrm{mse}}$, $b_{h,t}$ respectively and the bounding of sum of squares of bonuses in equations (94) and (108) using elliptical potential lemma D.6. Replacing these components with bonuses used general function approximation, as done in (Zhao et al., 2025b;c), extends our results to general function approximation settings.

Specifically the width of the uncertainty set at step $h$, time $t$ for $(s,i,j)$ in linear function approximation is specified using the covariance matrix $\mathcal{U}_{h,t}^{\mathrm{lin}}(s,i,j;\mathcal{D}_{t-1}) := \|\phi(s,i,j)\|_{\Sigma_{h,t}^{-1}}$ and the bonus is of the form $\eta \cdot \min\left\{1, \mathcal{U}_{h,t}^{\mathrm{lin}}(s,i,j;\mathcal{D}_{t-1})\right\}$ which is equal to $\eta \cdot \mathcal{U}_{h,t}^{\mathrm{lin}}(s,i,j;\mathcal{D}_{t-1})$ when regularization $\lambda > 1$ (both $b_{h,t}^{\mathrm{mse}}$ and $b_{h,t}$ take this form) with $\eta$ being a constant that depends on problem parameters. Under general function approximation (with a function class $\mathcal{F}$) the width of the uncertainty set is given by (Agarwal et al., 2023; Ye et al., 2023; Zhao et al., 2025b)

$$\mathcal{U}_{h,t}^{\mathrm{gen}}(s,i,j;\mathcal{D}_{t-1}) := \sup_{f,f'\in\mathcal{F}} \frac{|f(s,i,j) - f'(s,i,j)|}{\sqrt{\lambda + \sum\limits_{s_h,i_h,j_h\in\mathcal{D}_{t-1}} (f(s_h,i_h,j_h) - f'(s_h,i_h,j_h))^2}}$$

where $\lambda$ is the regularization parameter. To obtain bounds for general function approximation we use $\mathcal{U}_{h,t}^{\mathrm{gen}}$ instead of $\mathcal{U}_{h,t}^{\mathrm{lin}}$ to create confidence intervals in lemmas F.4 and F.5 and use eluder dimension (Agarwal et al., 2023; Zhao et al., 2025b)

$$d(\mathcal{F},T) := \sup_{s_{1:T}, i_{1:T}, j_{1:T}} \sum_{t=1}^{T} \min\big(1, [\mathcal{U}_{h,t}^{\mathrm{gen}}(s_t,i_t,j_t;\mathcal{D}_{t-1})]^2\big).$$

instead of elliptical potential lemma to bound the sum of squares of bonus terms. Similar arguments extend OMG to General function approximation.

### G.3. Discussion about lower bounds

There are no known lower bounds for sample complexity/Regret in KL regularized games. However, for the bandits setting, a sample complexity lower bound was presented in (Zhao et al., 2025a) (Theorem 3.6) which we restate here

**Theorem G.1** ((Zhao et al., 2025a)). *For any $\epsilon \in (0, 1/256), \beta < \frac{1}{4}$, and any algorithm $\mathcal{A}$, there exists a KL-regularized contextual bandit problem with reward function class $\mathcal{R}$ with covering number $O(N_\mathcal{R}(\epsilon))$ and such that $\mathcal{A}$ requires at least $\Omega\left(\min\left(\frac{\beta^{-1}\log N_\mathcal{R}(\epsilon)}{\epsilon}, \frac{\log N_\mathcal{R}(\epsilon)}{\epsilon^2}\right)\right)$ rounds to achieve a suboptimality $\epsilon$.*

For linear function approximation $\log N_\mathcal{R}(\epsilon)$ scales proportional to $d$ (lemma D.1). Since bandits is a single agent special case of both Matrix games (by setting the min player action set to singleton) and Markov games ($H = 1$ gives matrix games), the lower bound also applies to our setting and we note that our upper bounds obtains the optimal structure $\min\{\mathcal{O}(\beta^{-1}/\varepsilon), \mathcal{O}(1/\varepsilon^2)\}$ and dependency on $\beta$.

The best known regularization dependent regret upper bounds for KL regularized RL is $\tilde{\mathcal{O}}(\beta^{-1}H^5 d^3 \log^2(T))$ presented in (Zhao et al., 2025b) which gives a sample complexity of $\tilde{\mathcal{O}}\left(\frac{\beta^{-1}H^5 d^3}{\varepsilon}\right)$[6] (Zhao et al., 2025b; Tiapkin et al., 2024) match the

---

[5]The bound is adopted for linear MDP where the log covering number $\log(\mathcal{N})$ grows as $d^2 \log(T)$ (lemma F.14), we use $\sum_{h=1}^{H} r_h \in [0, H]$ while (Zhao et al., 2025b) use $\sum_{h=1}^{H} r_h \in [0, 1]$, translating this to our setting gives an additional $H^2$ factor due to dependency on the square of the bonus term. $d(\mathcal{F}, \lambda, T) = \sum_{h=1}^{H} d(\mathcal{F}_h, \lambda, T)$ scales at $dH$

[6]$\left\|\hat{\Lambda}_h^T\right\|_2$ in ((Tiapkin et al., 2024) Thm. 6 Page 53) should be $d(2 + (T - 1))$ (minor typo) and hence the dependency will be $d^3$

rates obtained by SOMG when specialized to the single agent setting. However we remark the dependence on $H$ and $d$ here is not tight. These can be potentially improved in future works using Bernstein based bonuses/reference advantage decomposition which is commonly used to obtain sharp rates in bonus based methods for both offline (Shi et al., 2022) and online (Chen et al., 2022) RL and games.

instead of $d^2$.

