# Achieving Logarithmic Regret in KL-Regularized Zero-Sum Markov Games

## Abstract

Reverse Kullback–Leibler (KL) divergence-based regularization with respect to a fixed reference policy is widely used in modern reinforcement learning to preserve the desired traits of the reference policy and sometimes to promote exploration (using uniform reference policy, known as entropy regularization). Beyond serving as a mere anchor, the reference policy can also be interpreted as encoding prior knowledge about good actions in the environment. In the context of alignment, recent game-theoretic approaches have leveraged KL regularization with pretrained language models as reference policies, achieving notable empirical success in self-play methods. Despite these advances, the theoretical benefits of KL regularization in game-theoretic settings remain poorly understood. In this work, we develop and analyze algorithms that provably achieve improved sample efficiency under KL regularization. We study both two-player zero-sum Matrix games and Markov games: for Matrix games, we propose OMG, an algorithm based on best response sampling with optimistic bonuses, and extend this idea to Markov games through the algorithm SOMG, which also uses best response sampling and a novel concept of superoptimistic bonuses. Both algorithms achieve a logarithmic regret in $T$ that scales inversely with the KL regularization strength $\beta$ in addition to the traditional $\widetilde{\mathcal{O}}(\sqrt{T})$ regret without the $\beta^{-1}$ dependence.

## 1. Introduction

Multi-agent reinforcement learning (MARL) has emerged as a key framework for modeling strategic interactions among multiple decision makers, providing a powerful tool for analyzing both cooperative and competitive dynamics in

[1]Anonymous Institution, Anonymous City, Anonymous Region, Anonymous Country. Correspondence to: Anonymous Author <anon.email@domain.com>.

Preliminary work. Under review by the International Conference on Machine Learning (ICML). Do not distribute.

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

 \text{KL}(\mu_k(\cdot|s_k) \| \mu_{\text{ref},k}(\cdot|s_k)) + \beta \text{KL}(\nu_k(\cdot|s_k) \| \nu_{\text{ref},k}(\cdot|s_k)) \,\middle|\, s_h = s \right],$$

This design of bonus terms and projection operators is possible due to the fact that when the min player action set is restricted to singleton the positive KL terms disappear and the value function (and hence the $Q$ functions) will now be bounded between $(-\infty, H]$ instead of $(-\infty, \infty)$. Specifically for a policy $\pi$ the value function in KL regularized RL is given by

$$V_h^\pi(s) := \mathbb{E}\left[ \sum_{k=h}^H r_k(s_k, i, j) - \beta \text{KL}(\pi_k(\cdot|s_k) \| \pi_{\text{ref},k}(\cdot|s_k)) \,\middle|\, s_h = s \right],$$

Thus the projection for best response $Q$ function can now use $\mathcal{O}(H)$ ceiling in equation (20b). We donot need a $Q^-$ ((15c),(16c)) in SOMG since the min player makes no decisions (action set is singleton) as shown Algorithm. As a result of this the we get a $H$ dependence in bonus term and hence a $\min\left\{ \widetilde{\mathcal{O}}\left( d^{3/2} H^2 \sqrt{T} \right), \mathcal{O}\left( \beta^{-1} d^3 H^5 \log^2(T/\delta) \right) \right\}$ regret. This

matches the best known regret bound obtained by (Zhao et al., 2025b)[5] in single agent KL regularized RL. Furthermore as a result of simpler function classes SOMG can also be directly be used without a discretization argument for single agent RL for the unregularized $\beta = 0$ case.

### F.2. Discussion about lower bounds

There are no known lower bounds for sample complexity/Regret in KL regularized games. However, for the bandits setting, a sample complexity lower bound was presented in (Zhao et al., 2025a) (Theorem 3.6) which we restate here

**Theorem F.1** ((Zhao et al., 2025a)). *For any $\epsilon \in (0, 1/256), \beta < \frac{1}{4}$, and any algorithm $\mathcal{A}$, there exists a KL-regularized contextual bandit problem with reward function class $\mathcal{R}$ with covering number $O(N_{\mathcal{R}}(\epsilon))$ and such that $\mathcal{A}$ requires at least $\Omega\left(\min\left(\frac{\beta^{-1} \log N_{\mathcal{R}}(\epsilon)}{\epsilon}, \frac{\log N_{\mathcal{R}}(\epsilon)}{\epsilon^2}\right)\right)$ rounds to achieve a suboptimality $\epsilon$.*

For linear function approximation $\log N_{\mathcal{R}}(\epsilon)$ scales proportional to $d$ (lemma C.1). Since bandits is a single agent special case of both Matrix games (by setting the min player action set to singleton) and Markov games ($H = 1$ gives matrix games), the lower bound also applies to our setting and we note that our upper bounds obtains the optimal structure $\min\{\mathcal{O}(\beta^{-1}/\varepsilon), \mathcal{O}(1/\varepsilon^2)\}$ and dependency on $\beta$.

The best known regularization dependent regret upper bounds for KL regularized RL is $\tilde{\mathcal{O}}(\beta^{-1} H^5 d^3 \log^2(T))$ presented in (Zhao et al., 2025b) which gives a sample complexity of $\tilde{\mathcal{O}}\left(\frac{\beta^{-1} H^5 d^3}{\varepsilon}\right)$[6] (Zhao et al., 2025b; Tiapkin et al., 2024) match the rates obtained by SOMG when specialized to the single agent setting. However we remark the dependence on $H$ and $d$ here is not tight. These can be potentially improved in future works using Bernstein based bonuses/reference advantage decomposition which is commonly used to obtain sharp rates in bonus based methods for both offline (Shi et al., 2022) and online (Chen et al., 2022) RL and games.

---

[5]The bound is adopted for linear MDP where the log covering number $\log(\mathcal{N})$ grows as $d^2 \log(T)$ (lemma E.14), we use $\sum_{h=1}^{H} r_h \in [0, H]$ while (Zhao et al., 2025b) use $\sum_{h=1}^{H} r_h \in [0, 1]$, translating this to our setting gives an additional $H^2$ factor due to dependency on the square of the bonus term. $d(\mathcal{F}, \lambda, T) = \sum_{h=1}^{H} d(\mathcal{F}_h, \lambda, T)$ scales at $dH$

[6]$\left\|\hat{\Lambda}_h^T\right\|_2$ in ((Tiapkin et al., 2024) Thm. 6 Page 53) should be $d(2 + (T - 1))$ (minor typo) and hence the dependency will be $d^3$ instead of $d^2$.