# OpenReview forum: "Achieving Logarithmic Regret in KL-Regularized Zero-Sum Markov Games"
_ICML.cc/2026/Conference — ICML 2026 regular_

### Official Review · Reviewer_ssCd · 2026-03-01

**Soundness:** 4
**Presentation:** 4
**Significance:** 4
**Originality:** 4
**Overall Recommendation:** 6
**Confidence:** 5

**Summary:**

This paper investigates the theoretical impact of Kullback-Leibler (KL) divergence regularization in the context of two-player zero-sum Markov Games. While KL regularization is a staple in empirical reinforcement learning (particularly in LLM alignment and self-play), its ability to accelerate convergence in competitive games was previously unquantified. The work examines a notable aspect of these dynamics by demonstrating that anchoring policies to a fixed reference $\rho$ provides a strong "pre-conditioning" effect. Overall, the authors address a major challenge by introducing the SOMG (Self-Play with Optimistic Model-based Gradient) algorithm and proving it achieves logarithmic regret $O(\frac{1}{\beta} \log T)$, a significant improvement over the standard sublinear rates.

**Compliance With Llm Reviewing Policy:**

Affirmed.

**Final Justification:**

I will keep my positive score. The paper is high quality, and the rebuttal addressed all the concerns.

**Key Questions For Authors:**

please refer to the weaknesses mentioned above

**Limitations:**

yes

**Strengths And Weaknesses:**

Strengths

1.The move from $\sqrt{T}$ to $\log T$ regret is a major theoretical milestone. It suggests that with proper regularization, competitive games can be learned with exponentially fewer samples than unregularized ones.

2.The model directly maps to current practices in LLM self-play and RLHF, providing a rigorous mathematical justification for using pretrained models as reference anchors.

Weaknesses:
1. The SOMG algorithm is model-based, which typically involves higher memory requirements for maintaining transition estimates compared to model-free approaches.

2. The current results are restricted to zero-sum interactions. While this is the standard for competitive games, many real-world multi-agent scenarios involve general-sum payoffs where these guarantees may not directly apply.

---

> ### Author Rebuttal · Authors · 2026-03-30
>
> We thank the reviewer for positive eval and support for our work. Here are the answers to your questions.
>
> > **Memory reqs on proposed algs**
>
> We acknowledge that the work was mostly theoretical and the memory requirement here is a concern we didn't discuss. One can often tradeoff memory with compute (only store past policies vs recompute them. We donot maintain transition estimates but we need memory for past $Q$ estimates). Another trick used commonly in extremely large scale settings is using a Neural Network based predictor trained on observed data to avoid storing all estimates and recomputing them as and when required.
>
> >**About extension to general sum settings**
>
> We currently donot have an answer to what algorithm will work well for Multi agent games with CCE, an additional challenge in CCE is the non smoothness of the CCE value to a change in Q function matrix [1] (smoothness in Q function and policy is used in both single agent and two player game analysis). The analysis fall outside the scope of current work.
>
> ### References
>
> - [1] Xie, Qiaomin, et al. *"Learning zero-sum simultaneous-move Markov games using function approximation and correlated equilibrium."* Conference on Learning Theory. PMLR, 2020.

---

> > ### Author Rebuttal · Reviewer_ssCd · 2026-04-01
> >
> > I am happy with the answers and keeping my scores.

---

> > > ### Author Response · Authors · 2026-04-03
> > >
> > > Thank you for your support! and acknowledging the rebuttal. We are glad that the rebuttal answered your questions.

---

### Official Review · Reviewer_6LBn · 2026-03-12

**Soundness:** 3
**Presentation:** 3
**Significance:** 3
**Originality:** 3
**Overall Recommendation:** 4
**Confidence:** 4

**Summary:**

The paper studies KL-regularized two-player zero-sum games under bandit feedback, motivated by settings where a fixed reference policy encodes prior knowledge, pretrained behavior, or a desirable policy anchor. The authors aim to show that, in game-theoretic settings, KL regularization can be exploited to obtain improved regret guarantees and sample efficiency. The significance lies in extending logarithmic-regret phenomena from single-agent settings to two-player zero-sum games.

**Compliance With Llm Reviewing Policy:**

Affirmed.

**Final Justification:**

The rebuttal addressed my concerns, and I will maintain my original score of weak accept.

**Key Questions For Authors:**

1. The proposed algorithms appear to require computing the Nash equilibrium of the intermediate KL-regularized game at each round or state, in addition to computing optimistic best responses. Since a closed-form Nash equilibrium policy does not seem to be available in general, the authors should clarify what computational oracle, optimization routine, or structural assumption is being used.

2. The algorithmic descriptions involve both the regularized equilibrium policy pair and the optimistic best-response pair, but the distinction between these two objects is not sufficiently clear. Could the authors explain more clearly why both are required, what respective roles they play in the algorithm and analysis, and whether they differ significantly in computation?

3. The paper presents the minimum of a regularization-dependent bound and a regularization-independent bound. Could the authors discuss more explicitly the regime in which the regularization-dependent logarithmic term is expected to be dominant, and what conceptual or practical insight this provides about the benefit of KL regularization?

4. Even if the required equilibria and best responses are well defined, how tractable are these computations in larger matrix games or large-scale Markov games? It would be helpful to understand whether the contribution should be viewed mainly as an information-theoretic result or whether the authors also expect the proposed procedures to be computationally practical.

**Limitations:**

1. The algorithms appear to require repeated computation of regularized Nash equilibria and optimistic best responses, which may be computationally burdensome in large-scale settings.

2. The paper is primarily theory-driven, so it provides limited empirical validation.

**Strengths And Weaknesses:**

Strengths:

1. The paper addresses an important and timely theoretical question.

2. The theoretical scope is comprehensive, covering several different settings.

Weaknesses:

1. Some definitions and notations appear incomplete or inconsistent. For instance, terms such as LMSE seem undefined, and there are notational issues or typos (such as the Input part in both pseudocodes, the definition of Q for equation (14), and so on) that make some parts of the paper harder to follow.

2. The computational requirements of the algorithms are not transparent. Since the methods appear to require both the Nash equilibrium of the current regularized game and optimistic best responses, the paper should clarify more explicitly what computation is assumed when closed-form NE policies are unavailable.

3. The practical implications should be discussed more concretely, especially regarding the potential implications for practical large-scale settings.

---

> ### Author Rebuttal · Authors · 2026-03-30
>
> We thank the reviewer for a thorough eval of our work. We try to address your concerns below. If our responses resolve your questions, we'd very much appreciate your consideration in raising the score.
>
> >**Missing defn and typos (W1)**
>
> We will clean these in the camera ready version (we already identified these and fixed it on our end locally post submission in both maintext and appendix). We hope you were able to infer these things from context (ofc we will fix them for readability). For the ones you mentioned,
>
> - LMSE is Least Mean Square Error (refers to estimates pre optimism injection).
> - Input is just the system parameters fed into the algorithm. The double mention of regularization was a typo.
> - $Q$: yes, the policy pair should be $\mu, \nu^{\dagger}$ in Eq. 14.
>
> >**Compute requirements of the algos (W2, Q1, L1 & Q4)**
>
> This is a good question, often goes under the rugs in sample complexity analyses which ignore computational aspects and usually assume oracle access (in this case, a Nash eq (NE) computation oracle) to isolate and emphasize the sample complexity component.
>
> Nonetheless, it is important to note that such oracles can be instantiated in practice: in KL-regularized games, the NE can be computed to arbitrary precision $\epsilon_{\text{comp}}$ within $\log(1/\epsilon_{\text{comp}})$ iterations, owing to the geometric convergence rates achieved by methods such as $[1],[2],[3]$. (Note that that algos in $[1],[3]$ are described using entropy reg but all the same results and algorithms hold for KL reg by changing the ref policy from uniform to appropriate one). We use the matrix games versions of these optimizers for our algos. SOMG solves stage wise matrix games. We used these methods in our exps.
>
>  In terms of theory, this approximation induced error will manifest as an additive term, linear in $\epsilon_{\text{comp}}$ ($\text{Regret}(T)+\text{poly}(d,H)T\epsilon_{\text{comp}}$) in the regret. However, they are ignored in sample complexity analyses as they can be made arbitrarily small compared to the regret (choosing $\epsilon_{\text{comp}} = 1/T^2$). (Since we have compute efficient methods to do so).
>
> **Actually the oracle we use in here is more efficient than all other discussed unregularized works** as briefly discussed in the "Computational benefits of regularization" paragraph before section 3.3.
>
> > **Extension to large scale settings (W3)**
>
> For large scale settings one can use neural network based estimators and switch to General Function Approximation (GFA), Both OMG and SOMG gracefully extend to GFA, we will add a discussion about extension to general function classes in the appendix. We focused on the linear setting primarily to simplify notation and maintain clarity, allowing readers to pay attention to the core method rather than the additional technical machinery associated with GFA (e.g., eluder dimension, confidence set construction, covering number etc). Our choice was therefore expository rather than fundamental, and the algorithms extend naturally beyond the linear case.
>
> > **Distinction between the role of equilibrium policies and best response policy pairs**
>
> The regularized equilibrium policies serve as good anchors (If we don't use good anchors the KL terms can blow up and go unbounded). The best response policies are where optimism/superoptimism is injected w.r.t. the anchors, ie for calculating the max player best response, we add the +ve bonus terms and make the game more favorable/optimistic for max player and compute the optimistic best response under this optimistic model to the regularized policy of the min player, similarly this favorable game for min player is obtained by subtracting the bonus term. More intuition is discussed in the proof overview section in the appendix.
> Computation of NE is done as mentioned in the replies above, best response has a closed form given opponent policy and doesn't require iterative methods.
>
> >**Reg dependent vs independent bound regimes**
>
> When the regularization strength $\beta$ is small/ $T$ is small, regularization independent bound is active, as $\beta$ and $T$ increase, regularization independent bound becomes active (min of the two). The main insight is that, under non zero KL regularization the regret can grow logarithmically in $T$ asymptotically as opposed to $\mathcal{O}(\sqrt{T})$ growth in prior works.
>
> > **Experimental results (L2)**
>
> We added small scale experiments (in the plots folder) as suggested here
> https://anonymous.4open.science/r/SOMG_ICML-FA03/README.md for randomly generated MDPs
> in order to show that SOMG stabilizes learning.
>
> 1. [1] Shicong Cen, et al. Faster last-iterate convergence of policy optimization in zero-sum Markov games.
> 2. [2] Sokota, Samuel, et al. A Unified Approach to Reinforcement Learning, Quantal Response Equilibria, and Two-Player Zero-Sum Games.
> 3. [3] Cen, S., et al Fast policy extragradient methods for competitive games with entropy regularization.

---

> > ### Author Rebuttal · Reviewer_6LBn · 2026-04-03
> >
> > I thank the authors for addressing my concern. I will keep my positive score.

---

> > > ### Author Response · Authors · 2026-04-03
> > >
> > > Thank you for your support and for acknowledging our rebuttal!. We are pleased that it addressed your questions.

---

### Official Review · Reviewer_Jqrz · 2026-03-12

**Soundness:** 2
**Presentation:** 3
**Significance:** 2
**Originality:** 3
**Overall Recommendation:** 3
**Confidence:** 3

**Summary:**

This study develops learning algorithms that achieve superior sample efficiency and regret bound in KL-regularized two-player zero-sum matrix and Markov games under bandit feedback. The proposed methods sample actions from optimistic best response strategies. The authors prove logarithmic regret bounds as well as fast average-iterate convergence rates.

**Compliance With Llm Reviewing Policy:**

Affirmed.

**Key Questions For Authors:**

My main concerns and questions are stated in the Weaknesses section. Additionally, I have the following minor questions and comments:

- (Section 2.3) Could the authors clarify how the sample complexity of $\tilde{\mathcal{O}}(1/\varepsilon)$ follows from Theorem 2.3? Is this obtained directly from the fact that the average-iterate convergence rate can be bounded by $\mathrm{Reg}(T)/T$?
- (Equation (13)) In the expression $Q^{\dagger}=Q^{\mu^{\dagger}, \nu}$, the definition of $Q^{\mu^{\dagger}, \nu}$ seems to be missing.

**Limitations:**

I see no negative societal impacts that need to be addressed.

**Strengths And Weaknesses:**

### Strengths

To the best of my knowledge, the regret bound of $\mathcal{O}(\beta^{-1}\log^2 (T))$ improves upon previously known results under bandit feedback. It also appears that this improvement is enabled by the strong convexity induced by KL divergence. I also find it interesting that the regret bound depends explicitly on the regularization parameter, meaning that stronger regularization can lead to better regret guarantees.

### Weaknesses

A main concern is the practical relevance of the proposed approach in large-scale domains. Prior works such as Munos et al. (2024) and Wang et al. (2025) study decoupled online learning algorithms, including mirror descent-type methods, which do not require estimation of the full game matrix. In contrast, the proposed method appears to rely on game-matrix estimation, which may become substantially less practical in applications with extremely large action spaces, such as LLM fine-tuning or alignment. Since the abstract and introduction explicitly mention LLM alignment as a motivating application of KL regularization, it would be helpful for the authors to clarify in what sense the proposed approach is expected to be preferable to such decoupled online learning methods in those domains.

In addition, although I understand that the paper is primarily theoretical, some experimental validation would still strengthen the work. Given that KL regularization has been incorporated into many learning algorithms and often shows strong empirical performance, I believe it would be valuable to include at least a small-scale experiment in a Markov game setting.

---

> ### Author Rebuttal · Authors · 2026-03-30
>
> Thank you for taking the time to carefully review our paper and for providing valuable feedback. We would like to address the questions you raise. If our responses resolve your questions, we'd appreciate your consideration in raising the score. Please don't hesitate to request any further clarification.
>
> > **About comparison with decoupled online learning methods (W1)**
>
> Thanks for the comment, The works (Munos et al. (2024) and Wang et al. (2025) cited in the paper) mentioned above study a decoupled paradigm in which one first learns a reward model from data and then performs optimization on the learned model. This separation implicitly assumes access to sufficiently informative data for reward-model learning which is essentially them operating in the complete information setting (enough data available to learn good RM) as they optimize for the learned RM. As a result, their works target computational complexity and not statistical efficiency as the central theme.
>
> While such an assumption may be  reasonable in general language domains at training time where such datasets are available, it may be much less realistic in new domains, where the reward model must be learned simultaneously with the policy. A similar issue arises in test-time training/discovery settings/fine tuning and alignment for personalization, where reward feedback is only gathered during deployment. In such settings right exploration strategy and sampling efficiency are critical and practical versions of our method will be useful.
>
> That being said our methods donot structurally depend on language modeling and extend beyond language model RL, practical versions of our method can be also adopted to general deep RL, self play settings where KL regularization with reference policy initialized using imitation learning is used. Many of these setting donot have enough samples apriori to learn a good reward model and are expected to explore to collect data and learn a good policy (to exploit) simultaneously.
>
> > **Experimental results (W2)**
>
> We added small scale experiments (in the plots folder) as suggested here
> https://anonymous.4open.science/r/SOMG_ICML-FA03/README.md for randomly generated MDPs
> in order to show that SOMG stabilizes learning. We will add this in the camera ready appendix.
>
> > **Deriving sample complexity from the Regret Bound (Q1)**
>
> Thanks for the comment, if we interpret the comment correctly, your intuition is right!. The sample complexities come from standard online regret-to-batch
> conversion for the time-averaged policy used in online/active learning literature. Given regret is bounded by $\text{Reg}(T)$ one can construct a time averaged policy whose suboptimality will be bounded by $\text{Reg}(T)/T$. More details can be found in [1]
>
> >**Missing definitions (Q2)**
>
> We will add this definition to the camera ready version, it was removed last minute to save space. It is presented in the appendix. Its just eqn 9,10 with policies $(\mu^{\dagger},\nu)$
>
> ### References
>
> 1. [1] Zhang, T. *Mathematical analysis of machine learning algorithms.* Cambridge University Press, 2023.

---

> > ### Author Rebuttal · Reviewer_Jqrz · 2026-04-03
> >
> > Thank you for the clarification.
> > My concern is that the proposed method is not decoupled in an essential way. I understand that in Munos et al. (2024), one first needs to learn a reward model separately. However, the strategy update itself is based on a decoupled learning algorithm, namely Mirror Descent, which seems more suitable for practical large-scale applications. In contrast, my understanding is that your proposed method still relies on a more tightly coupled procedure.

---

> > > ### Author Response · Authors · 2026-04-03
> > >
> > > Thank you for the acknowledgement and for engaging with our rebuttal!. We would like to provide one additional clarification concerning the comparison with Munos et al. (2024).
> > >
> > > The learning NE problem has two elements
> > > 1) Computing NE given a reward model (payoff matrix in matrix game)/ reward model + transition probability matrix/Q function in Markov game) - computational complexity problem
> > > 2) Estimating the reward model - sample complexity problem
> > >
> > > Munos et al. addresses the first subproblem while our algorithms solve the second sub-problem. Both are complementary and can be combined potentially.
> > >
> > > Step 6 in OMG and Step 7 in SOMG, where we compute the Nash equilibrium is where we can use the algorithm from Munos et al.
> > >
> > > Munos et al. proposes an algorithm for the inner game-solving problem once the payoff matrix is available. In this sense, the algorithm of Munos et al. can naturally be viewed as a computationally efficient subroutine within our algorithm.
> > >
> > > Infact, we use a very related algorithm in our implementation for those steps. ( More specifically we use mirror descent type algorithm [1], similar ones can be also found in [2],[3]) https://anonymous.4open.science/r/SOMG_ICML-FA03/README.md
> > >
> > > Our contribution is about the second part: we study how to estimate the correct reward model online, namely the sample-complexity problem. Given such a computationally efficient subroutine, our main contribution is a more sample-efficient approach than methods that first learn a reward model and then optimize it through iterative estimation and optimization.
> > >
> > > This is in spirit with much of the sample complexity works which say that data collection can be adaptive in the presence of efficient solvers.
> > >
> > > Our initial rebuttal comments were mostly comparing our algorithm to their collect data and solve approach (not their central result), and put less emphasis on the fact that they are mainly solving different subproblem from ours and the fact that they can be combined.
> > >
> > > Collect all data first by random exploration and solve is just one (the simplest) strategy for step 2 which is actually inefficient in large state action spaces. Munos et al use it as their focus is to study computational efficiency and not downstream sample efficiency.
> > >
> > > In large scale settings, even when you have access to prior data, its better to use adaptive methods when you're required to  collect more data [4]
> > >
> > > - [1] Shicong Cen, et al. Faster last-iterate convergence of policy optimization in zero-sum Markov games.
> > > - [2] Sokota, Samuel, et al. A Unified Approach to Reinforcement Learning, Quantal Response Equilibria, and Two-Player Zero-Sum Games.
> > > - [3] Cen, S., et al Fast policy extragradient methods for competitive games with entropy regularization.
> > > - [4] Bose, Avinandan, et al. "Hybrid preference optimization for alignment: Provably faster convergence rates by combining offline preferences with online exploration." arXiv preprint arXiv:2412.10616 (2024).
> > >
> > > We hope this clarification helps address your concern, and if it does, we would be grateful if you would consider raising your score accordingly.

---

### Official Review · Reviewer_J2Zh · 2026-03-17

**Soundness:** 4
**Presentation:** 4
**Significance:** 4
**Originality:** 4
**Overall Recommendation:** 5
**Confidence:** 4

**Summary:**

The paper discuss a central concept in modern reinforcement learning: how KL divergence regularization with respect to a reference policy can be leveraged to achieve provably superior sample efficiency in competitive multi-agent settings. An important area analyzed by the paper is the gap between single-agent and game-theoretic settings with respect to logarithmic regret guarantees under KL regularization.
Specifically, the paper proposes two algorithms: OMG for two-player zero-sum matrix games, and SOMG for Markov games. Both achieve regret bounds that scale logarithmically in the number of episodes T when the KL regularization strength β > 0, improving upon the standard O(√T) rates achieved by prior work.

**Compliance With Llm Reviewing Policy:**

Affirmed.

**Final Justification:**

The response has resolved my concerns and I remain positive about the paper.

**Key Questions For Authors:**

- H dependence: Is the H⁷ dependence in Theorem 3.3 tight? Can it be improved to H⁵ (matching the single-agent case) with Bernstein-based bonuses, and if so, is this straightforward or does it require new ideas?
- The log(1/min{1,β}) term suggests the covering number of the superoptimistic value function class blows up as β → 0. Does this imply the analysis does not gracefully recover the unregularized setting from the regularization-dependent bound, or is this handled entirely by switching to the regularization-independent bound?
-  For the CCE objective in general-sum games, best responses no longer have closed-form expressions. Do the authors see any path toward extending their approach to this setting, even with additional assumptions?

**Limitations:**

Yes

**Strengths And Weaknesses:**

Strengths

-  Clear and significant theoretical contribution. The gap between single-agent and game-theoretic settings under KL regularization was a well-identified open problem. Prior work (Zhao et al., 2025b; Tiapkin et al., 2024) established logarithmic regret for single-agent RL, but no analogous result existed for games. This paper closes that gap.
- Technically non-trivial. The paper identifies and carefully addresses the key challenges unique to the game-theoretic setting: The absence of closed-form Nash equilibrium policies (unlike the Gibbs-form optimal policies in single-agent settings), addressed via best-response sampling. Unbounded value functions due to positive KL terms in both players' objectives, addressed via the superoptimistic projection operator with a ceiling of O((H−h+1)²). Non-trivial covering number bounds for the superoptimistic value function class, exploiting smoothness properties of KL-regularized Nash equilibria (Lemma E.15, E.17).

- Algorithm design is well-motivated.
The concept of superoptimistic bonuses ($b_h^{sup} = b_h + 2b_h^{mse}$) is a well motivated idea. The distinction between standard optimism (sufficient for √T bounds) and superoptimism (necessary for logarithmic bounds in games) is explained clearly in Section B.2.

- Both OMG and SOMG provide simultaneous regularization-dependent (logarithmic) and regularization-independent (√T) bounds. This ensures the algorithms never perform worse than unregularized baselines, which is important for practitioners.

- The authors show that SOMG recovers the best known regret bound of Zhao et al. (2025b) when specialized to single-agent RL, lending additional credibility to the tightness of their analysis.
- Table 1 is clear and well-organized. The comparison with prior work is presented comprehensively.

Weaknesses
- Dependence on H appears suboptimal. The regularization-dependent Markov game bound scales as O(β⁻¹d³H⁷ log²(T)), while the single-agent analogue from Zhao et al. (2025b) scales as O(β⁻¹d³H⁵ log²(T)). The authors acknowledge this H² gap and attribute it to the presence of positive KL terms in the game-theoretic value function requiring a quadratic projection ceiling. However, the paper does not discuss whether this H dependence is tight or an artifact of the analysis technique, beyond a brief remark in Section F.2 about Bernstein-based bonuses.

- No experiments. This is a theoretical paper and experiments are not strictly required, but even simple synthetic experiments illustrating the logarithmic vs. √T regret curves for varying β would strengthen the paper and help readers build intuition.


- Extension to general-sum games or CCE is left entirely open. The paper's approach fundamentally relies on the zero-sum structure so that best responses admit closed-form expressions. While this limitation is acknowledged in the conclusion, the gap relative to practical MARL applications (which often involve general-sum settings) is significant and deserves more discussion.

---

> ### Author Rebuttal · Authors · 2026-03-30
>
> Thank you for taking the time to carefully review our paper and overall positive evaluation of our work. We would like to address the questions you raise. If
> our responses resolve your questions, we’d appreciate your consideration in raising the score. Please don’t
> hesitate to request any further clarification.
>
> > **Dependence on $H$ (W1 and Q1)**
>
> Yes, Bernstein-type bonuses can potentially remove an additional $H^2$ factor from both the bounds. To the best of our knowledge, applying Bernstein-style techniques to SOMG and related methods is fairly standard, though it requires careful and rather extensive bookkeeping. Additionally, as briefly discussed in the appendix, with a modified projection operator, SOMG achieves an $H^5$ dependence when specialized the single-agent setting. However, since no matching lower bounds are known, it is unclear whether these dependencies are tight. We believe that both the single-agent bound and the bound for two-player zero-sum Markov games are likely loose and can be further improved. Establishing lower bounds, alongside tighter upper bounds thus remains an important open direction.
>
> > **Experimental results (W2)**
>
> We added small scale experiments (in the plots folder) as suggested here
> https://anonymous.4open.science/r/SOMG_ICML-FA03/README.md for randomly generated MDPs
> in order to show that SOMG stabilizes learning. We will add this in the camera ready appendix.
>
> > **About extension to CCE for General sum games (W3 and Q3)**
>
> We currently donot have an answer to what algorithm will work well for CCE, an additional challenge in CCE is the non smoothness of the CCE value to a change in Q function matrix [1] (smoothness in Q function and policy is used in both single agent and two player game analysis)
>
> This is an important point, though addressing it is beyond the scope of this paper and left out for future work. At present, most self-play work focuses on the two-player zero-sum setting, which motivated us to frame the discussion in those terms.
>
> > **About the $\beta =0$ case (Q2)**
>
> This is a good point, one can do a superficial fix (extra step in analysis for the same proposed algorithm)  for this specific case here, i.e., choose a '$\beta$' close to zero (eg: $\beta' = $ $10^{-8}$) and the regret at $\beta=0$ will be equal to regret at $\beta' =10^{-8}$ + a linear in $\beta'$ error term (also $\sqrt{T}$ dependence), so this will produce the same $\mathcal{O}(\sqrt{T})$ bound for the $\beta =0$ case as the one achieved by methods designed specifically for unregularized setting.
>
> That being said, the bound is probably loose in that sense (extra $\log(\beta^{-1})$ dependence) which is a artifact of the function class induced by our algorithm.
>
> Assuming a finite covering number, as is common in many general function approximation (GFA) settings, can implicitly hide this dependency via the complexity of the function class. Unlike in linear function approximation, where the covering number is derived explicitly, in GFA it is typically imposed as an assumption on the function class. SOMG gracefully extends to GFA, we will add a discussion about extension to general function classes in the appendix. We focused on the linear setting primarily to simplify notation and maintain clarity, allowing readers to pay attention to the core method rather than the additional technical machinery associated with GFA (e.g., eluder dimension, confidence set construction, covering number etc). Our choice was therefore expository rather than fundamental, and the algorithms extend naturally beyond the linear case.
>
> ### References
>
> - [1] Xie, Qiaomin, et al. *"Learning zero-sum simultaneous-move Markov games using function approximation and correlated equilibrium."* Conference on Learning Theory. PMLR, 2020.

---

> > ### Author Rebuttal · Reviewer_J2Zh · 2026-04-01
> >
> > Thank you for the response. I do not have follow-up questions. I remain positive about the work as in my initial evaluation.

---

> > > ### Author Response · Authors · 2026-04-03
> > >
> > > Thank you for your support and for acknowledging our rebuttal. We are glad to hear that it resolved all your questions.

---

### Decision · Program_Chairs · 2026-04-30

**Decision:**

Accept (regular)

**Comment:**

This paper makes a significant theoretical contribution by bridging the gap between single-agent reinforcement learning and game-theoretic settings regarding KL regularization. The authors successfully demonstrate that anchoring policies to a fixed reference can yield logarithmic regret in two-player zero-sum Markov games, a substantial improvement over the standard sublinear rates.

Reviewers have a consensus that establishing logarithmic regret in this setting is a major contribution that fills a well-identified gap in the literature. They specifically highlighted the non-trivial nature of the "superoptimistic" bonuses and best-response sampling required to handle the lack of closed-form Nash equilibrium policies. While some concerns regarding computational "coupling" versus "decoupled" methods (like Mirror Descent) were raised, reviewers agree that the sample complexity benefits are conceptually important and provide a rigorous foundation for modern self-play practices.

Overall, the paper is technically sound, well-presented, and addresses a timely problem in the theory of multi-agent learning.